# Selective Sampling for Online Best-arm Identification

**Romain Camilleri**[*], **Zhihan Xiong**[*], **Maryam Fazel, Lalit Jain, Kevin Jamieson**
University of Washington, Seattle, WA
{camilr,zhihanx,mfazel,lalitj,jamieson}@uw.edu

## Abstract

This work considers the problem of *selective-sampling for best-arm identification.* Given a set of potential options $\mathcal{Z} \subset \mathbb{R}^d$, a learner aims to compute with probability greater than $1 - \delta$, $\arg\max_{z \in \mathcal{Z}} z^\top \theta_*$ where $\theta_*$ is unknown. At each time step, a potential measurement $x_t \in \mathcal{X} \subset \mathbb{R}^d$ is drawn IID and the learner can either choose to take the measurement, in which case they observe a noisy measurement of $x^\top \theta_*$, or to abstain from taking the measurement and wait for a potentially more informative point to arrive in the stream. Hence the learner faces a fundamental trade-off between the number of labeled samples they take and when they have collected enough evidence to declare the best arm and stop sampling. The main results of this work precisely characterize this trade-off between labeled samples and stopping time and provide an algorithm that nearly-optimally achieves the minimal label complexity given a desired stopping time. In addition, we show that the optimal decision rule has a simple geometric form based on deciding whether a point is in an ellipse or not. Finally, our framework is general enough to capture binary classification improving upon previous works.

## 1 Introduction

In this work we consider *selective sampling for online best-arm identification.* In this setting, at every time step $t = 1, 2, \ldots$, Nature reveals a potential measurement $x_t \in \mathcal{X} \subset \mathbb{R}^d$ to the learner. The learner can choose to either *query* $x_t$ ($\xi_t = 1$) or abstain ($\xi_t = 0$) and immediately move on to the next time. If the learner chooses to take a query ($\xi_t = 1$), then Nature reveals a noisy linear measurement of an unknown $\theta_* \in \mathbb{R}^d$, i.e. $y_t = \langle x_t, \theta_* \rangle + \epsilon_t$ where $\epsilon_t$ is mean zero sub-Gaussian noise. Before the start of the game, the learner has knowledge of a set $\mathcal{Z} \subset \mathbb{R}^d$. The objective of the learner is to identify $z_* := \arg\max_{z \in \mathcal{Z}} \langle z, \theta_* \rangle$ with probability at least $1 - \delta$ at a learner specified stopping time $\mathcal{U}$. It is desirable to minimize both the stopping time $\mathcal{U}$ which counts the total number of unlabeled or labeled queries and the number of labeled queries requested $\mathcal{L} := \sum_{t=1}^{\mathcal{U}} \mathbf{1}\{\xi_t = 1\}$. In this setting, at each time $t$ the learner must make the decision of whether to accept the available measurement $x_t$, or abstain and wait for an even more informative measurement. While abstention may result in a smaller total labeled sample complexity $\mathcal{L}$, the stopping time $\mathcal{U}$ may be very large. This paper characterizes the set of feasible pairs $(\mathcal{U}, \mathcal{L})$ that are necessary and sufficient to identify $z_*$ with probability at least $1 - \delta$ when $x_t$ are drawn IID at each time $t$ from a distribution $\nu$. Moreover, we propose an algorithm that nearly obtains the minimal information theoretic label sample complexity $\mathcal{L}$ for any desired unlabeled sample complexity $\mathcal{U}$.

While characterizing the sample complexity of selective sampling for online best arm identification is the primary theoretical goal of this work, the study was initially motivated by fundamental questions about how to optimally trade-off the value of information versus time. Even for this idealized linear setting, it is far from obvious a priori what an optimal decision rule $\xi_t$ looks like and if it can even be succinctly described, or if it is simply the solution to an opaque optimization problem. Remarkably,

---

[*]Equal contribution. Alphabetical order.

35th Conference on Neural Information Processing Systems (NeurIPS 2021).

we show that for every feasible, optimal operating pair $(\mathcal{U}, \mathcal{L})$ there exists a matrix $A \in \mathbb{R}^{d \times d}$ such that the optimal decision rule takes on the form $\xi_t = \mathbf{1}\{x^\top A x \geq 1\}$ when $x_t \sim \nu$ iid. The fact that for any smooth distribution $\nu$ the decision rule is a hard decision equivalent to $x_t$ falling outside a fixed ellipse or not, and not a stochastic rule that varies complementarily with the density of $\nu$ over space is perhaps unexpected.

To motivate the problem description, suppose on each day $t = 1, 2, \ldots$ a food blogger posts the *Cocktail of the Day* with a recipe described by a feature vector $x_t \in \mathbb{R}^d$. You have the ingredients (and skills) to make any possible cocktail in the space of all cocktails $\mathcal{Z}$, but you don't know which one you'd like the most, i.e., $z_* := \arg\max_{z \in \mathcal{Z}} \langle z, \theta_* \rangle$, where $\theta_*$ captures your preferences over cocktail recipes. You decide to use the *Cocktail of the Day* to inform your search. That is, each day you are presented with the cocktail recipe $x_t \in \mathbb{R}^d$, and if you choose to make it ($\xi_t = 1$) you observe your preference for the cocktail $y_t$ with $\mathbb{E}[y_t] = \langle x_t, \theta_* \rangle$. Of course, making cocktails can get costly, so you don't want to make each day's cocktail, but rather you will only make the cocktail if $x_t$ is informative about $\theta_*$ (e.g., uses a new combination of ingredients). At the same time, waiting too many days before making the next cocktail of the day may mean that you never get to learn (and hence drink) the cocktail $z_*$ you like best. The setting above is not limited to cocktails, but rather naturally generalizes to discovering the efficacy of drugs and other therapeutics where blood and tissue samples come to the clinic in a stream and the researcher has to choose whether to take a potentially costly measurement.

Our results hold for arbitrary $\theta_* \in \mathbb{R}^d$, sets $\mathcal{X} \subset \mathbb{R}^d$ and $\mathcal{Z} \subset \mathbb{R}^d$, and measures $\nu \in \triangle_{\mathcal{X}}$[1] for which we assume $x_t \sim \nu$ is drawn IID. The assumption that each $x_t$ is IID allows us to make very strong statements about optimality. To summarize, our contributions are as follows:

- We present fundamental limits on the trade-off between the amount of unlabelled data and labelled data in the form of (the first) information theoretic lower bounds for selective sampling problems that we are aware of. Naturally, they say that there is an absolute minimum amount of unlabelled data that is necessary to solve the problem, but then for any amount of unlabelled data beyond this critical value, the bounds say that the amount of labelled data must exceed some value as a function of the unlabelled data used.

- We propose an algorithm that nearly matches the lower bound at all feasible trade-off points in the sense that given any unlabelled data budget that exceeds the critical threshold, the algorithm takes no more labels than the lower bound suggests. Thus, the upper and lower bounds sketch out a curve of all possible operating points, and the algorithm achieves any point on this curve.

- We characterize the optimal decision rule of whether to take a sample or not, based on any critical point is a simple test: Accept $x_t \in \mathbb{R}^d$ if $x_t^\top A x_t \geq 1$ for some matrix $A$ that depends on the desired operating point and geometry of the task. Geometrically, this is equivalent to $x_t$ falling inside or outside an ellipsoid.

- Our framework is also general enough to capture binary classification, and consequently, we prove results there that improve upon state of the art.

## 1.1 Related Work

**Selective Sampling in the Streaming Setting:** Online prediction, the setting in which the selective sampling framework was introduced, is a closely related problem to the one studied in this paper and enjoys a much more developed literature [6, 9, 1, 7]. In the linear online prediction setting, for $t = 1, 2, \ldots$ Nature reveals $x_t \in \mathbb{R}^d$, the learner predicts $\widehat{y}_t$ and incurs a loss $\ell(\widehat{y}_t, y_t)$, and then the learner decides whether to observe $y_t$ (i.e., $\xi_t = 1$) or not ($\xi_t = 0$), where $y_t$ is a label generated by a composition of a known link function with a linear function of $x_t$. For example, in the classification setting [1, 6, 9], one setting assumes $y_t \in \{-1, 1\}$ with $\mathbb{E}[y_t | x_t] = \langle x_t, \theta_* \rangle$ for some unknown $\theta_* \in \mathbb{R}^d$, and $\ell(\widehat{y}_t, y_t) = \mathbf{1}\{\widehat{y}_t \neq y_t\}$. In the regression setting [7], one observes $y_t \in [-1, 1]$ with $\mathbb{E}[y_t | x_t] = \langle x_t, \theta_* \rangle$ again, and $\ell(\widehat{y}_t, y_t) = (\widehat{y}_t - y_t)^2$. After any amount of time $\mathcal{U}$, the learner is incentivized to minimize both the amount of requested labels $\sum_{t=1}^{\mathcal{U}} \mathbf{1}\{\xi_t = 1\}$ and the cumulative loss $\sum_{t=1}^{\mathcal{U}} \ell(y_t, \widehat{y}_t)$ (or some measure of regret which compares to predictions using the unknown $\theta_*$). If every label $y_t$ is requested then $\mathcal{L} = \mathcal{U}$ and this is just the classical online learning setting.

---

[1]We denote the set of probability measures over $\mathcal{X}$ as $\triangle_{\mathcal{X}}$.

These works give a guarantee on the regret and labeled points taken in terms of the hardness of the stream relative to a learner which would see the label at every time. Most do not give the learner the ability to select an operating point that provides a trade-off between the amount of unlabeled versus labeled data taken. Those few works that propose algorithms that do provide this functionality do not provide lower bounds that match their given upper bounds, leaving it unclear whether their algorithm optimally negotiates this trade-off. In contrast, our work fully characterizes the trade-off between the amount of unlabeled and labeled data through an information-theoretic lower bound and a matching upper bound. Specifically, our algorithm includes a tuning parameter, call it $\tau$, that controls the trade-off between the evaluation metric of interest (for us, the quality of the recommended $z \in \mathcal{Z}$), the label complexity $\mathcal{L}$, and the amount of unlabelled data $\mathcal{U}$ that is necessary before the metric of interest can be non-trivial. We prove that each possible setting of $\tau$ parametrizes *all* possible trade-offs between unlabeled and labeled data.

Our work is perhaps closest to the streaming setting for agnostic active classification [8, 15] where each $x_s$ is drawn i.i.d. from an underlying distribution $\nu$ on $\mathcal{X}$, and indeed our results can be specialized to this setting as we discuss in Section 3. These papers also evaluate themselves at a single point on the tradeoff curve, namely the number of samples needed in passive supervised learning to obtain a learner with excess risk at most $\epsilon$. They provide minimax guarantees on the amount of labeled data needed in terms of the disagreement coefficient [12]. In contrast, again, our results characterize the full trade-off between the amount of unlabeled data seen, and the amount of labeled data needed to achieve the target excess risk $\epsilon$. We note that using online-to-batch conversion methods, [9, 1, 6] also provide results on the amount of labeled data needed but they assume a very specific parametric form to their label distribution unlike our setting which is agnostic. Other works have characterized selective sampling for classification in the realizable setting that assumes there exists a classifer among the set under consideration that perfectly labels every $y_t$ [13]–our work addresses the agnostic setting where no such assumption is made. Finally, our results apply under the more general setting of *domain adaptation under covariate shift* where we are observing data drawn from the stream $\nu$, but we will evaluate the excess risk of our resulting classifier on a different stream $\pi$ [22, 23, 26].

**Best-Arm Identification and Online Experimental Design.** Our techniques are based on experimental design methods for best-arm identification in linear bandits, see [24, 11, 5]. In the setting of these works, there exists a pool of examples $\mathcal{X}$ and at each time any $x \in \mathcal{X}$ can be selected with replacement. The goal is to identify the best arm using as few total selections (labels) as possible. Their algorithms are based on arm-elimination. Specifically, they select examples with probability proportional to an approximate $G$-optimal design with respect to the current remaining arms. Then, during each round after taking measurements, those arms with high probability of being suboptimal will be eliminated. Remarkably, near-optimal sample complexity has been achieved under this setting. While we apply these techniques of arm-elimination and sampling through $G$-optimal design, the major difference is that we are facing a stream instead of a pool of examples. Finally, [10] considers a different online experiment design setup where (adversarially chosen) experiments arrive sequentially and a primal-dual algorithm decides whether to choose each, subject to a total budget. [10] studies the competitive ratio of such algorithms (in the manner of online packing algorithms) for problems such as $D$-optimal experiment design.

## 2  Selective Sampling for Best Arm Identification

Consider the following game: Given known $\mathcal{X}, \mathcal{Z} \subset \mathbb{R}^d$ and unknown $\theta_* \in \mathbb{R}^d$ at each time $t = 1, 2, \ldots$:

1. Nature reveals $x_t \overset{iid}{\sim} \nu$ with support$(\nu) = \mathcal{X}$
2. Player chooses $Q_t \in \{0, 1\}$. If $Q_t = 1$ then nature reveals $y_t$ with $\mathbb{E}[y_t] = \langle x_t, \theta_* \rangle$
3. Player optionally decides to stop at time $t$ and output some $\widehat{z} \in \mathcal{Z}$

If the player stops at time $\mathcal{U}$ after observing $\mathcal{L} = \sum_{t=1}^{\mathcal{U}} Q_t$ labels, the objective is to identify $z_* = \arg\max_{z \in \mathcal{Z}} \langle z, \theta_* \rangle$ with probability at least $1 - \delta$ while minimizing a trade-off of $\mathcal{U}, \mathcal{L}$.

This paper studies the relationship between $\mathcal{U}$ and $\mathcal{L}$ in the context of necessary and sufficient conditions to identify $z_*$ with probability at least $1 - \delta$. Clearly $\mathcal{U}$ must be "large enough" for $z_*$ to

be identifiable even if all labels are requested (i.e., $\mathcal{L} = \mathcal{U}$). But if $\mathcal{U}$ is very large, the player can start to become more picky with their decision to observe the label or not. Indeed, one can easily imagine scenarios in which it is advantageous for a player to forgo requesting the label of the current example in favor of waiting for a more informative example to arrive later if they wished to minimize $\mathcal{L}$ alone. Intuitively, $\mathcal{L}$ should decrease as $\mathcal{U}$ increases, but how?

Any selective sampling algorithm for the above protocol at time $t$ is defined by 1) a selection rule $P_t : \mathcal{X} \to [0, 1]$ where $Q_t \sim \text{Bernoulli}(P_t(x_t))$, 2) a stopping rule $\mathcal{U}$, and 3) a recommendation rule $\widehat{z} \in \mathcal{Z}$. The algorithm's behavior at time $t$ can use all information collected up to time $t$

**Definition 1.** *For any $\delta \in (0, 1)$ we say a selective sampling algorithm is $\delta$-PAC for $\nu \in \triangle_{\mathcal{X}}$ if for all $\theta \in \mathbb{R}^d$ the algorithm terminates at time $\mathcal{U}$ which is finite almost surely and outputs $\arg\max_{z \in \mathcal{Z}} \langle z, \theta \rangle$ with probability at least $1 - \delta$.*

## 2.1 Optimal design

Before introducing our own algorithm, let us consider a seemingly optimal procedure. For any $\lambda \in \triangle_{\mathcal{X}} = \{p : \sum_{x \in \mathcal{X}} p_x = 1, \ p_x \geq 0 \ \forall x \in \mathcal{X}\}$ define

$$\rho(\lambda) := \max_{z \in \mathcal{Z} \setminus \{z_*\}} \frac{\|z - z_*\|^2_{\mathbb{E}_{X \sim \lambda}[XX^\top]^{-1}}}{\langle \theta_*, z_* - z \rangle^2}. \tag{1}$$

Intuitively, $\rho(\lambda)$ captures the number of labeled examples drawn from distribution $\lambda$ to identify $z_*$. Specifically, for any $\tau \geq \rho(\lambda) \log(|\mathcal{Z}|/\delta)$, if $x_1, \ldots, x_\tau \sim \lambda$ and $y_i = \langle x_i, \theta_* \rangle + \epsilon_i$ where $\epsilon_i$ is iid 1 sub-Gaussian noise, then there exists an estimator $\widehat{\theta} := \widehat{\theta}(\{(x_i, y_i)\}_{i=1}^\tau)$ such that $\langle \widehat{\theta}, z_* \rangle > \max_{z \in \mathcal{Z} \setminus z_*} \langle \widehat{\theta}, z \rangle$ with probability at least $1 - \delta$ [11]. In particular, $\tau \geq \rho(\lambda) \log(|\mathcal{Z}|/\delta)$ samples suffice to guarantee that $\arg\max_{z \in \mathcal{Z}} \langle \widehat{\theta}, z \rangle = \arg\max_{z \in \mathcal{Z}} \langle \theta_*, z \rangle =: z_*$.

Thus, if our $\tau$ samples are coming from $\nu$, we would expect any reasonable algorithm to require at least $\rho(\nu) \log(|\mathcal{Z}|/\delta)$ examples and labels. However, since we only want to take informative examples, we instead choose to select the $t$th example $x_t = x$ according to a probability $P(x)$ so that our final labeled samples are coming from the distribution $\lambda$ where $\lambda(x) \propto P(x)\nu(x)$. In particular, $P(x)$ should be chosen according to the following optimization problem

$$P^* = \underset{P : \mathcal{X} \to [0,1]}{\arg\min} \ \tau \mathbb{E}_{X \sim \nu}[P(X)] \quad \text{subject to} \quad \max_{z \in \mathcal{Z} \setminus \{z_*\}} \frac{\|z_* - z\|^2_{\mathbb{E}_{X \sim \nu}[\tau P(X) X X^\top]^{-1}}}{\langle z_* - z, \theta_* \rangle^2} \beta_\delta \leq 1 \quad (2)$$

for $\beta_\delta = \log(|\mathcal{Z}|/\delta)$ where the objective captures the number of samples we select using $P^*$, and the constraint captures the fact that we have solved the problem. Remarkably, we can reparametrize this result in terms of an optimization problem over $\lambda \in \Delta_{\mathcal{X}}$ instead of $P^* : \mathcal{X} \to [0, 1]$ as

$$\tau \mathbb{E}_{X \sim \nu}[P^*(X)] = \min_{\lambda \in \triangle_{\mathcal{X}}} \rho(\lambda) \beta_\delta \quad \text{subject to} \quad \tau \geq \|\lambda/\nu\|_\infty \rho(\lambda) \beta_\delta$$

where $\|\lambda/\nu\|_\infty = \max_{x \in \mathcal{X}} \lambda(x)/\nu(x)$, as shown in Proposition 2. Note that as $\tau \to \infty$ the constraint becomes inconsequential. Also notice that $\rho(\nu)\beta_\delta$ appears to be a necessary amount of labels to solve the problem even if $P(x) \equiv 1$ (albeit, by arguing about minimizing the upperbound of above).

## 2.2 Main results

In this section we formally justify the sketched argument of the previous section, showing nearly matching upper and lower bounds.

**Theorem 1** (Lower bound). *Fix any $\delta \in (0, 1)$, $\mathcal{X}, \mathcal{Z} \subset \mathbb{R}^d$, and $\theta_* \in \mathbb{R}^d$. Any selective sampling algorithm that is $\delta$-PAC for $\nu \in \triangle_{\mathcal{X}}$ and terminates after drawing $\mathcal{U}$ unlabelled examples from $\nu$ and requests the labels of just $\mathcal{L}$ of them satisfies*

- $\mathbb{E}[\mathcal{U}] \geq \rho(\nu) \log(1/\delta)$*, and*

- $\mathbb{E}[\mathcal{L}] \geq \min_{\lambda \in \triangle_{\mathcal{X}}} \rho(\lambda) \log(1/\delta) \quad \text{subject to} \quad \mathbb{E}[\mathcal{U}] \geq \|\lambda/\nu\|_\infty \rho(\lambda) \log(1/\delta)$.

The first part of the theorem quantifies the number of rounds or unlabelled draws $\mathcal{U}$ that *any* algorithm must observe before it could hope to stop and output $z_*$ correctly. The second part describes a

trade-off between $\mathcal{U}$ and $\mathcal{L}$. One extreme is if $\mathbb{E}[\mathcal{U}] \to \infty$, which effectively removes the constraint so that the number of observed labels must scale like $\min_{\lambda \in \triangle_{\mathcal{X}}} \rho(\lambda) \log(1/\delta)$. Note that this is precisely the number of labels required in the pool-based setting where the agent can choose *any* $x \in \mathcal{X}$ that she desires at each time $t$ (e.g. [11]). In the other extreme, $\mathbb{E}[\mathcal{U}] = \rho(\nu) \log(1/\delta)$ so that the constraint in the label complexity $\mathbb{E}[\mathcal{L}]$ is equivalent to $\rho(\nu) \geq \|\lambda/\nu\|_{\infty} \rho(\lambda)$. This implies that the minimizing $\lambda$ must either stay very close to $\nu$, or must obtain a substantially smaller value of $\rho(\lambda)$ relative to $\rho(\nu)$ to account for the inflation factor $\|\lambda/\nu\|_{\infty}$. In some sense, this latter extreme is the most interesting point on the trade-off curve because its asking the algorithm to stop as quickly as the algorithm that observes all labels, but after requesting a minimal number of labels. Note that this lower bound holds even for algorithms that known $\nu$ exactly. The proof of Theorem 1 relies on standard techniques from best arm identification lower bounds (see e.g. [17, 11]).

Remarkably, every point on the trade-off suggested by the lower bound is nearly achievable.

**Theorem 2** (Upper bound). *Fix any $\delta \in (0,1)$, $\mathcal{X}, \mathcal{Z} \subset \mathbb{R}^d$, and $\theta_* \in \mathbb{R}^d$. Let $\Delta = \min_{z \in \mathcal{Z} \setminus \{z_*\}} \langle z_* - z, \theta_* \rangle$ and $\beta_\delta \propto \log(\log(\frac{1}{\Delta})|\mathcal{Z}|/\delta)$ where the precise constant is given in the appendix. For any $\tau \geq \rho(\nu) \beta_\delta$ there exists a $\delta$-PAC selective sampling algorithm that observes $\mathcal{U}$ unlabeled examples and requests just $\mathcal{L}$ labels that satisfies with probability at least $1 - \delta$*

- $\mathcal{U} \leq \log_2(\frac{4}{\Delta}) \tau$, *and*

- $\mathcal{L} \leq 3 \log_2(\frac{4}{\Delta}) \min_{\lambda \in \triangle_{\mathcal{X}}} \rho(\lambda) \beta_\delta$   *subject to*   $\tau \geq \|\lambda/\nu\|_{\infty} \rho(\lambda) \beta_\delta$.

Aside from the $\log(\frac{1}{\Delta})$ factor and the $\log(|\mathcal{Z}|)$ that appears in the $\beta_\delta$ term, this nearly matches the lower bound. Note that the parameter $\tau$ parameterizes the algorithm and makes the trade-off between $\mathcal{U}$ and $\mathcal{L}$ explicit. The next section describes the algorithm that achieves this theorem.

## 2.3   Selective Sampling Algorithm

Algorithm 1 contains the pseudo-code of our selective sampling algorithm for best-arm identification. Note that it takes a confidence level $\delta \in (0, 1)$ and a parameter $\tau$ that controls the unlabeled-labeled budget trade-off as input. The algorithm is effectively an elimination style algorithm and closely mirrors the RAGE algorithm for the pool-based setting of best-arm identification problem [11]. The key difference, of course, is that instead of being able to plan over the pool of measurements, this algorithm must plan over the $x$'s that the algorithm may *potentially* see and account for the case that it might not see the $x$'s it wants.

---

**Algorithm 1** Selective Sampling for Best-arm Identification

1: **Input** $\mathcal{Z} \subset \mathbb{R}^d, \delta \in (0,1), \tau$
2: **while** $|\mathcal{Z}_\ell| \geq 1$ **do**
3:    Let $\widehat{P}_\ell, \widehat{\Sigma}_{\widehat{P}_\ell} \leftarrow$ OPTIMIZEDESIGN$(\mathcal{Z}_\ell, 2^{-\ell}, \tau)$ // $\widehat{\Sigma}_{\widehat{P}_\ell}$ approximates $\mathbb{E}_{X \sim \nu}[\widehat{P}_\ell(X) X X^\top]$
4:    **for** $t = (\ell-1)\tau + 1, \ldots, \ell\tau$ **do**
5:       Nature reveals $x_t$ drawn iid from $\nu$ (with support $\mathbb{R}^d$)
6:       Sample $Q_t(x_t) \sim$ Bernoulli$(\widehat{P}_\ell(x_t))$. If $Q_t = 1$ then observe $y_t$    // $\mathbb{E}[y_t|x_t] = \langle \theta_*, x_t \rangle$
7:    **end for**
8:    Let $\widehat{\theta}_\ell \leftarrow$ RIPS$(\{\widehat{\Sigma}_{\widehat{P}_\ell}^{-1} Q_s(x_s) x_s y_s\}_{s=(\ell-1)\tau+1}^{\ell\tau}, \mathcal{Z} \times \mathcal{Z})$          // $\widehat{\theta}_\ell$ approximates $\theta_*$
9:    $\mathcal{Z}_{\ell+1} = \mathcal{Z}_\ell \setminus \{z \in \mathcal{Z}_\ell : \max_{z' \in \mathcal{Z}_\ell} \langle z' - z, \widehat{\theta}_\ell \rangle \geq 2^{-\ell}\}$
10: **end while**

---

In round $\ell$, the algorithm maintains an active set $\mathcal{Z}_\ell \subseteq \mathcal{Z}$ with the guarantee that each remaining $z \in \mathcal{Z}_\ell$ satisfies, $\langle z_* - z, \theta_* \rangle \leq 8 \cdot 2^{-\ell}$. In each round, on Line 3 of the algorithm, it calls out to a sub-routine OPTIMIZEDESIGN$(\mathcal{Z}, \epsilon, \tau)$ that is trying to approximate the ideal optimal design of (2). In particular, the ideal response to OPTIMIZEDESIGN$(\mathcal{Z}, \epsilon, \tau)$ would return a $P_\epsilon^*$ and $\Sigma_{P^*} = \mathbb{E}_{X \sim \nu}[P_\epsilon^*(X) X X^\top]$ where $P_\epsilon^*$ is the solution to Equation 2 with the one exception that the denominator of the constraint is replaced with $\max\{\epsilon^2, \langle \theta_*, z_* - z \rangle^2\}$. Of course, $\theta_*$ is unknown so we cannot solve Equation 2 (as well as other outstanding issues that we will address shortly). Consequently, our implementation will aim to *approximate* the optimization problem of Equation 2.

But assuming our sample complexity is not too far off from this ideal, each round should not request more labels than the number of labels requested by the ideal program with $\epsilon = 0$. Thus, the total number of samples should be bounded by the ideal sample complexity times the number of rounds, which is $O(\log(\Delta^{-1}))$. We will return to implementation issues in the next section.

Assuming we are returned $(\widehat{P}_\ell, \widehat{\Sigma}_{\widehat{P}_\ell})$ that approximate their ideals as just described, the algorithm then proceeds to process the incoming stream of $x_t \sim \nu$. As described above, the decision to request the label of $x_t$ is determined by a coin flip coming up heads with probability $\widehat{P}_\ell(x_t)$–otherwise we do not request the label. Given the collected dataset $\{(x_t, y_t, Q_t, \widehat{P}_\ell(x_t))\}_t$, line 8 then computes an estimate $\widehat{\theta}_\ell$ of $\theta_*$ using the RIPS estimator of [5] which will satisfy

$$|\langle z_* - z, \widehat{\theta}_\ell - \theta_* \rangle| \leq O\left( \|z_* - z\|_{\mathbb{E}_{X \sim \nu}[\tau \widehat{P}_\ell(X) X X^\top]^{-1}} \sqrt{\log(2\ell^2|\mathcal{Z}|^2/\delta)} \right) \leq 2^{-\ell}$$

for all $z \in \mathcal{Z}_\ell$ simultaneously with probability at least $1 - \delta$. Thus, the final line of the algorithm eliminates any $z \in \mathcal{Z}_\ell$ such that there exists another $z' \in \mathcal{Z}_\ell$ (think $z_*$) that satisfies $\langle \widehat{\theta}_\ell, z' - z \rangle > 2^{-\ell}$. The process continues until $\mathcal{Z}_\ell = \{z_*\}$.

### 2.4 Implementation of OPTIMIZEDESIGN

For the subroutine OPTIMIZEDESIGN passed $(\mathcal{Z}_\ell, \epsilon, \tau)$ the next best thing to computing Equation 2 with the denominator of the constraint replaced with $\max\{\epsilon^2, \langle \theta_*, z_* - z \rangle^2\}$, is to compute

$$P_\epsilon = \underset{P:\mathcal{X} \to [0,1]}{\operatorname{argmin}} \ \mathbb{E}_{X \sim \nu}[P(X)] \text{ subject to } \max_{z,z' \in \mathcal{Z}_\ell} \frac{\|z - z'\|^2_{\mathbb{E}_{X \sim \nu}[\tau P(X) X X^\top]^{-1}}}{\epsilon^2} \beta_\delta \leq 1 \quad (3)$$

and $\Sigma_{P_\epsilon} = \mathbb{E}_{X \sim \nu}[P_\epsilon(X) X X^\top]$ for an appropriate choice of $\beta_\delta = \Theta(\log(|\mathcal{Z}|/\delta))$. To see this, firstly, any $z \in \mathcal{Z}$ with gap $\langle \theta_*, z_* - z \rangle$ that we could accurately estimate would not be included in $\mathcal{Z}_\ell$, thus we don't need it in the $\max$ of the denominator. Secondly, to get rid of $z_*$ in the numerator (which is unknown, of course), we note that for any norm $\max_{z,z'} \|z - z'\| \leq \max_z 2\|z - z_*\| \leq \max_{z,z'} 2\|z - z'\|$. Assuming we could solve this directly and compute $\Sigma_{P_\epsilon} = \mathbb{E}_{X \sim \nu}[P_\epsilon(X) X X^\top]$, we can obtain the result of Theorem 2 (proven in the Appendix).

However, even if we knew $\nu$ exactly, the optimization problem of Equation 3 is quite daunting as it is a potentially infinite dimensional optimization problem over $\mathcal{X}$. Fortunately, after forming the Lagrangian with dual variables for each $z - z' \in \mathcal{Z} \times \mathcal{Z}$, optimizing the dual amounts to a finite dimensional optimization problem over the finite number of dual variables. Moreover, this optimization problem is maximizing a simple expectation with respect to $\nu$ and thus we can apply standard stochastic gradient ascent and results from stochastic approximation [20]. Given the connection to stochastic approximation, instead of sampling a fresh $\widetilde{x} \sim \nu$ each iteration, it suffices to "replay" a sequence of $\widetilde{x}$'s from historical data. Summing up, this construction allows us to compute a satisfactory $P_\epsilon$ and avoid both an infinite-dimensional optimization problem and requiring knowledge of $\nu$ (as long as historical data is available).

Meanwhile, with historical data, we can also empirically compute $\mathbb{E}_{X \sim \nu}[P_\epsilon(X) X X^\top]$. Historical data could mean offline samples from $\nu$ or just samples from previous rounds. In this setting, Theorem 2 still holds albeit with larger constants. Theorem 7 in the appendix characterizes the necessary amount of historical data needed. Unfortunately (in full disclosure) the theoretical guarantees on the amount of historical data needed is absurdly large, though we suspect this arises from a looseness in our analysis. Similar assumptions and approaches to historical or offline data have been used in other works in the streaming setting e.g. [15].

## 3 Selective Sampling for Binary Classification

We now review streaming Binary Classification in the agnostic setting [8, 12, 15] and show that our approach can be adapted to this setting. Consider a binary classification problem where $\mathcal{X}$ is the example space and $\mathcal{Y} = \{-1, 1\}$ is the label space. Fix a hypothesis class $\mathcal{H}$ such that each $h \in \mathcal{H}$ is a classifier $h : \mathcal{X} \to \mathcal{Y}$. Assume there exists a fixed regression function $\eta : \mathcal{X} \to [0, 1]$ such that the label of $x$ is Bernoulli with probability $\eta(x) = \mathbb{P}(Y = 1|X = x)$. Being in the agnostic setting, we make no assumption on the relationship between $\mathcal{H}$ and $\eta$. Finally, fix any $\nu \in \triangle_\mathcal{X}$ and $\pi \in \triangle_\mathcal{X}$. Given known $\mathcal{X}, \mathcal{H}$ and unknown regression function $\eta$, at each time $t = 1, 2, \dots$:

1. Nature reveals $x_t \sim \nu$

2. Player chooses $Q_t \in \{0, 1\}$. If $Q_t = 1$ then nature reveals $y_t \sim \text{Bernoulli}(\eta(x_t)) \in \{-1, 1\}$

3. Player optionally decides to stop at time $t$ and output some $\widehat{h} \in \mathcal{H}$.

Define the *risk* of any $h \in \mathcal{H}$ as $R_\pi(h) := \mathbb{P}_{X \sim \pi, Y \sim \eta(X)}(Y \neq h(X))$. If the player stops at time $\mathcal{U}$ after observing $\mathcal{L} = \sum_{t=1}^{\mathcal{U}} Q_t$ labels, the objective is to identify $h_* = \arg\min_{h \in \mathcal{H}} R_\pi(h)$ with probability at least $1 - \delta$ while minimizing a trade-off of $\mathcal{U}, \mathcal{L}$. Note that $h_*$ is the true risk minimizer with respect to distribution $\pi$ but we observe samples $x_t \sim \nu$; $\pi$ is not necessarily equal to $\nu$. While we have posed the problem as identifying the potentially unique $h^*$, our setting naturally generalizes to identifying an $\epsilon$-good $h$ such that $R_\pi(h) - R_\pi(h_*) \leq \epsilon$.

We will now reduce selective sampling for binary classification problem to selective sampling for best arm identification, and thus immediately obtain a result on the sample complexity. For simplicity, assume that $\mathcal{X}$ and $\mathcal{H}$ are finite. Enumerate $\mathcal{X}$ and for each $h \in \mathcal{H}$ define a vector $z^{(h)} \in [0, 1]^{|\mathcal{X}|}$ such that $z_x^{(h)} := \pi(x)\mathbf{1}\{h(x) = 1\}$ for $z^{(h)} = [z_x^{(h)}]_{x \in \mathcal{X}}$. Moreover, define $\theta^* := [\theta_x^*]_{x \in \mathcal{X}}$ where $\theta_x^* := 2\eta(x) - 1$. Then

$$R_\pi(h) = \mathbb{E}_{X \sim \pi, Y \sim \eta(X)}[\mathbf{1}\{Y \neq h(X)\}] = \sum_{x \in \mathcal{X}} \pi(x)(\eta(x)\mathbf{1}\{h(x) \neq 1\} + (1 - \eta(x))\mathbf{1}\{h(x) \neq 0\})$$

$$= \sum_{x \in \mathcal{X}} \pi(x)\eta(x) + \sum_{x \in \mathcal{X}} \pi(x)(1 - 2\eta(x))\mathbf{1}\{h(x) = 1\} = c - \langle z^{(h)}, \theta^* \rangle$$

where $c = \sum_{x \in \mathcal{X}} \pi(x)\eta(x)$ does not depend on $h$. Thus, if $\mathcal{Z} := \{z^{(h)}\}_{h \in \mathcal{H}}$ then identifying $h_* = \arg\min_{h \in \mathcal{H}} R_\pi(h)$ is equivalent to identifying $z_* = \arg\max_{z \in \mathcal{Z}} \langle z, \theta^* \rangle$. We can now apply Theorem 2 to obtain a result describing the sample complexity trade-off. First define,

$$\rho_\pi(\lambda, \varepsilon) := \max_{z \in \mathcal{Z} \setminus \{z_*\}} \frac{\|z - z_*\|_{\mathbb{E}_{X \sim \lambda}[XX^\top]^{-1}}^2}{\max\{\langle \theta_*, z_* - z \rangle^2, \varepsilon^2\}} = \max_{h \in \mathcal{H} \setminus \{h_*\}} \frac{\mathbb{E}_{X \sim \pi}\left[\mathbf{1}\{h(X) \neq h'(X)\}\frac{\pi(X)}{\lambda(X)}\right]}{\max\{(R_\pi(h) - R_\pi(h^*))^2, \varepsilon^2\}}$$

An important case of the above setting is when $X \sim \nu$ and $\pi = \nu$, i.e. we are evaluating the performance of a classifier relative to the same distribution our samples are drawn from. This is the setting of [8, 15, 12]. The following theorem shows that the sample complexity obtained by our algorithm is at least as good as the results they present.

**Theorem 3.** *Fix any $\delta \in (0, 1)$, domain $\mathcal{X}$ with distribution $\nu$, finite hypothesis class $\mathcal{H}$, regression function $\eta : \mathcal{X} \to [0, 1]$. Set $\epsilon \geq 0$ and $\beta_\delta = 2048 \log(4 \log_2^2(4/\epsilon)|\mathcal{H}|/\delta)$. Then for $\tau \geq \rho_\pi(\nu, \epsilon)\beta_\delta$ there exists a selective sampling algorithm that returns $h \in \mathcal{H}$ satisfying $R_\pi(h) - R_\pi(h^*) \leq \epsilon$ by observing $\mathcal{U}$ unlabeled examples and requesting just $\mathcal{L}$ labels such that*

- $\mathcal{U} \leq \log_2(4/\epsilon)\tau$

- $\mathcal{L} \leq 3 \log_2(\frac{4}{\varepsilon}) \min_{\lambda \in \triangle_\mathcal{X}} \rho_\pi(\lambda, \varepsilon)\beta_\delta \quad s.t. \quad \tau \geq \|\lambda/\nu\|_\infty \rho_\pi(\lambda, \varepsilon)\beta_\delta$

*with probability at least $1 - \delta$. Furthermore when $\nu = \pi$ and if $\tau \geq 16\rho(\nu, \epsilon)\beta_\delta$ we have that*

$$\mathcal{L} \leq 36 \log_2(4/\epsilon)\left(\frac{R_\nu(h^*)^2}{\epsilon^2} + 4\right) \sup_{\xi \geq \epsilon} \theta^*(2R_\nu(h^*) + \xi, \nu)\beta_\delta$$

*where $\theta^*(u, \nu)$ is the disagreement coefficient, defined in Appendix E.*

Note that if $\tau$ is sufficiently large then the labeled sample complexity we obtain $\min_{\lambda \in \Delta_\mathcal{X}} \rho(\lambda, \epsilon)$ could be significantly smaller than previous results in the streaming setting, e.g. see [16]. The proof of Theorem 3 can be found in Appendix E.

## 4 Solving the Optimization Problem

Recall that in Algorithm 1, during round $\ell$, we need to solve optimization problem (3). Solving this optimization problem is not trivial because the number of variables can potentially be infinite if $\mathcal{X}$ is

an infinite set. In this section, we will demonstrate how to reduce it to a finite-dimensional problem by considering its dual problem. To simplify the notation, let $\mathcal{Y}_\ell = \{z - z' : z, z' \in \mathcal{Z}_\ell, z \neq z'\}$, and rewrite the problem as follows, where $c_\ell > 0$ is a constant that may depend on round $\ell$.

$$\begin{aligned}
\min_P \quad & \mathbb{E}_{X \sim \nu}\left[P(X)\right] \\
\text{subject to} \quad & y^\top \mathbb{E}_{X \sim \nu}\left[P(X)XX^\top\right]^{-1} y \leq c_\ell^2, \quad \forall y \in \mathcal{Y}_\ell, \\
& 0 \leq P(x) \leq 1, \quad \forall x \in \mathcal{X}.
\end{aligned} \tag{4}$$

Using the Schur complement technique, we show in Lemma 13 (Appendix C) the following equivalence: $y^\top \mathbb{E}_{X \sim \nu}\left[P(X)XX^\top\right]^{-1} y \leq c_\ell^2 \iff \mathbb{E}_{X \sim \nu}\left[P(X)XX^\top\right] \succeq \frac{1}{c_\ell^2} yy^\top$. This transforms a constraint involving matrix inversion into one with ordering between PSD matrices. Then, we remove the bound constraints $0 \leq P(x) \leq 1, \forall x \in \mathcal{X}$ by introducing the barrier function $-\log(1 - x) - \log(x)$. That is, instead of working with the objective $\mathbb{E}_{X \sim \nu}\left[P(X)\right]$ directly, we consider the following problem.

$$\begin{aligned}
\min_P \quad & \mathbb{E}_{X \sim \nu}[P(X) - \mu_b(\log(1 - P(X)) + \log(P(X)))] \\
\text{subject to} \quad & \mathbb{E}_{X \sim \nu}\left[P(X)XX^\top\right] \succeq \frac{1}{c_\ell^2} yy^\top, \quad \forall y \in \mathcal{Y}_\ell.
\end{aligned} \tag{5}$$

Here, $\mu_b \in (0, 1)$ is some small constant that controls how strong the barrier is. Intuitively, a smaller $\mu_b$ will make problem (5) closer to the original problem. We now show that unlike the primal, the dual problem is indeed finite-dimensional. For each constraint of $y \in \mathcal{Y}_\ell$, let the matrix $\Lambda_y \succeq \mathbf{0}$ be its dual variable. Further, let $\Lambda = \sum_{y \in \mathcal{Y}_\ell} \Lambda_y$ and $\mathbf{\Lambda} = (\Lambda_y)_{y \in \mathcal{Y}_\ell}$. The corresponding Lagrangian is

$$\mathcal{L}(\mathbf{\Lambda}, P) = \mathbb{E}_{X \sim \nu}\left[P(X) - \mu_b\left(\log(1 - P(X)) + \log(P(X))\right) - P(X)X^\top\Lambda X\right] + \frac{1}{c_\ell^2}\sum_{y \in \mathcal{Y}_\ell} y^\top \Lambda_y y.$$

The dual problem is $\max_{\Lambda_y \succeq \mathbf{0}, \forall y \in \mathcal{Y}_\ell} \min_P \mathcal{L}(\mathbf{\Lambda}, P)$. Notice that minimization over $P : \mathcal{X} \mapsto [0, 1]$ can be done via minimizing $P(x)$ point-wise for each $x \in \mathcal{X}$. To do this, we take the gradient with respect to each $P(x)$ and set it to zero to get

$$1 + \frac{\mu_b}{1 - P(x)} - \frac{\mu_b}{P(x)} - x^\top \Lambda x = 0. \tag{6}$$

Solving this equation and defining $q_\Lambda(x) = x^\top \Lambda x - 1$, we get

$$P_\Lambda(x) = \frac{1}{2} - \frac{\mu_b}{q_\Lambda(x)} + \frac{\sqrt{(2\mu_b - q_\Lambda(x))^2 + 4\mu_b q_\Lambda(x)}}{2q_\Lambda(x)}. \tag{7}$$

Note that if $\mu_b = 0$ (no barrier), the above reduces to the "threshold" decision rule $P_\Lambda(x) = \frac{1}{2} + \frac{|q_\Lambda(x)|}{2q_\Lambda(x)}$, which gives 0 when $q_\Lambda(x) < 0$ and 1 when $q_\Lambda(x) > 0$.[2] This is exactly the hard elliptical threshold rule mentioned before, in which whether to query the label for $x$ depends on whether it falls inside ($x^\top \Lambda x < 1$) or outside ($x^\top \Lambda x > 1$) of the ellipsoid defined by the positive semidefinite matrix $\Lambda$. A visualization of the decision rule $P_\Lambda$ is given in Figure 2 in the Appendix.

Now, by plugging in $P_\Lambda(x)$, our dual problem becomes $\max_{\Lambda_y \succeq \mathbf{0}, \forall y} D(\mathbf{\Lambda}) := \mathcal{L}(\mathbf{\Lambda}, P_\Lambda)$. This is a finite-dimensional optimization problem, and can be solved by projected gradient ascent (or projected stochastic gradient ascent when we have only samples from $\nu$). The gradient of $D(\mathbf{\Lambda})$ is

$$\begin{aligned}
\nabla_{\Lambda_y} D(\mathbf{\Lambda}) &= \mathbb{E}_{X \sim \nu}\left[\left(1 + \frac{\mu_b}{1 - P_\Lambda(x)} - \frac{\mu_b}{P_\Lambda(X)} - X^\top \Lambda X\right)\nabla_{\Lambda_y} P_\Lambda(X) - P_\Lambda(X)XX^\top\right] + \frac{yy^\top}{c_\ell^2} \\
&= \frac{yy^\top}{c_\ell^2} - \mathbb{E}_{X \sim \nu}\left[P_\Lambda(X)XX^\top\right]. \qquad\qquad \text{(Since } P_\Lambda(X) \text{ solves Eq. (6))}
\end{aligned}$$

The algorithm to solve the problem has been summarized in Algorithm 2, in which the gradient during $k$th iteration is replaced by its unbiased estimator $\frac{yy^\top}{c_\ell^2} - P_{\hat{\Lambda}^{(k)}}(x_k)x_k x_k^\top$. The adaptive learning rate is chosen by following the discussion in chapter 4 of [21]. Optimizing the assignment of $\hat{\Lambda}_y$ to each y in line 10 ensures that the re-scaling step in line 11 increases the function value in an optimized way. Finally, the re-scaling step is used to ensure that the output primal objective value $\mathbb{E}_{X \sim \nu}\left[P(X)\right]$ is bounded well, which will be explained in more details in Appendix C.

---

[2]When $q_\Lambda(x) = 0$, $P_\Lambda(x)$ is undetermined from the dual.

---

**Algorithm 2** Projected Stochastic Gradient Ascent to Solve OPTIMIZEDESIGN

---

1: **Input:** Number of iterations $K$; number of samples $u$; barrier weight $\mu_b \in (0,1)$
2: Initialize $\hat{\Lambda}_y^{(0)} = \mathbf{0}$ for each $y \in \mathcal{Y}_\ell$
3: **for** $k = 0,1,2,\ldots,K-1$ **do**
4:     Sample $x_k \sim \nu$
5:     Set $g_{k,y} = \frac{yy^\top}{c_\ell^2} - P_{\hat{\Lambda}^{(k)}}(x_k)x_k x_k^\top$, where $P_\Lambda$ is defined in Eq. (7)
6:     Set $\hat{\Lambda}_y^{(k+1)} \leftarrow \hat{\Lambda}_y^{(k)} + \eta_k g_{k,y}$ for each $y \in \mathcal{Y}_\ell$, where $\eta_k = \frac{1}{\sqrt{2\sum_{s=1}^k \sum_{y \in \mathcal{Y}_\ell} \|g_{s,y}\|_2^2}}$
7:     Update $\hat{\Lambda}_y^{(k+1)} \leftarrow \Pi_{\mathbb{S}_+^d}(\hat{\Lambda}_y^{(k+1)})$ for each $y \in \mathcal{Y}_\ell$, a projection to the set of $d \times d$ PSD matrices
8: **end for**
9: Let $\hat{\Lambda}_y = \frac{1}{K}\sum_{k=1}^K \hat{\Lambda}_y^{(k)}$ for each $y \in \mathcal{Y}_\ell$ and $\hat{\Lambda} = \sum_{y \in \mathcal{Y}_\ell} \hat{\Lambda}_y$
10: Update $(\hat{\Lambda}_y)_{y \in \mathcal{Y}_\ell} \leftarrow \text{argmax}_\Lambda \sum_{y \in \mathcal{Y}_\ell} y^\top \Lambda_y y$, subject to $\sum_{y \in \mathcal{Y}_\ell} \Lambda_y = \hat{\Lambda}, \Lambda_y \succeq \mathbf{0}, \forall y \in \mathcal{Y}_\ell$.
11: Find $s^* \leftarrow \text{argmax}_{s \in [0,1]} D_E(s \cdot \hat{\Lambda})$, where $D_E$ empirically evaluates $D$ using $u$ i.i.d. samples
12: **return** $\widetilde{\Lambda} = s^* \cdot \sum_{y \in \mathcal{Y}_\ell} \hat{\Lambda}_y$

---

Let $\Lambda^*$ be an optimal solution for $D(\Lambda)$. Intuitively, as long as we run this algorithm with sufficiently large number of iterations $K$ and number of samples $u$, we can guarantee that $D(\widetilde{\Lambda})$ and $D(\Lambda^*)$ are close enough with high probability, which in turn guarantees that the primal constraints are violated by only a tiny amount and $\mathbb{E}_{X \sim \nu}[P_{\widetilde{\Lambda}}(X)]$ is close enough to the optimal value. Specifically, we can prove the following theorem.

**Theorem 4.** *Suppose* $\|x\|_2 \leq M$ *for any* $x \in \text{supp}(\nu)$ *and* $\Sigma = \mathbb{E}_{X \sim \nu}[XX^\top]$ *is invertible. Let* $\Lambda^* \in \text{argmax}_{\Lambda_y \succeq \mathbf{0}, \forall y \in \mathcal{Y}_\ell} D(\Lambda)$ *and* $\kappa(\Sigma) = \frac{\lambda_{\max}(\Sigma)}{\lambda_{\min}(\Sigma)}$ *be its condition number. Assume* $\|\Lambda^*\|_F > 0$ *and define* $\omega = \min_{\Gamma \in \mathbb{S}^d : \|\Gamma\|_F = 1} \mathbb{E}_{X \sim \nu}\left[(X^\top \Gamma X)^2\right]$, *where* $\mathbb{S}^d$ *is the set of* $d \times d$ *symmetric matrices.*

*Then,* $\Lambda^* = \sum_{y \in \mathcal{Y}_\ell} \Lambda_y^*$ *is unique. Further, for any* $\epsilon > 0$ *and* $\delta > 0$, *if it holds that* $\mu_b \leq O\left(\sqrt{\|\Lambda^*\|_F}\,\kappa(\Sigma)M\right) \cdot \sqrt{(1+\epsilon)/\epsilon}$ *and*

$$K \geq O\left(\frac{|\mathcal{Y}_\ell|^3 \kappa(\Sigma)^2 \|\Lambda^*\|_F^8 M^{16} \log(1/\delta)}{\omega^2 \mu_b^6}\right) \cdot \left(\frac{1+\epsilon}{\epsilon}\right)^2, u \geq O\left(\frac{\kappa(\Sigma)^2 \|\Lambda^*\|_F^6 M^{16} \log(1/\delta)}{\omega^2 \mu_b^6}\right) \cdot \left(\frac{1+\epsilon}{\epsilon}\right)^2,$$

*then, with probability at least* $1 - \delta$, *Algorithm 2 will output* $\widetilde{\Lambda}$ *that satisfies*

- $y^\top \mathbb{E}_{X \sim \nu}\left[P_{\widetilde{\Lambda}}(X)XX^\top\right]^{-1} y \leq (1+\epsilon)c_\ell^2, \quad \forall y \in \mathcal{Y}_\ell.$

- $\mathbb{E}_{X \sim \nu}\left[P_{\widetilde{\Lambda}}(X)\right] \leq \mathbb{E}_{X \sim \nu}\left[\widetilde{P}(X)\right] + 4\sqrt{\mu_b}$, *where* $\widetilde{P}$ *is the optimal solution to problem* (4) *with barrier constraint repaced by* $0 \leq P(x) \leq 1 - \mu_b, \forall x \in \mathcal{X}$.

The proof is in Appendix C. Although $\widetilde{P}$ is not exactly the same as the optimal solution of the original problem (4), when $\mu_b$ is sufficiently small, they will be very close. Meanwhile, it should be noted that Theorem 4 mainly reveals that with sufficiently large number of iterations and number of samples, Algorithm 2 can output sufficiently good solution. In future work, we plan to examine how much this bound can be improved via a tighter analysis.

Finally, notice that Algorithm 2 needs to maintain $|\mathcal{Y}_\ell|\,d^2 = O(|\mathcal{Z}_\ell|^2 d^2)$ variables, which can be large when we have a large set $\mathcal{Z}_\ell$. Therefore, as an alternative, we also propose Algorithm 3 that only needs to maintain $d^2$ variables but requires more computational power in each iteration. The details are given in Appendix C.

## 5 Empirical results

In this section we present a benchmark experiment validating the fundamental trade-offs that are theoretically characterized in Theorem 1 and Theorem 2. We take inspiration from [24] to define our experimental protocol:

- $d = 2$, a two-dimensional problem.
- $\mathcal{Z} = [\mathbf{e}_1, \mathbf{e}_2, (\cos(\omega), \sin(\omega))]$ for $\omega = 0.3$, where $\mathbf{e}_1, \mathbf{e}_2$ are canonical vectors.
- $\theta_* = 2\mathbf{e}_1$ and $y = x^\top \theta_* + \eta$, where $\eta \sim \mathcal{N}(0, 1)$.
- The distribution $\nu$ for streaming measurements $x_t \overset{i.i.d.}{\sim} \nu$ is such that $x_t = (\cos(2I_t\pi/N), \sin(2I_t\pi/N))$ where $I_t \in \{0, \ldots, N-1\}$, $\mathbb{P}(I_t = i) \propto \cos(2i\pi/N)^2$, and $N = 30$.

In this problem, the angle $\omega$ is small enough that the item $(\cos(\omega), \sin(\omega))$ is hard to discriminate from the best item $\mathbf{e}_1$. As argued in [24], an efficient sampling strategy for this problem instance would be to pull arms in the direction of $\pm\mathbf{e}_2$ in order to reduce the uncertainty in the direction of interest, $\mathbf{e}_1 - (\cos(\omega), \sin(\omega))$. However, the distribution $\nu$ is defined such that it is more likely to receive a vector $x_t$ in the direction of $\pm\mathbf{e}_1$ rather than $\pm\mathbf{e}_2$. Thus, if one seeks a small label complexity, then $P$ should be taken to reject measurements in the direction of $\pm e_1$.

In the benchmark experiment, we compare the following three algorithms which all use Algorithm 1 as a meta-algorithm and just swap out the definition of $\widehat{P}_\ell$. `Naive Algorithm` uses no selective sampling so that $\widehat{P}_\ell(x) = 1$ for all $x$; the `Oracle Algorithm` uses $\widehat{P}_\ell = P_*$ where $P_*$ is the ideal solution to (2), and `Our Algorithm` uses the solution to (5) for $\widehat{P}_\ell$, where we take $\mu_b = 2 \times 10^{-5}$. We swept over the values of $\tau$ and plotted on the y-axis the amount of labeled data needed before termination, as shown in Figure 1.

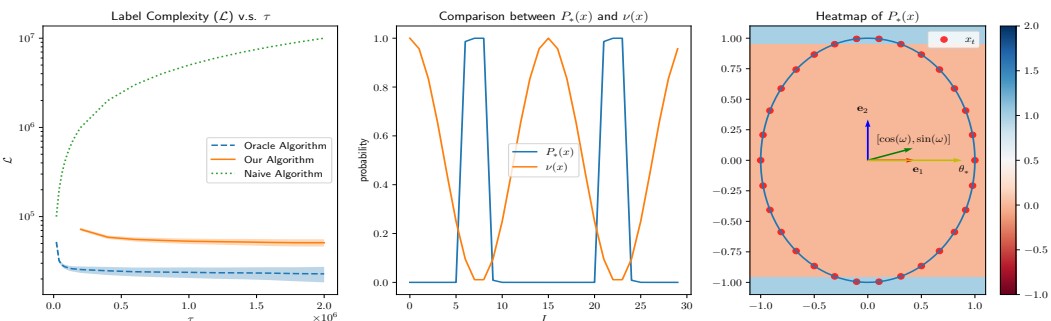

Figure 1: (left) For each value of $\tau$, we plot the average label complexity over 50 repeated trials. (middle) Visualization of $P_*(x)$ and $\nu(x)$ v.s. $x$, where $x$ is indexed by $I$ such that $x_I = (\cos(2I\pi/N), \sin(2I\pi/N))$. Here, $P_*$ is solved with $\tau = 4 \times 10^5$ and distribution $\nu$ is not normalized. (right) A heat map of $P_*(x)$ along with the setting of experimental protocol.

We observe in Figure 1 that the algorithms using non-naive selection rules require far less label complexity than the naive algorithm for all $\tau$. This reflects the intuition that selection strategies that focus on requesting the more informative streaming measurements are much more efficient than naively observing every streaming measurement. Meanwhile, the trade-off between label complexity $\mathcal{L}$ and sample complexity $\mathcal{U}$ characterized in Theorem 1 and Theorem 2 is precisely illustrated in Figure 1. Indeed, we see the number of labels queried by the two selective sampling algorithms decrease as the number of unlabeled data seen in each round increases.

## 6 Conclusion

In this paper, we proposed a new approach for the important problem of *selective sampling for best arm identification*. We provide a lower bound that quantifies the trade-off between labeled samples and stopping time and also presented an algorithm that nearly achieves the minimal label complexity given a desired stopping time.

One of the main limitations of this work is that our approach depends on a well-specified model following stationary stochastic assumptions. In practice, dependencies over time and model mismatch are common. Utilizing the proposed algorithm outside of our assumptions may lead to poor performance and unexpected behavior with adverse consequences. While negative results justify some of the most critical assumptions we make (e.g., allowing the stream $x_t$ to be arbitrary, rather than iid, can lead to trivial algorithms, see Theorem 7 of [7]), exploring what theoretical guarantees are possible under relaxed assumptions is an important topic of future work.

# Acknowledgements

We sincerely thank Chunlin Sun for the insightful discussion on the alternative approach to the optimal design. This work was supported in part by the NSF TRIPODS II grant DMS 2023166, NSF TRIPODS CCF 1740551, NSF CCF 2007036 and NSF TRIPODS+X DMS 1839371.

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
