# Contents

# A  Selective Sampling Lower Bound

First, we review the standard argument for best-arm identification lower bounds applied to linear bandits. Fix $\theta_* \in \mathbb{R}^d$ and let $z_* = \arg\max_{z \in \mathcal{Z}} \langle z, \theta_* \rangle$. Define the set $\mathcal{C} = \{\theta \in \mathbb{R}^d : \exists z \in \mathcal{Z} \text{ s.t. } \langle \theta, z - z_* \rangle \geq 0\}$ as those $\theta$ in which $z_*$ is note the best arm under $\theta$. We now recall the transportation lemma of [17]. Under a $\delta$-PAC strategy for finding the best arm for the bandit instance $(\mathcal{X}, \mathcal{Z}, \theta_*)$, let $T_x$ denote the random variable which is the number of times arm $x$ is pulled. In addition let $\mathcal{N}_{\theta,x}$ denote the reward distribution of the arm $x$ of $\mathcal{X}$, i.e. $\mathcal{N}_{\theta,x} = \mathcal{N}(x^\top \theta, 1)$. Then for any $\delta$-PAC algorithm

$$
\begin{aligned}
\log(1/2.4\delta) &\leq \min_{\theta \in \mathcal{C}} \sum_{x \in \mathcal{X}} \mathbb{E}[T_x] \mathrm{KL}(\mathcal{N}_{\theta_*,x}, \mathcal{N}_{\theta,x}) \\
&= \min_{\theta \in \mathcal{C}} \sum_{x \in \mathcal{X}} \mathbb{E}[T_x] \tfrac{1}{2} \|\theta_* - \theta\|^2_{xx^\top} \\
&= \min_{\theta \in \mathcal{C}} \tfrac{1}{2} \|\theta_* - \theta\|^2_{(\sum_{x \in \mathcal{X}} \mathbb{E}[T_x]\, xx^\top)} \\
&\leq \min_{z \in \mathcal{Z} \setminus z_*} \tfrac{1}{2} \|\theta_* - \theta_z(\epsilon)\|^2_{(\sum_{x \in \mathcal{X}} \mathbb{E}[T_x]\, xx^\top)}
\end{aligned}
$$

where

$$
\theta_z(\varepsilon) = \theta_* - \frac{((z_* - z)^\top \theta_* + \varepsilon)(\sum_{x \in \mathcal{X}} \mathbb{E}[T_x]\, xx^\top)^{-1}(z_* - z)^\top}{(z_* - z)^\top (\sum_{x \in \mathcal{X}} \mathbb{E}[T_x]\, xx^\top)^{-1}(z_* - z)}
$$

for some small $\epsilon$. This is a valid choice since for all $z \in \mathcal{Z} \setminus z_*$ we have $(z_* - z)^\top \theta_z(\varepsilon) = -\varepsilon < 0$ and thus $\theta_z(\varepsilon) \in \mathcal{C}$. A straightforward calculation shows that

$$
\|\theta_* - \theta_z(\epsilon)\|^2_{(\sum_{x \in \mathcal{X}} \mathbb{E}[T_x]\, xx^\top)} = \frac{(\langle z_* - z, \theta_* \rangle + \varepsilon)^2}{\|z_* - z\|^2_{(\sum_{x \in \mathcal{X}} \mathbb{E}[T_x]\, xx^\top)^{-1}}}
$$

so that after rearranging and lettering $\epsilon \to 0$ we have that any $\delta$-PAC algorithm satisfies

$$
\max_{z \in \mathcal{Z} \setminus z_*} \frac{2\|z_* - z\|^2_{(\sum_{x \in \mathcal{X}} \mathbb{E}[T_x]\, xx^\top)^{-1}}}{\langle z_* - z, \theta_* \rangle^2} \log(1/2.4\delta) \leq 1. \tag{8}
$$

This series of steps will be applied for each bullet point of the theorem.

## A.1  Proof of Theorem 1, part I

We use the consequence of Lemma 19 of [17]. Consider a $\delta$-PAC algorithm that sets $P(x) = 1$ for all $x \in \mathcal{X}$ for all time until it exits at time $\mathcal{U}$ after this many unlabelled examples have been observed. If $T_x$ denotes the number of times $x \in \mathcal{X}$ was observed before stopping time $\mathcal{U}$, then by Wald's identity we have that

$$
\mathbb{E}[T_x] = \mathbb{E}\left[\sum_{t=1}^{\mathcal{U}} \mathbf{1}\{x_t = x\}\right] = \nu(x) \mathbb{E}[\mathcal{U}].
$$

Plugging this into Equation 8 and rearranging we conclude that

$$
\mathbb{E}[\mathcal{U}] \geq \max_{z \in \mathcal{Z} \setminus z_*} \frac{2\|z_* - z\|^2_{(\sum_{x \in \mathcal{X}} \nu(x)\, xx^\top)^{-1}}}{\langle z_* - z, \theta_* \rangle^2} \log(1/2.4\delta) =: \rho(\nu) \log(1/2.4\delta)
$$

which concludes the proof of the first bullet.

## A.2  Proof of Theorem 1, part II

By definition, the (random) number of times we measure $x$ is

$$
\mathcal{L}_x = \sum_{s=1}^{\mathcal{U}} \mathbf{1}\{x_s = x, Q_s(x) = 1\}
$$

and we want to show that $\mathbb{E}[\mathcal{L}_x] = \nu(x)\mathbb{E}\left[\sum_{\ell=1}^{\mathcal{U}} P_\ell(x)\right]$. To do so, we define

$$M_t = \sum_{s=1}^{t} \left(\mathbf{1}\{x_s = x, Q_s(x) = 1\} - \nu(x)P_s(x)\right)$$

It is easy to check that $P_{t+1} \in \mathcal{F}_t := \{(x_s, y_s, Q_s)\}_{s=1}^{t}$ and that

$$\mathbb{E}[M_{t+1}|\mathcal{F}_t] = M_t + \mathbb{E}[\mathbf{1}\{x_s = x, Q_s(x) = 1\} - \nu(x)P_s(x)|\mathcal{F}_t] = M_t$$

Applying Doob's equality $\mathbb{E}[M_\mathcal{U}] = \mathbb{E}[M_0] = 0$. Consequence:

$$\mathbb{E}[\mathcal{L}_x] = \mathbb{E}\left[\sum_{s=1}^{\mathcal{U}} \mathbf{1}\{x_s = x, Q_s(x) = 1\}\right] = \nu(v)\mathbb{E}\left[\sum_{s=1}^{\mathcal{U}} P_s(x)\right]$$

Define $\alpha(x) := \frac{\mathbb{E}[\sum_{s=1}^{\mathcal{U}} P_s(x)]}{\mathbb{E}[\mathcal{U}]}$ and note that each $\alpha_x \in [0, 1]$. Then $\mathbb{E}[\mathcal{L}_x] = \mathbb{E}[\mathcal{U}]\alpha(x)\nu(x)$ so applying equation (18) of [17] again, we have

$$\log(1/2.4\delta) \leq \min_{\theta \in \mathcal{C}} \sum_{x \in \mathcal{X}} \mathbb{E}[\mathcal{L}_x]\mathrm{KL}(\mathcal{N}_{\theta_*,x}, \mathcal{N}_{\theta,x})$$

$$= \min_{\theta \in \mathcal{C}} \sum_{x \in \mathcal{X}} \mathbb{E}[\mathcal{L}_x] \|\theta - \theta_*\|_{xx^\top}^2/2$$

$$= \min_{z \in \mathcal{Z} \setminus z_*} \frac{\langle \theta_*, z_* - z \rangle^2}{2\|z - z_*\|_{(\sum_{x \in \mathcal{X}} \mathbb{E}[\mathcal{L}_x]xx^\top)^{-1}}^2}$$

$$= \min_{z \in \mathcal{Z} \setminus z_*} \frac{\langle \theta_*, z_* - z \rangle^2}{2\|z - z_*\|_{(\sum_{x \in \mathcal{X}} \nu(x)\alpha(x)xx^\top)^{-1}}^2} \mathbb{E}[\mathcal{U}].$$

Rearranging, and applying the identity $\mathbb{E}_{X \sim \nu}[\alpha(X)XX^\top] = \sum_{x \in \mathcal{X}} \nu(x)\alpha(x)xx^\top$, the above implies that

$$\mathbb{E}[\mathcal{U}] \geq \max_{z \in \mathcal{Z} \setminus z_*} \frac{2\|z - z_*\|_{\mathbb{E}_{X \sim \nu}[\alpha(X)XX^\top]^{-1}}^2}{\langle \theta_*, z_* - z \rangle^2} \log(1/2.4\delta).$$

Noting that the total expected number of labels is equal to

$$\mathbb{E}[\mathcal{L}] = \sum_{x \in \mathcal{X}} \mathbb{E}[\mathcal{L}_x] = \sum_{x \in \mathcal{X}} \mathbb{E}[\mathcal{U}]\alpha(x)\nu(x) = \mathbb{E}[\mathcal{U}]\,\mathbb{E}_{X \sim \nu}[\alpha(X)]$$

we conclude that

$$\mathbb{E}[\mathcal{L}] \geq \min_{\alpha:\mathcal{X} \to [0,1]} \quad \mathbb{E}[\mathcal{U}]\,\mathbb{E}_{X \sim \nu}[\alpha(X)]$$

$$\text{subject to} \quad \mathbb{E}[\mathcal{U}] \geq \max_{z \in \mathcal{Z} \setminus \{z_*\}} \frac{2\|z - z_*\|_{\mathbb{E}_{X \sim \nu}[\alpha(X)XX^\top]^{-1}}^2}{\langle \theta_*, z_* - z \rangle^2} \log(1/2.4\delta).$$

The second bullet point result follows by denoting $\alpha$ as $P$ and applying Proposition 2.

# B  Selective Sampling Algorithm for Known Distribution $\nu$

## B.1  Proof of Theorem 2, upper bound

At each round $\ell$ we assume an implementation such that $\widehat{P}_\ell, \widehat{\Sigma}_{\widehat{P}_\ell} \leftarrow \textsc{OptimizeDesign}(\mathcal{Z}_\ell, 2^{-\ell}, \tau)$ returns the solution of Equation 3 with $\epsilon = 2^{-\ell}$, essentially. More explicitly, let $\epsilon_\ell := 2^{-\ell}$, $B < \infty$ such that $\max_{x \in \mathcal{X}} |\langle x, \theta_* \rangle| \leq B$, and $\sigma < \infty$ such that $\mathbb{E}[(y_s - \langle \theta_*, x_s \rangle)^2|x_s] \leq \sigma^2$. If

$$\beta_{\delta,\ell} := 16(B^2 + \sigma^2)\log(2\ell^2|\mathcal{Z}|^2/\delta)$$

then $\widehat{P}_\ell = P_\ell$ where

$$P_\ell := \operatorname*{argmin}_{P:\mathcal{X} \to [0,1]} \mathbb{E}_{X \sim \nu}[P(X)] \text{ subject to } \max_{z,z' \in \mathcal{Z}_\ell} \frac{\|z - z'\|_{\mathbb{E}_{X \sim \nu}[\tau P(X)XX^\top]^{-1}}^2}{\epsilon_\ell^2}\beta_{\delta,\ell} \leq 1$$

and $\widehat{\Sigma}_{\widehat{P}_\ell} := \mathbb{E}_{X \sim \nu}[P_\ell(X)XX^\top]$

We first provide an intermediate lemma on the correctness of Algorithm 1 that relies on the feasibility of $P_\ell$ which we will show shortly.

**Lemma 1.** *With probability at least $1 - \delta$ we have for all stages $\ell \in \mathbb{N}$ such that $P_\ell$ is feasible, that $z_* \in \mathcal{Z}_\ell$ and $\max_{z \in \mathcal{Z}_\ell} \langle z_* - z, \theta_* \rangle \le 4\epsilon_\ell$.*

*Proof.* Define the event $\mathcal{E}$ as

$$\mathcal{E} := \bigcap_{\ell=1}^{\infty} \bigcap_{z,z' \in \mathcal{Z}_\ell} \left\{ |\langle z - z', \widehat{\theta}_\ell - \theta_* \rangle| \le \epsilon_\ell \right\}$$

By Lemma 2, we know that $\mathbb{P}(\mathcal{E}) \ge 1 - \delta$. Then, the rest of the proof is the same as the one in [11], but we include it here for completeness. Assume that $\mathcal{E}$ holds. Then for any $z' \in \mathcal{Z}_\ell$

$$\langle z' - z^*, \widehat{\theta}_\ell \rangle = \langle z' - z^*, \widehat{\theta}_\ell - \theta^* \rangle + \langle z' - z^*, \theta^* \rangle$$
$$= \langle z' - z^*, \widehat{\theta}_\ell - \theta^* \rangle$$
$$\le \epsilon_\ell$$

so that $z^*$ would survive to round $\mathcal{Z}_{\ell+1}$. And for any $z \in \mathcal{Z}_\ell$ such that $\langle z^* - z, \theta^* \rangle > 2\epsilon_\ell$, we have

$$\max_{z' \in \mathcal{Z}_\ell} \langle z' - z, \widehat{\theta}_\ell \rangle \ge \langle z^* - z, \widehat{\theta}_\ell \rangle$$
$$= \langle z^* - z, \widehat{\theta}_\ell - \theta^* \rangle + \langle z^* - z, \theta^* \rangle$$
$$> -\epsilon_\ell + 2\epsilon_\ell$$
$$= \epsilon_\ell$$

which implies this $z$ would be kicked out. Note that this implies that $\max_{z \in \mathcal{Z}_{\ell+1}} \langle z^* - z, \theta^* \rangle \le 2\epsilon_\ell = 4\epsilon_{\ell+1}$. $\square$

We can now prove Theorem 2. After $L := \lceil \log_2(\frac{4}{\Delta}) \rceil$ rounds $\mathcal{Z}_\ell = \{z_*\}$ by the above lemma. Thus, the total number of labels requested after $L$ rounds is equal to $\mathcal{L} := \sum_{\ell=1}^{L} \sum_{t=(\ell-1)\tau+1}^{\ell\tau} Q_\ell(x_t)$. By Freedman's inequality (c.f., Theorem 1 of [4]) we have that

$$\sum_{\ell=1}^{L} \sum_{t=(\ell-1)\tau+1}^{\ell\tau} Q_\ell(x_t) \le 2 \sum_{\ell=1}^{L} \tau \mathbb{E}_{X \sim \nu}[P_\ell(X)|\mathcal{Z}_\ell] + \log(1/\delta)$$

We can now bound the expected sample complexity of this algorithm.

$$\sum_{\ell=1}^{L} \tau \mathbb{E}_{X \sim \nu}[P_\ell(X)|\mathcal{Z}_\ell]$$
$$= \sum_{\ell=1}^{L} \left[ \min_{P:\mathcal{X} \to [0,1]} \tau \mathbb{E}_{X \sim \nu}[P(X)] \quad \text{subject to} \quad \max_{z,z' \in \mathcal{Z}_\ell} \frac{\|z - z'\|^2_{\mathbb{E}_{X \sim \nu}[\tau P(X)XX^\top]^{-1}}}{\epsilon_\ell^2} \beta_{\delta,\ell} \le 1 \right].$$

Using Lemma 3, we have

$$\max_{z,z' \in \mathcal{Z}_\ell} \frac{\|z - z'\|^2_{\mathbb{E}_{X \sim \nu}[\tau P(X)XX^\top]^{-1}}}{\epsilon_\ell^2} \beta_{\delta,\ell} \le \beta_{\delta,L} \max_{z,z' \in \mathcal{Z}_\ell} \frac{\|z - z'\|^2_{\mathbb{E}_{X \sim \nu}[\tau P(X)XX^\top]^{-1}}}{\epsilon_\ell^2}$$
$$\le 64\beta_{\delta,L} \max_{z \in \mathcal{Z} \setminus z_*} \frac{\|z - z_*\|^2_{\mathbb{E}_{X \sim \nu}[\tau P(X)XX^\top]^{-1}}}{\langle z - z_*, \theta_* \rangle^2}$$
$$=: \max_{z \in \mathcal{Z} \setminus z_*} \frac{\|z - z_*\|^2_{\mathbb{E}_{X \sim \nu}[\tau P(X)XX^\top]^{-1}}}{\langle z - z_*, \theta_* \rangle^2} \beta_\delta$$

Note that the last line also describes a condition for which a $P_\ell$ is feasible. Indeed, at round $\ell$, a sufficient condition for a feasible $P_\ell$ (i.e., the RHS $\le 1$) is if $\tau$ exceeds $\rho(\nu)\beta_\delta$ with $\beta_\delta := 1024(B^2 +$

$\sigma^2) \log(2L^2|\mathcal{Z}|^2/\delta)$ and $\rho(\nu) = \max_{z \in \mathcal{Z} \setminus z_*} \frac{\|z - z_*\|^2_{\mathbb{E}_{X \sim \nu}[XX^\top]^{-1}}}{\langle z - z_*, \theta_* \rangle^2}$, which holds by assumption in the theorem.

Plugging this constraint back into above we have

$$\sum_{\ell=1}^{L} \tau \mathbb{E}_{X \sim \nu}[P_\ell(X)|\mathcal{Z}_\ell]$$

$$\leq \sum_{\ell=1}^{L} \left[ \min_{P: \mathcal{X} \to [0,1]} \tau \mathbb{E}_{X \sim \nu}[P(X)] \quad \text{subject to} \quad \max_{z \in \mathcal{Z} \setminus z_*} \frac{\|z - z_*\|^2_{\mathbb{E}_{X \sim \nu}[\tau P(X)XX^\top]^{-1}}}{\langle z - z_*, \theta_* \rangle^2} \beta_\delta \leq 1 \right]$$

$$\leq L \min_{\lambda \in \triangle_{\mathcal{X}}} \rho(\lambda)\beta_\delta \quad \text{subject to} \quad \|\lambda/\nu\|_\infty \rho(\lambda)\beta_\delta \leq \tau$$

where the last line follows by applying the reparameterization of Proposition 2.

### B.1.1 High-probability Events

**Lemma 2.** *We have* $\mathbb{P}(\mathcal{E}) \geq 1 - \delta$.

*Proof.* For any $\mathcal{V} \subseteq \mathcal{Z}$ and $z, z' \in \mathcal{V}$ define

$$\mathcal{E}_{z,z',\ell}(\mathcal{V}) = \{|\langle z - z', \widehat{\theta}_\ell(\mathcal{V}) - \theta_* \rangle| \leq \epsilon_\ell\}$$

where $\widehat{\theta}_\ell(\mathcal{V})$ is the estimator that would be constructed by the algorithm at stage $\ell$ with $\mathcal{Z}_\ell = \mathcal{V}$. For fixed $\mathcal{V} \subset \mathcal{Z}$ and $\ell \in \mathbb{N}$ we apply Proposition 1 so that with probability at least $1 - \frac{\delta}{\ell^2|\mathcal{Z}|^2}$ we have that for any $z, z' \in \mathcal{V}$

$$|\langle z - z', \widehat{\theta}_\ell(\mathcal{V}) - \theta_* \rangle| \leq \|z - z'\|_{\mathbb{E}_{X \sim \nu}[\tau P_\ell(X)XX^\top]^{-1}} \sqrt{16(B^2 + \sigma^2) \log(2\ell^2|\mathcal{Z}|^2/\delta)}$$

$$\leq \epsilon_\ell$$

Noting that $\mathcal{E} := \bigcap_{\ell=1}^{\infty} \bigcap_{z,z' \in \mathcal{Z}_\ell} \mathcal{E}_{z,z',\ell}(\mathcal{Z}_\ell)$ we have

$$\mathbb{P}\left( \bigcup_{\ell=1}^{\infty} \bigcup_{z,z' \in \mathcal{Z}_\ell} \{\mathcal{E}^c_{z,z',\ell}(\mathcal{Z}_\ell)\} \right) \leq \sum_{\ell=1}^{\infty} \mathbb{P}\left( \bigcup_{z,z' \in \mathcal{Z}_\ell} \{\mathcal{E}^c_{z,z',\ell}(\mathcal{Z}_\ell)\} \right)$$

$$= \sum_{\ell=1}^{\infty} \sum_{\mathcal{V} \subseteq \mathcal{Z}} \mathbb{P}\left( \bigcup_{z,z' \in \mathcal{V}} \{\mathcal{E}^c_{z,z',\ell}(\mathcal{V})\}, \mathcal{Z}_\ell = \mathcal{V} \right)$$

$$= \sum_{\ell=1}^{\infty} \sum_{\mathcal{V} \subseteq \mathcal{Z}} \mathbb{P}\left( \bigcup_{z,z' \in \mathcal{V}} \{\mathcal{E}^c_{z,z',\ell}(\mathcal{V})\} \right) \mathbb{P}(\mathcal{Z}_\ell = \mathcal{V})$$

$$\leq \sum_{\ell=1}^{\infty} \sum_{\mathcal{V} \subseteq \mathcal{Z}} \frac{\delta}{\ell^2|\mathcal{Z}|^2} \binom{|\mathcal{V}|}{2} \mathbb{P}(\mathcal{Z}_\ell = \mathcal{V})$$

$$\leq \sum_{\ell=1}^{\infty} \sum_{\mathcal{V} \subseteq \mathcal{Z}} \frac{\delta}{2\ell^2} \mathbb{P}(\mathcal{Z}_\ell = \mathcal{V}) \leq \delta$$

$\square$

## B.2 Technical Lemmas

The following definition characterizes the RIPS estimator we used in Algorithm 1.

**Definition 2.** *Let* $X_1, \ldots, X_n$ *be i.i.d. random variables with mean* $\bar{x}$ *and variance* $\nu^2$. *Let* $\delta \in (0, 1)$. *We say that* $\widehat{\mu}(X_1, \ldots, X_n)$ *is a* $\delta$-*robust estimator if there exist universal constants* $c_1, c_0 > 0$ *such that if* $n \geq c_1 \log(1/\delta)$, *then with probability at least* $1 - \delta$

$$|\widehat{\mu}(\{X_t\}_{t=1}^{n}) - \bar{x}| \leq c_0 \sqrt{\frac{\nu^2 \log(1/\delta)}{n}}.$$

Examples of $\delta$-robust estimators include the median-of-means estimator and Catoni's estimator [18]. This work employs the use of the Catoni estimator which satisfies $|\widehat{\mu}(\{X_t\}_{t=1}^n) - \bar{x}| \leq \sqrt{\frac{2\nu^2 \log(1/\delta)}{n - 2\log(1/\delta)}}$ for $n > 2\log(1/\delta)$ which leads to an optimal leading constant as $n \to \infty$. See [5] or [18] for more details.

**Proposition 1.** *Let $x_1, \ldots, x_n$ be drawn IID from a distribution $\nu$. Assume that $|\langle \theta, x_s \rangle| \leq B$ and $\mathbb{E}[|\langle \theta, x_s \rangle - y_s|^2] \leq \sigma^2$. Let $P : \mathcal{X} \to [0, 1]$ be arbitrary. Let $Q(x_s) \sim Bernoulli(P(x_s))$ independently for all $s \in [n]$. For a given finite set $\mathcal{V} \subset \mathbb{R}^d$ define for any $v \in \mathcal{V}$*

$$w_v = \text{Catoni}(\{\langle v, \mathbb{E}_{X \sim \nu}[P(X)XX^\top]^{-1}Q(x_s)x_s y_s \rangle\}_{s=1}^n).$$

*If $\widehat{\theta} = \arg\min_\theta \max_v \frac{|w_v - \langle \theta, v \rangle|}{\|v\|_{\mathbb{E}_{X \sim \nu}[P(X)XX^\top]^{-1}}}$ and $n \geq 4\log(2|\mathcal{V}|/\delta)$, then with probability at least $1 - \delta$, it holds that*

$$|\langle v, \widehat{\theta} - \theta \rangle| \leq \|v\|_{\mathbb{E}_{X \sim \nu}[nP(X)XX^\top]^{-1}} \sqrt{16(B^2 + \sigma^2)\log(2|\mathcal{V}|/\delta)}$$

*Proof.* Inspired by [5], we note that

$$\max_{v \in \mathcal{V}} \frac{|\langle \widehat{\theta}, v \rangle - \langle \theta, v \rangle|}{\|v\|_{\mathbb{E}_{X \sim \nu}[nP(X)XX^\top]^{-1}}} = \max_{v \in \mathcal{V}} \frac{|\langle \widehat{\theta}, v \rangle - w_v + w_v - \langle \theta, v \rangle|}{\|v\|_{\mathbb{E}_{X \sim \nu}[nP(X)XX^\top]^{-1}}}$$

$$\leq \max_{v \in \mathcal{V}} \frac{|\langle \widehat{\theta}, v \rangle - w_v|}{\|v\|_{\mathbb{E}_{X \sim \nu}[nP(X)XX^\top]^{-1}}} + \max_{v \in \mathcal{V}} \frac{|w_v - \langle \theta, v \rangle|}{\|v\|_{\mathbb{E}_{X \sim \nu}[nP(X)XX^\top]^{-1}}}$$

$$= \min_\theta \max_{v \in \mathcal{V}} \frac{|\langle \theta, v \rangle - w_v|}{\|v\|_{\mathbb{E}_{X \sim \nu}[nP(X)XX^\top]^{-1}}} + \max_{v \in \mathcal{V}} \frac{|w_v - \langle \theta, v \rangle|}{\|v\|_{\mathbb{E}_{X \sim \nu}[nP(X)XX^\top]^{-1}}}$$

$$\leq 2 \max_{v \in \mathcal{V}} \frac{|\langle \theta, v \rangle - w_v|}{\|v\|_{\mathbb{E}_{X \sim \nu}[nP(X)XX^\top]^{-1}}}$$

So it suffices to show that each $|\langle \theta, v \rangle - w_v|$ is small. We begin by fixing some $v \in \mathcal{V}$ and bounding the variance of $v^\top \mathbb{E}_{X \sim \nu}[P(X)XX^\top]^{-1}Q(x_s)x_s y_s$ for any $s \leq n$ which is necessary to use the robust estimator. For readability purposes, we shorten $\mathbb{E}_{x_s \sim \nu, Q(x_s) \sim P(x_s)}$ as $\mathbb{E}_{x_s, Q}$ in the rest of this proof. Note that

$$\mathbb{V}\text{ar}_{x_s \sim \nu, Q(x_s) \sim P(x_s)}(v^\top \mathbb{E}_{X \sim \nu}[P(X)XX^\top]^{-1}Q(x_s)x_s y_s)$$

$$= \mathbb{E}_{x_s, Q}[(v^\top \mathbb{E}_{X \sim \nu}[P(X)XX^\top]^{-1}Q(x_s)x_s y_s)^2]$$

$$- \mathbb{E}_{x_s, Q}[v^\top \mathbb{E}_{X \sim \nu}[P(X)XX^\top]^{-1}Q(x_s)x_s y_s]^2$$

which means we can drop the second term to bound the variance by

$$\mathbb{E}_{x_s, Q}[((v^\top \mathbb{E}_{X \sim \nu}[P(X)XX^\top]^{-1}Q(x_s)x_s y_s)^2]$$

$$= \mathbb{E}_{x_s, Q}[(v^\top \mathbb{E}_{X \sim \nu}[P(X)XX^\top]^{-1}Q(x_s)x_s(x_s^\top \theta + \xi_s))^2]$$

$$= \mathbb{E}_{x_s, Q}[(v^\top \mathbb{E}_{X \sim \nu}[P(X)XX^\top]^{-1}Q(x_s)x_s(x_s^\top \theta))^2]$$

$$+ \mathbb{E}_{x_s, Q}[(v^\top \mathbb{E}_{X \sim \nu}[P(X)XX^\top]^{-1}Q(x_s)x_s)^2 \xi_t^2]$$

$$\leq B^2 \mathbb{E}_{x_s, Q}[(v^\top \mathbb{E}_{X \sim \nu}[P(X)XX^\top]^{-1}Q(x_s)x_s)^2]$$

$$+ \sigma^2 \mathbb{E}_{x_s, Q}[(v^\top \mathbb{E}_{X \sim \nu}[P(X)XX^\top]^{-1}Q(x_s)x_s)^2]$$

$$= \mathbb{E}_{x_s \sim \nu}[(B^2 + \sigma^2)\mathbb{E}_{Q(x_s) \sim P(x_s)}[v^\top \mathbb{E}_{X \sim \nu}[P(X)XX^\top]^{-1}Q(x_s)x_s x_s^\top Q(x_s)\mathbb{E}_{X \sim \nu}[P(X)XX^\top]^{-1}v]]$$

$$\overset{(i)}{=} \mathbb{E}_{x_s \sim \nu}[(B^2 + \sigma^2)\mathbb{E}_{Q(x_s) \sim P(x_s)}[v^\top \mathbb{E}_{X \sim \nu}[P(X)XX^\top]^{-1}Q(x_s)x_s x_s^\top \mathbb{E}_{X \sim \nu}[P(X)XX^\top]^{-1}v]]$$

$$\leq \mathbb{E}_{x_s \sim \nu}[(B^2 + \sigma^2)v^\top \mathbb{E}_{X \sim \nu}[P(X)XX^\top]^{-1}P(x_s)x_s x_s^\top \mathbb{E}_{X \sim \nu}[P(X)XX^\top]^{-1}v],$$

where we used that $Q(x_s)^2 = Q(x_s)$ in equality (i) above. Thus, we have

$$\mathbb{V}\text{ar}(v^\top \mathbb{E}_{X \sim \nu}[P(X)XX^\top]^{-1}Q(x_s)x_s y_s)$$

$$\leq (B^2 + \sigma^2)v^\top (\mathbb{E}_{X \sim \nu}[P(X)XX^\top]^{-1}\mathbb{E}_{x_s \sim \nu}[P(x_s)x_s x_s^\top](\mathbb{E}_{X \sim \nu}[P(X)XX^\top]^{-1})v$$

$$= (B^2 + \sigma^2)\|v\|_{(\mathbb{E}_{X \sim \nu}[P(X)XX^\top]^{-1}}^2$$

By using the property of Catoni estimator stated in Definition 2, we have $c_0 = \sqrt{2}$ and

$$|\langle\theta_*, v\rangle - w_v|$$
$$=|\text{Catoni}(\{\langle v, \mathbb{E}_{X\sim\nu}[P(X)XX^\top]^{-1}Q(x_s)x_sy_s\rangle\}_{s=1}^n) - \mathbb{E}[\langle v, \mathbb{E}_{X\sim\nu}[P(X)XX^\top]^{-1}Q(x_s)x_sy_s\rangle]|$$
$$\leq\sqrt{2}\sqrt{(\mathbb{V}\text{ar}(\langle v, \mathbb{E}_{X\sim\nu}[P(X)XX^\top]^{-1}Q(x_s)x_sy_s\rangle))\frac{\log(\frac{2}{\delta})}{n/2}}$$

(with probability at least $1 - \delta$ if $n \geq 4\log(2/\delta)$)

$$\leq\|v\|_{(\mathbb{E}_{X\sim\nu}[P(X)XX^\top]^{-1}}\sqrt{\frac{4(B^2 + \sigma^2)\log(\frac{2}{\delta})}{n}}$$
$$=\|v\|_{\mathbb{E}_{X\sim\nu}[nP(X)XX^\top]^{-1}}\sqrt{4(B^2 + \sigma^2)\log(2/\delta)}.$$

Finally, the proof is complete by taking union bounding over all $v \in \mathcal{V}$. $\qquad\square$

**Lemma 3.** *Holds*

$$\max_{z,z'\in\mathcal{Z}_\ell}\frac{\|z - z'\|^2_{\mathbb{E}_{X\sim\nu}[\tau P(X)XX^\top]^{-1}}}{\epsilon_\ell^2} \leq 64\max_{z\in\mathcal{Z}\setminus z_*}\frac{\|z - z_*\|^2_{\mathbb{E}_{X\sim\nu}[\tau P(X)XX^\top]^{-1}}}{\langle z - z_*, \theta_*\rangle^2}$$

*Proof.* Let $\mathcal{S}_\ell = \{z \in \mathcal{Z} : \langle z_* - z, \theta_*\rangle \leq 4\epsilon_\ell\}$. We have

$$\max_{z,z'\in\mathcal{Z}_\ell}\frac{\|z - z'\|^2_{\mathbb{E}_{X\sim\nu}[\tau P(X)XX^\top]^{-1}}}{\epsilon_\ell^2} \leq \max_{z,z'\in\mathcal{S}_\ell}\frac{\|z - z'\|^2_{\mathbb{E}_{X\sim\nu}[\tau P(X)XX^\top]^{-1}}}{\epsilon_\ell^2}$$
$$= 16\max_{z,z'\in\mathcal{S}_\ell}\frac{\|z - z'\|^2_{\mathbb{E}_{X\sim\nu}[\tau P(X)XX^\top]^{-1}}}{(4\epsilon_\ell)^2}$$
$$\leq 64\max_{z\in\mathcal{S}_\ell}\frac{\|z - z_*\|^2_{\mathbb{E}_{X\sim\nu}[\tau P(X)XX^\top]^{-1}}}{(4\epsilon_\ell)^2}$$
$$= 64\max_{z\in\mathcal{S}_\ell\setminus z_*}\frac{\|z - z_*\|^2_{\mathbb{E}_{X\sim\nu}[\tau P(X)XX^\top]^{-1}}}{\max\{(4\epsilon_\ell)^2, \langle z - z_*, \theta_*\rangle^2\}}$$
$$\leq 64\max_{z\in\mathcal{Z}\setminus z_*}\frac{\|z - z_*\|^2_{\mathbb{E}_{X\sim\nu}[\tau P(X)XX^\top]^{-1}}}{\langle z - z_*, \theta_*\rangle^2}.$$

$\qquad\square$

### B.2.1 Reparameterization

**Proposition 2.** *Fix $\nu \in \triangle_\mathcal{X}$ and any $\lambda \in \triangle_\mathcal{X}$. Define $\|\lambda/\nu\|_\infty = \sup_{x\in\mathcal{X}} \lambda(x)/\nu(x)$ and $\rho(\lambda) = \max_{z\neq z_*}\frac{\|z-z_*\|^2_{\mathbb{E}_{X\sim\lambda}[XX^\top]^{-1}}}{\langle z_*-z,\theta_*\rangle^2}$. For any $t, \beta \in \mathbb{R}_+$ the following optimization problems achieve the same value*

- $\min\limits_{P:\mathcal{X}\to[0,1]} t\,\mathbb{E}_{X\sim\nu}[P(X)]$ *subject to* $\max_{z\neq z_*}\frac{\|z-z_*\|^2_{\mathbb{E}_{X\sim\nu}[P(X)XX^\top]^{-1}}}{\langle z_*-z,\theta_*\rangle^2}\beta \leq t$

- $\min\limits_{\lambda\in\triangle_\mathcal{X}}\rho(\lambda)\beta$ *subject to* $\|\lambda/\nu\|_\infty\rho(\lambda)\beta \leq t$

Let us first prove a simple lemma.

**Lemma 4.** *Let $\mathcal{P}$ denote the set of all functions $P : \mathcal{X} \to [0, 1]$. And for any $\nu \in \triangle_\mathcal{X}$ with support $\mathcal{X}$ let $\mathcal{P}' = \{\kappa\lambda_x/\nu_x : \lambda \in \triangle_\mathcal{X}, \kappa \geq 0 : \kappa\lambda_x/\nu_x \in [0, 1]\}$. Then $\mathcal{P} = \mathcal{P}'$.*

*Proof.* Fix any $P \in \mathcal{P}$. If $\lambda_x = P_x\nu_x/\|P \circ \nu\|_1$ and $\kappa = \|P \circ \nu\|_1$ then $\kappa\lambda/\nu \in \mathcal{P}'$ and is equal to $P$. This implies $\mathcal{P} \subseteq \mathcal{P}'$.

For the other direction, fix any $\lambda \in \triangle_\mathcal{X}$ and $\kappa \geq 0$ such that $\kappa\lambda_x/\nu_x \in [0, 1]$ for all $x$. If $P = \kappa\lambda/\nu$ then $P \in \mathcal{P}$ which implies $\mathcal{P}' \subseteq \mathcal{P}$ and concludes the proof. $\qquad\square$

*Proof of Proposition 2.* Using the above lemma we have that

$$\min_{P:\mathcal{X}\to[0,1]} t\,\mathbb{E}_{X\sim\nu}[P(X)] \quad \text{subject to} \quad \max_{z\neq z_*} \frac{\|z-z_*\|^2_{\mathbb{E}_{X\sim\nu}[P(X)XX^\top]^{-1}}}{\langle z_*-z,\theta_*\rangle^2}\beta \leq t$$

is equivalent to

$$\min_{\kappa\geq 0,\lambda\in\triangle_\mathcal{X}} t\,\mathbb{E}_{X\sim\nu}[\kappa\lambda(X)/\nu(X)] \quad \text{subject to} \quad \max_{z\neq z_*} \frac{\|z-z_*\|^2_{\mathbb{E}_{X\sim\nu}[\kappa\lambda(X)/\nu(X)XX^\top]^{-1}}}{\langle z_*-z,\theta_*\rangle^2}\beta \leq t$$
$$\kappa\lambda(x)/\nu(x) \leq 1 \quad \forall x\in\mathcal{X}$$

which is equal to, after simplifying,

$$\min_{\kappa\geq 0,\lambda\in\triangle_\mathcal{X}} t\,\kappa \quad \text{subject to} \quad \max_{z\neq z_*} \frac{\|z-z_*\|^2_{\mathbb{E}_{X\sim\lambda}[XX^\top]^{-1}}}{\langle z_*-z,\theta_*\rangle^2}\beta \leq t\kappa$$
$$\kappa\lambda(x)/\nu(x) \leq 1 \quad \forall x\in\mathcal{X}$$

which is equal to

$$\min_{u\geq 0,\lambda\in\triangle_\mathcal{X}} u \quad \text{subject to} \quad \rho(\lambda)\beta \leq u$$
$$\|\lambda/\nu\|_\infty \leq \frac{t}{u}.$$

Note, there exists a feasible $(\lambda,u)$ precisely when there exists a $\lambda\in\triangle_\mathcal{X}$ such that $\|\lambda/\nu\|_\infty\rho(\lambda)\leq t$, in which case the optimization problem is equal to

$$\min_{\lambda\in\triangle_\mathcal{X}} \rho(\lambda)\beta \quad \text{subject to} \quad \|\lambda/\nu\|_\infty\rho(\lambda)\beta \leq t$$

$\square$

## C  Analysis of the Optimization Problem

### C.1  Proof of Theorem 4

For simplicity, we will use $\mu$ instead of $\mu_b$ to denote the number that controls the intensity of barrier function.

The proof relies on analyzing another function $\overline{D}:\mathbb{R}^{d\times d}_{\succeq 0}\mapsto\mathbb{R}$. For simplicity, first, we define

$$h_\Lambda(x) = P_\Lambda(x) - \mu\left(\log(1-P_\Lambda(x)) + \log(P_\Lambda(x))\right) - P_\Lambda(x)x^\top\Lambda x. \tag{9}$$

Recall that our dual objective is $D(\mathbf{\Lambda}) = \mathbb{E}_{X\sim\nu}[h_\Lambda(X)] + \frac{1}{c_\ell^2}\sum_{y\in\mathcal{Y}_\ell} y^\top\Lambda_y y$. Since the first term in $\mathbb{E}_{X\sim\nu}[h_\Lambda(X)]$ only depends on $\Lambda = \sum_{y\in\mathcal{Y}_\ell}\Lambda_y$, we can consider the following optimization problem.

$$f(\Lambda) = \max_{\Lambda_y} \quad \sum_{y\in\mathcal{Y}_\ell} y^\top\Lambda_y y$$
$$\text{subject to} \quad \sum_{y\in\mathcal{Y}_\ell}\Lambda_y = \Lambda \tag{10}$$
$$\Lambda_y \succeq \mathbf{0}, \quad \forall y\in\mathcal{Y}_\ell.$$

Then, the alternative dual objective $\overline{D}(\Lambda)$ is defined as $\overline{D}(\Lambda) = \mathbb{E}_{X\sim\nu}[h_\Lambda(X)] + \frac{1}{c_\ell^2}f(\Lambda)$. We can immediately see that maximizing $\overline{D}(\cdot)$ is equivalent to maximizing $D(\cdot)$. In particular, let $\Lambda^* \in \operatorname{argmax}_{\Lambda\succeq\mathbf{0}}\overline{D}(\Lambda)$ and $(\Lambda_y^*)_{y\in\mathcal{Y}_\ell}$ be the set of PSD matrices that solve problem (10) and evaluate $f(\Lambda^*)$. We can see that $(\Lambda_y^*)_{y\in\mathcal{Y}_\ell}$ also maximizes $D(\cdot)$. Conversely, for $\mathbf{\Lambda}^* = (\Lambda_y^*)_{y\in\mathcal{Y}_\ell} \in \operatorname{argmax}_{\Lambda_y\succeq\mathbf{0},\forall y} D(\mathbf{\Lambda})$, we also have $\sum_{y\in\mathcal{Y}_\ell}\Lambda_y^* \in \operatorname{argmax}_{\Lambda\succeq\mathbf{0}}\overline{D}(\Lambda)$.

Further, we also define their empirical version $D_E$ and $\overline{D}_E$ with extra i.i.d. samples $x_1,\ldots,x_u$ as

$$D_E(\mathbf{\Lambda}) = \frac{1}{u}\sum_{i=1}^{u}h_\Lambda(x_i) + \frac{1}{c_\ell^2}\sum_{y\in\mathcal{Y}_\ell} y^\top\Lambda_y y \quad \text{and} \quad \overline{D}_E(\Lambda) = \frac{1}{u}\sum_{i=1}^{u}h_\Lambda(x_i) + \frac{1}{c_\ell^2}f(\Lambda). \tag{11}$$

Recall that the problem Algorithm 2 tries to solve is

$$\begin{aligned}
\min_P \quad & \mathbb{E}_{X\sim\nu}[P(X) - \mu(\log(1 - P(X)) + \log(P(X)))] \\
\text{subject to} \quad & \mathbb{E}_{X\sim\nu}\left[P(X)XX^\top\right] \succeq \tfrac{1}{c_\ell^2} yy^\top, \quad \forall y \in \mathcal{Y}_\ell.
\end{aligned} \tag{12}$$

We will restate a more precise version of Theorem 4 and then prove it.

**Theorem 5.** *Suppose $\|x\|_2 \leq M$ for any $x \in \operatorname{supp}(\nu)$ and $\Sigma = \mathbb{E}_{X\sim\nu}\left[XX^\top\right]$ is invertible. Let $\boldsymbol{\Lambda}^* \in \operatorname{argmax}_{\Lambda_y \succeq \mathbf{0}, \forall y} D(\boldsymbol{\Lambda})$ and $\kappa(\Sigma) = \frac{\lambda_{\max}(\Sigma)}{\lambda_{\min}(\Sigma)}$ be condition number. Assume $\|\Lambda^*\|_F > 0$ and define $\omega = \min_{\Gamma \in \mathbb{S}^d: \|\Gamma\|_F = 1} \mathbb{E}_{X\sim\nu}\left[\left(X^\top\Gamma X\right)^2\right]$, where $\mathbb{S}^d$ is the set of $d \times d$ symmetric matrices. Let $|\mathcal{Y}_\ell| \, C_\ell^2 = \frac{1}{c_\ell^2} \sum_{y\in\mathcal{Y}_\ell} \|y\|_2^4$.*

*Then, $\Lambda^* = \sum_{y\in\mathcal{Y}_\ell} \Lambda_y^*$ is unique. Further, for any $\epsilon > 0$ and $\delta > 0$, suppose it holds that*

$$\mu \leq \min\left\{ \sqrt{\frac{3\kappa(\Sigma)\,\|\Lambda^*\|_F\,M^2}{8} \cdot \frac{1+\epsilon}{\epsilon}}, \frac{4}{9}\|\Lambda^*\|_F^2\,M^4, \frac{1}{2\sqrt{3}} \right\}$$

$$K \geq \frac{288\kappa(\Sigma)^2\,|\mathcal{Y}_\ell|^3\,\|\Lambda^*\|_F^4\,M^4(M^4 + C_\ell^2)\cdot\left(2\,\|\Lambda^*\|_F\,M^2 + 1\right)^4\log(6/\delta)}{\omega^2\mu^6}\cdot\left(\frac{1+\epsilon}{\epsilon}\right)^2$$

$$u \geq \frac{576\kappa(\Sigma)^2\,\|\Lambda^*\|_F^2\,M^8\cdot\left(2\,\|\Lambda^*\|_F\,M^2 + 1\right)^4\log(6/\delta)}{\omega^2\mu^6}\cdot\left(\frac{1+\epsilon}{\epsilon}\right)^2.$$

*Then, with probability at least $1 - \delta$, Algorithm 2 will output $\widetilde{\Lambda}$ that satisfies*

- $y^\top \mathbb{E}_{X\sim\nu}\left[P_{\widetilde{\Lambda}}(X)XX^\top\right]^{-1} y \leq (1+\epsilon)c_\ell^2, \quad \forall y \in \mathcal{Y}_\ell.$

- $\mathbb{E}_{X\sim\nu}\left[P_{\widetilde{\Lambda}}(X)\right] \leq \mathbb{E}_{X\sim\nu}\left[\widetilde{P}(X)\right] + 4\sqrt{\mu}$, *where $\widetilde{P}$ is the optimal solution to problem* (20).

*Proof.* **First Bullet Point.** Fix some $\epsilon > 0$. Let $\hat{\boldsymbol{\Lambda}}$ and corresponding $\hat{\Lambda} = \sum_{y\in\mathcal{Y}_\ell} \hat{\Lambda}_y$ be the parameters obtained by Algorithm 2 just before the re-scaling step, which means that at line 10 of Algorithm 2, the assignment of $\hat{\Lambda}_y$ to each $y \in \mathcal{Y}_\ell$ has been optimized by solving problem (10). That is, we have $D(\hat{\boldsymbol{\Lambda}}) = \overline{D}(\hat{\Lambda})$ and $D_E(\hat{\boldsymbol{\Lambda}}) = \overline{D}_E(\hat{\Lambda})$. Let $\widetilde{\boldsymbol{\Lambda}}$ and $\widetilde{\Lambda}$ be the ones after the re-scaling step. Then, by Theorem 3.13 of [21], with probability at least $1 - \frac{\delta}{3}$, it holds that

$$\overline{D}(\Lambda^*) - \overline{D}(\hat{\Lambda}) = D(\boldsymbol{\Lambda}^*) - D(\hat{\boldsymbol{\Lambda}}) \leq \frac{\operatorname{Reg}(K) + 2\sqrt{2K\log(6/\delta)}}{K},$$

where $\operatorname{Reg}(K)$ is the regret of running projected stochastic gradient ascent for $K$ steps with $\eta_k$ specified in Algorithm 2. Meanwhile, by Theorem 4.14 of [21] also, we have $\operatorname{Reg}(K) = \sqrt{2}B^2\sqrt{\sum_{k=1}^K \sum_{y\in\mathcal{Y}_\ell} \|g_{k,y}\|_2^2}$, where $B = \sqrt{|\mathcal{Y}_\ell|}\|\Lambda^*\|_F$ bound the norm of $\boldsymbol{\Lambda}^* = \left(\Lambda_y^*\right)_{y\in\mathcal{Y}_\ell}$. Since $g_{k,y} = \frac{yy^\top}{c_\ell^2} - P_{\hat{\Lambda}^{(k)}}(x_k)x_k x_k^\top$, we can easily get $\sum_{y\in\mathcal{Y}_\ell} \|g_{k,y}\|_2^2 \leq 2\,|\mathcal{Y}_\ell|\,M^4 + \frac{2}{c_\ell^2}\sum_{y\in\mathcal{Y}_\ell} \|y\|_2^4 = 2\,|\mathcal{Y}_\ell|\,M^4 + 2\,|\mathcal{Y}_\ell|\,C_\ell^2$. Thus, we have

$$\operatorname{Reg}(K) \leq 2\,|\mathcal{Y}_\ell|\,\|\Lambda^*\|_F^2\,\sqrt{|\mathcal{Y}_\ell|\,M^4 + |\mathcal{Y}_\ell|\,C_\ell^2}\cdot\sqrt{K} := C_{\operatorname{Reg}}\sqrt{K} \tag{13}$$

$$\implies \overline{D}(\Lambda^*) - \overline{D}(\hat{\Lambda}) \leq \frac{C_{\operatorname{Reg}} + 2\sqrt{2\log(6/\delta)}}{\sqrt{K}}, \tag{14}$$

We now consider the effect of using $u$ i.i.d. samples in the re-scaling step. First, since re-scaling always increases the function value, we must have $D_E(\hat{\boldsymbol{\Lambda}}) \leq D_E(\widetilde{\boldsymbol{\Lambda}})$. Meanwhile, since $D_E(\hat{\boldsymbol{\Lambda}}) = \overline{D}_E(\hat{\Lambda})$, by Lemma 10, we have $D_E(\hat{\boldsymbol{\Lambda}}) = \overline{D}_E(\hat{\Lambda})$, which together implies $\overline{D}_E(\hat{\Lambda}) \leq \overline{D}_E(\widetilde{\Lambda})$.

By Lemma 5, we know that $\Lambda^*$ is unique and as long as $\mu \leq \frac{1}{2\sqrt{3}}$, $\overline{D}(\Lambda)$ is $G$-strongly concave with respect to $\ell_2$ norm over $\mathcal{S} = \{\Lambda \succeq \mathbf{0} : \|\Lambda\|_F \leq 2\|\Lambda^*\|_F\}$, where $G$ is defined in Eq. (21). Thus, by Lemma 11, if $K$ is large enough such that

$$\overline{D}(\Lambda^*) - \overline{D}(\hat{\Lambda}) \leq \frac{C_{\text{Reg}} + 2\sqrt{2\log(6/\delta)}}{\sqrt{K}} \leq \frac{G\|\Lambda^*\|_F}{2},$$

then $\left\|\hat{\Lambda} - \Lambda^*\right\|_F \leq \|\Lambda^*\|_F$, which implies $\left\|\hat{\Lambda}\right\|_F \leq 2\|\Lambda^*\|_F$. That is, $\hat{\Lambda} \in \mathcal{S}$. Then, under this condition, by using Lemma 8, when $\mu \leq \frac{4}{9}\|\Lambda^*\|_F M^4$ and

$$u \geq \left(\frac{6\kappa(\Sigma)\|\Lambda^*\|_F M^4 \left(2 + \sqrt{2\log(6/\delta)}\right)}{G\mu^2} \cdot \frac{1+\epsilon}{\epsilon}\right)^2, \tag{15}$$

for $\widetilde{\Lambda}$ after re-scaling, with probability at least $1 - \frac{\delta}{3}$, it holds simultaneously that

$$\left|\overline{D}_E(\hat{\Lambda}) - \overline{D}(\hat{\Lambda})\right| \leq \frac{G\mu^2}{3M^2\kappa(\Sigma)} \cdot \frac{\epsilon}{1+\epsilon} \quad \text{and} \quad \left|\overline{D}_E(\widetilde{\Lambda}) - \overline{D}(\widetilde{\Lambda})\right| \leq \frac{G\mu^2}{3M^2\kappa(\Sigma)} \cdot \frac{\epsilon}{1+\epsilon} \tag{16}$$

$$\begin{aligned}
\implies \overline{D}(\Lambda^*) - \overline{D}(\widetilde{\Lambda}) &\leq \overline{D}(\Lambda^*) - \overline{D}(\hat{\Lambda}) + \overline{D}(\hat{\Lambda}) - \overline{D}(\widetilde{\Lambda}) \\
&\leq \overline{D}(\Lambda^*) - \overline{D}(\hat{\Lambda}) + \overline{D}(\hat{\Lambda}) - \overline{D}_E(\hat{\Lambda}) + \overline{D}_E(\widetilde{\Lambda}) - \overline{D}(\widetilde{\Lambda}) \\
&\hspace{4cm} (\text{Since } \overline{D}_E(\hat{\Lambda}) \leq \overline{D}_E(\widetilde{\Lambda})) \\
&\leq \frac{C_{\text{Reg}} + 2\sqrt{2\log(6/\delta)}}{\sqrt{K}} + \frac{2G\mu^2}{3M^2\kappa(\Sigma)} \cdot \frac{\epsilon}{1+\epsilon}. \quad (\text{By Eq. (14) and (16)})
\end{aligned}$$

Since $\widetilde{\Lambda}$ is a smaller re-scaling of $\hat{\Lambda}$, we have $\widetilde{\Lambda} \in \mathcal{S}$, which implies $\frac{G}{2}\left\|\Lambda^* - \widetilde{\Lambda}\right\|_F \leq \overline{D}(\Lambda^*) - \overline{D}(\widetilde{\Lambda})$ by property of strongly concave function [3]. Therefore, by Lemma 12, to guarantee an at most $\epsilon$ multiplicative constraint violation, it is sufficient to choose $K$ such that

$$\begin{aligned}
\frac{G}{2}\left\|\Lambda^* - \widetilde{\Lambda}\right\|_F &\leq \overline{D}(\Lambda^*) - \overline{D}(\widetilde{\Lambda}) \\
&\leq \frac{C_{\text{Reg}} + 2\sqrt{2\log(6/\delta)}}{\sqrt{K}} + \frac{2G\mu^2}{3M^2\kappa(\Sigma)} \cdot \frac{\epsilon}{1+\epsilon} \\
&\leq \min\left\{\frac{4G\mu^2}{3M^2\kappa(\Sigma)} \cdot \frac{\epsilon}{1+\epsilon}, \frac{G\|\Lambda^*\|_F}{2}\right\} \\
&= \frac{4G\mu^2}{3M^2\kappa(\Sigma)} \cdot \frac{\epsilon}{1+\epsilon}. \quad (\text{If } \mu \leq \sqrt{\frac{3\kappa(\Sigma)\|\Lambda^*\|_F M^2}{8} \cdot \frac{1+\epsilon}{\epsilon}})
\end{aligned}$$

An algebraic rearrangement gives us

$$K \geq \left(\frac{3\kappa(\Sigma)M^2 \left(C_{\text{Reg}} + 2\sqrt{2\log(6/\delta)}\right)}{2G\mu^2} \cdot \frac{1+\epsilon}{\epsilon}\right)^2. \tag{17}$$

**Second Bullet Point.** We then prove the upper bound for primal objective value $\mathbb{E}_{X \sim \nu}\left[P_{\widetilde{\Lambda}}(X)\right]$, which explains the reason why an extra re-scaling step is needed. Define $g(s) = D_E(s \cdot \widetilde{\Lambda})$. By construction, we know that $g(s)$ is maximized at $s = 1$ because $\widetilde{\Lambda} = s^* \cdot \hat{\Lambda}$, where $s^* = \arg\max_{s \in [0,1]} D_E(s \cdot \hat{\Lambda})$. Therefore, we have $g'(1) \geq 0$, which in turn gives us

$$g'(1) = \frac{1}{c_\ell^2}\sum_{y \in \mathcal{Y}_\ell} y^\top \widetilde{\Lambda}_y y - \frac{1}{u}\sum_{i=1}^u P_{\widetilde{\Lambda}}(x_i)x_i^\top \widetilde{\Lambda}x_i \geq 0.$$

By the concentration inequality in Lemma 8, we know that when

$$u \geq \left(\frac{2\|\Lambda^*\|_F M^2 \left(\|\Lambda^*\|_F M^2 + \mu\sqrt{2\log(6/\delta)}\right)}{\mu^{3/2}}\right)^2, \tag{18}$$

with probability at least $1 - \frac{\delta}{3}$, it holds that

$$\left| \mathbb{E}_{X\sim\nu} \left[ P_\Lambda(X) X^\top \Lambda X \right] - \frac{1}{u} \sum_{i=1}^{u} P_\Lambda(x_i) x_i^\top \Lambda x_i \right| \leq \sqrt{\mu}$$

$$\implies \frac{1}{c_\ell^2} \sum_{y\in\mathcal{Y}_\ell} y^\top \widetilde{\Lambda}_y y - \mathbb{E}_{X\sim\nu} \left[ P_{\widetilde{\Lambda}}(X) X^\top \widetilde{\Lambda} X \right] + \sqrt{\mu} \geq 0. \tag{19}$$

Now, let $\widetilde{P}$ be the optimal solution of problem (20) and $\hat{P}$ be the optimal solution of the same problem with bound constraint $\mu \leq P(x) \leq 1 - \mu$.

$$\begin{aligned} \min_P \quad & \mathbb{E}_{X\sim\nu} \left[ P(X) \right] \\ \text{subject to} \quad & y^\top \mathbb{E}_{X\sim\nu} \left[ P(X) XX^\top \right]^{-1} y \leq c_\ell^2, \quad \forall y \in \mathcal{Y}_\ell, \\ & 0 \leq P(x) \leq 1 - \mu, \quad \forall x \in \mathcal{X}. \end{aligned} \tag{20}$$

Then, we can notice that

$$\mathbb{E}_{X\sim\nu} \left[ P_{\widetilde{\Lambda}}(X) \right]$$

$$\leq \mathbb{E}_{X\sim\nu} \left[ P_{\widetilde{\Lambda}}(X) - \mu(\log(1 - P_{\widetilde{\Lambda}}(X)) + \log(P_{\widetilde{\Lambda}}(X))) \right]$$

$$\leq \mathbb{E}_{X\sim\nu} \left[ P_{\widetilde{\Lambda}}(X) - \mu(\log(1 - P_{\widetilde{\Lambda}}(X)) + \log(P_{\widetilde{\Lambda}}(X))) \right]$$

$$\qquad + \frac{1}{c_\ell^2} \sum_{y\in\mathcal{Y}_\ell} y^\top \widetilde{\Lambda}_y y - \mathbb{E}_{X\sim\nu} \left[ P_{\widetilde{\Lambda}}(X) X^\top \widetilde{\Lambda} X \right] + \sqrt{\mu} \qquad \text{(By Eq. (19))}$$

$$= \inf_P \mathcal{L}(P, \widetilde{\boldsymbol{\Lambda}}) + \sqrt{\mu} \qquad \text{(By definition of Lagrangian function and how we solve for } P_\Lambda)$$

$$\leq \max_{\Lambda_y \succeq \mathbf{0}, \forall y \in \mathcal{Y}_\ell} \inf_P \mathcal{L}(P, \boldsymbol{\Lambda}) + \sqrt{\mu}$$

$$= \mathbb{E}_{X\sim\nu} \left[ P_{\Lambda^*}(X) - \mu(\log(1 - P_{\Lambda^*}(X)) + \log(P_{\Lambda^*}(X))) \right] + \sqrt{\mu}$$

$$\leq \mathbb{E}_{X\sim\nu} \left[ \hat{P}(X) - \mu\log(1 - \hat{P}(X)) \right] - \mu\log\left( \hat{P}(X) \right) + \sqrt{\mu}$$

$$\qquad\qquad\qquad\qquad\qquad\qquad\qquad \text{(Since } \hat{P} \text{ is feasible to problem (12))}$$

$$\leq \mathbb{E}_{X\sim\nu} \left[ \hat{P}(X) \right] + 3\sqrt{\mu}, \qquad\qquad \text{(Since } -a\log(a) \leq \sqrt{a} \text{ for } a \in (0,1))$$

$$\leq \mathbb{E}_{X\sim\nu} \left[ \widetilde{P}(X) \right] + 4\sqrt{\mu}. \qquad \text{(Since } \hat{P}(x) \text{ can have at most } \mu \text{ more contribution than } \widetilde{P})$$

Therefore, in summary, Suppose $K$ and $u$ satisfy conditions specified in Eq. (17), (15) and (18) and $\mu \leq \min\left\{ \sqrt{\frac{3\kappa(\Sigma)\|\Lambda^*\|_F M^2}{8} \cdot \frac{1+\epsilon}{\epsilon}}, \frac{4}{9}\|\Lambda^*\|_F^2 M^4, \frac{1}{2\sqrt{3}} \right\}$, where $C_{\text{Reg}}$ and $G$ are defined in Eq. (13) and (21), respectively. Then. by applying a simple union bound, with probability at least $1 - \delta$, the output of Algorithm 2 $\widetilde{\Lambda}$ satisfies $y^\top \mathbb{E}_{X\sim\nu} \left[ P(X) XX^\top \right]^{-1} y \leq (1+\epsilon)c_\ell^2, \forall y \in \mathcal{Y}_\ell$ and $\mathbb{E}_{X\sim\nu} \left[ P_{\widetilde{\Lambda}}(X) \right] \leq \mathbb{E}_{X\sim\nu} \left[ \widetilde{P}(X) \right] + 4\sqrt{\mu}$. $\qquad\square$

## C.2   Relevant Lemmas

### C.2.1   Strong Concavity of $\overline{D}(\Lambda)$

**Lemma 5.** *As long as $\mu \leq \frac{1}{2\sqrt{3}}$, $\overline{D}(\Lambda)$ is $G$-strongly concave with respect to $\ell_2$-norm on the bounded region $\mathcal{S} = \{\Lambda \succeq \mathbf{0} : \|\Lambda\|_F \leq 2\|\Lambda^*\|_F\}$ with coefficient*

$$G = \frac{\mu}{2\left( 2\|\Lambda^*\|_F M^2 + 1 \right)^2} \cdot \min_{\Gamma\in\mathbb{S}^d:\|\Gamma\|_F=1} \mathbb{E}_{X\sim\nu} \left[ \left( X^\top \Gamma X \right)^2 \right]. \tag{21}$$

*Because of this, as a corollary, $\Lambda^*$ will be unique.*

*Proof.* By Lemma 6, since $f(\Lambda)$ is concave in $\Lambda$, it is sufficient to prove that $\mathbb{E}_{X\sim\nu} \left[ h_\Lambda(X) \right]$ is $G$-strongly concave on $\mathcal{S}$, where $h_\Lambda(x)$ is defined in Eq. (9). Then, we have

$$-\nabla_\Lambda^2 \mathbb{E}_{X\sim\nu} \left[ h_\Lambda(X) \right] = \mathbb{E}_{X\sim\nu} \left[ \frac{\mathrm{d}P_\Lambda}{\mathrm{d}q_\Lambda}(X) \mathrm{vec}\left( XX^\top \right) \mathrm{vec}\left( XX^\top \right)^\top \right].$$

Since $\|x\|_2 \leq M$, for any $\Lambda \in \mathcal{S}$, we have $q_\Lambda(x) = x^\top \Lambda x - 1 \leq 2 \|\Lambda^*\|_F M^2 + 1$. By Lemma, 14, we know that if $12\mu^2 \leq \left(2 \|\Lambda^*\|_F M^2 + 1\right)^2$, which can be done by choosing $\mu \leq \frac{1}{2\sqrt{3}}$, we have $\frac{\mathrm{d}P_\Lambda}{\mathrm{d}q_\Lambda}(x) \geq \frac{\mu}{2\left(2\|\Lambda^*\|_F M^2+1\right)^2}$ for any $x \in \mathcal{X}$ and $\Lambda \in \mathcal{S}$. Therefore, we have

$$-\nabla_\Lambda^2 \mathbb{E}_{X\sim\nu}\left[h_\Lambda(X)\right] \succeq \gamma \cdot \mathbb{E}_{X\sim\nu}\left[\mathrm{vec}\left(XX^\top\right)\mathrm{vec}\left(XX^\top\right)^\top\right]$$

Now, let $\mathbb{S}$ be the set of all $d \times d$ symmetric matrices. It is obvious that $\mathbb{S}$ is a subspace of the vector space of all $d \times d$ matrices and $\mathcal{S} \subseteq \mathbb{S}$. Thus, by applying Lemma 7, we can conclude that $\mathbb{E}_{X\sim\nu}\left[h_\Lambda(X)\right]$ is $G$-strongly concave on $\mathcal{S}$ with respect to $\ell_2$ norm and

$$G = \frac{\mu}{2\left(2\|\Lambda^*\|_F M^2+1\right)^2} \cdot \min_{\Gamma\in\mathbb{S}^d:\|\Gamma\|_F=1} \mathrm{vec}(\Gamma)^\top \mathbb{E}_{X\sim\nu}\left[\mathrm{vec}\left(XX^\top\right)\mathrm{vec}\left(XX^\top\right)^\top\right]\mathrm{vec}(\Gamma)$$

$$= \frac{\mu}{2\left(2\|\Lambda^*\|_F M^2+1\right)^2} \cdot \min_{\Gamma\in\mathbb{S}^d:\|\Gamma\|_F=1} \mathbb{E}_{X\sim\nu}\left[\left(X^\top\Gamma X\right)^2\right].$$

Thus the proof is complete. $\qquad \square$

**Lemma 6.** $f(\Lambda)$ *defined in Eq.* (10) *is concave in* $\Lambda$.

*Proof.* To show its concavity, consider $\Lambda^{(1)} \succeq 0$, $\Lambda^{(2)} \succeq 0$ and some $\gamma \in (0,1)$. Let $(\Lambda_y^{(i)})_{y\in\mathcal{Y}_\ell}$ be the optimal solution obtained by evaluating $f(\Lambda^{(i)})$ for $i \in \{1,2\}$. Then, we can notice that

$$\gamma f(\Lambda^{(1)}) + (1-\gamma)f(\Lambda^{(2)}) = \gamma \sum_{y\in\mathcal{Y}_\ell} y^\top\Lambda_y^{(1)}y + (1-\gamma)\sum_{y\in\mathcal{Y}_\ell} y^\top\Lambda_y^{(2)}y$$

$$= \sum_{y\in\mathcal{Y}_\ell} y^\top(\gamma\Lambda_y^{(1)} + (1-\gamma)\Lambda_y^{(2)})y$$

$$\leq f(\gamma\Lambda^{(1)} + (1-\gamma)\Lambda^{(2)}).$$

The last inequality above holds because $\sum_{y\in\mathcal{Y}_\ell}\Lambda_y^{(i)} = \Lambda^{(i)}$ for $i \in \{1,2\}$ and thus $\sum_{y\in\mathcal{Y}_\ell}\left(\gamma\Lambda_y^{(1)} + (1-\gamma)\Lambda_y^{(2)}\right) = \gamma\Lambda^{(1)} + (1-\gamma)\Lambda^{(2)}$, which means that $(\gamma\Lambda_y^{(1)} + (1-\gamma)\Lambda_y^{(2)})_{y\in\mathcal{Y}_\ell}$ is a feasible solution for problem (10) with parameter $\gamma\Lambda^{(1)} + (1-\gamma)\Lambda^{(2)}$. Therefore, we can conclude that $f(\Lambda)$ is concave in $\Lambda$. $\qquad \square$

**Lemma 7.** *Let* $f : \mathbb{R}^d \mapsto \mathbb{R}$ *be a convex and twice differentiable function in* $\mathbb{R}^d$. *If for some subspace* $S \subseteq \mathbb{R}^d$, *we have* $\min_{w\in S:\|w\|_2=1} w^\top\nabla^2 f(x)w \geq \sigma > 0$, $\forall x \in S$, *then* $f$ *is* $\sigma$-*strongly convex with respect to* $\ell_2$-*norm on* $S$.

*Proof.* Suppose $S$ has dimension $m$ and let $v_1, \ldots, v_m$ be a set of orthonormal basis that span $S$. Then, for each $x \in S$, there exists unique $z \in \mathbb{R}^m$ such that $x = Vz$, where $V = [v_1 \quad \cdots \quad v_m]$. That is, there is one-to-one correspondence between $S$ and $\mathbb{R}^m$.

Now, we define $g : \mathbb{R}^m \mapsto \mathbb{R}$ as $g(z) = f(Vz)$. It is easy to compute $\nabla^2 g(z) = V^\top\nabla^2 f(Vz)V$. Then, notice that for any $w' \in \mathbb{R}^m$ such that $\|w'\|_2 = 1$, we have $Vw' \in S$ and $\|Vw'\|_2 = \sqrt{w'^\top V^\top V w'} = \sqrt{w'^\top w'} = 1$. Thus, we have

$$\min_{w'\in\mathbb{R}^m:\|w'\|_2=1} w'^\top\nabla^2 g(z)w' = \min_{w'\in\mathbb{R}^m:\|w'\|_2=1} w'^\top V^\top\nabla^2 f(Vz)Vw'$$

$$= \min_{w\in S:\|w\|_2=1} w^\top\nabla^2 f(Vz)w \geq \sigma.$$

Therefore, $g$ is $\sigma$-strongly convex with respect to $\ell_2$ norm. Then, for any $x_1, x_2 \in S$, there exists unique $z_1, z_2 \in \mathbb{R}^m$ such that $x_1 = Vz_1$ and $x_2 = Vz_2$. Notice that $\|z_1 - z_2\|_2 = \|x_1 - x_2\|_2$ since $V$ preserves the norm. Further, by definition of strong convexity, for any $\alpha \in [0,1]$, we have

$$g(\alpha z_1 + (1-\alpha)z_2) + \frac{\sigma}{2}\alpha(1-\alpha)\|z_1 - z_2\|_2^2 \leq \alpha g(z_1) + (1-\alpha)g(z_2)$$

$$\implies f(\alpha Vz_1 + (1-\alpha)Vz_2) + \frac{\sigma}{2}\alpha(1-\alpha)\|x_1 - x_2\|_2^2 \leq \alpha f(Vz_1) + (1-\alpha)f(Vz_2)$$

$$\implies f(\alpha x_1 + (1-\alpha)x_2) + \frac{\sigma}{2}\alpha(1-\alpha)\|x_1 - x_2\|_2^2 \leq \alpha f(x_1) + (1-\alpha)f(x_2).$$

Thus, $f$ is also $\sigma$-strongly convex with respect to $\ell_2$ norm on $S$. $\qquad \square$

### C.2.2 Concentration Inequalities

**Lemma 8.** *Let $x_1, \ldots, x_u \sim \nu$ be i.i.d. samples. If $\left\|\hat{\Lambda}\right\|_F \leq 2 \left\|\Lambda^*\right\|_F$, $\|x\|_2 \leq M$ for any $x \in \mathcal{X}$ and $\mu \leq \frac{4}{9} \left\|\Lambda^*\right\|_F^2 M^4$, then with probability at least $1 - \frac{2\delta}{3}$, it holds for any $\Lambda \in \Theta = \left\{ s \cdot \hat{\Lambda} : s \in [0, 1] \right\}$ simultaneously that*

$$\left| \mathbb{E}_{X \sim \nu} \left[ h_\Lambda(X) \right] - \frac{1}{u} \sum_{i=1}^u h_\Lambda(x_i) \right| \leq \frac{2 \left\|\Lambda^*\right\|_F M^2 \left( 2 + \sqrt{2 \log(6/\delta)} \right)}{\sqrt{u}}$$

$$\left| \mathbb{E}_{X \sim \nu} \left[ P_\Lambda(X) X^\top \Lambda X \right] - \frac{1}{u} \sum_{i=1}^u P_\Lambda(x_i) x_i^\top \Lambda x_i \right| \leq \frac{2 \left\|\Lambda^*\right\|_F M^2 \left( \left\|\Lambda^*\right\|_F M^2 + \mu \sqrt{2 \log(6/\delta)} \right)}{\mu \sqrt{u}}.$$

*Proof.* To prove the first inequality, first, notice that we have $h_\Lambda(x) = -P_\Lambda(x) q_\Lambda(x) - \mu \left( \log(1 - P_\Lambda(x)) + \log(P_\Lambda(x)) \right)$, where $q_\Lambda(x) = x^\top \Lambda x - 1$. Since $P_\Lambda(x)$, defined in Eq. (7), explicitly only depends on $q_\Lambda(x)$ instead of $x$ directly, we can treat $h_\Lambda$ as a function of $q_\Lambda$ and define a function class $\mathcal{F} = \left\{ x \mapsto x^\top (s \cdot \hat{\Lambda}) x : s \in [0, 1] \right\}$. It is well-known that if $h_\Lambda$ is $L_1$-Lipschitz in $q_\Lambda$ and $|h_\Lambda(x)| \leq R_1$ for any $\Lambda \in \Theta$ and $x \sim \nu$, then, with probability at least $1 - \frac{\delta}{3}$, it holds simultaneously for all $\Lambda \in \Theta$ that [2, 19]

$$\left| \mathbb{E}_{X \sim \nu} \left[ h_\Lambda(X) \right] - \frac{1}{u} \sum_{i=1}^u h_\Lambda(x_i) \right| \leq 2 L_1 \cdot \mathcal{R}_u(\mathcal{F}) + R_1 \sqrt{\frac{2 \log(6/\delta)}{u}}, \tag{22}$$

where $\mathcal{R}_u(\mathcal{F})$ is the Rademacher complexity of $\mathcal{F}$.

To find $L_1$, we can compute

$$\begin{aligned}
\frac{\mathrm{d} h_\Lambda}{\mathrm{d} q_\Lambda} &= -\frac{\mathrm{d} P_\Lambda}{\mathrm{d} q_\Lambda} q_\Lambda - P_\Lambda + \frac{\mathrm{d} P_\Lambda}{\mathrm{d} q_\Lambda} \left( \frac{\mu}{1 - P_\Lambda} - \frac{\mu}{P_\Lambda} \right) \\
&= -\frac{\mathrm{d} P_\Lambda}{\mathrm{d} \cdot q_\Lambda} q_\Lambda - P_\Lambda + \frac{\mathrm{d} P_\Lambda}{\mathrm{d} q_\Lambda} \cdot q_\Lambda &&\text{(Since } P_\Lambda \text{ satisfies Eq. (6))} \\
&= -P_\Lambda
\end{aligned}$$

Therefore, we have $\frac{\mathrm{d} h_\Lambda}{\mathrm{d} q_\Lambda} \in \left[ -1, -\frac{\mu}{3} \right]$ by Lemma 14. Therefore, we can set $L_1 = 1$.

Let $h_0$ be the value of $h_\Lambda$ when $q_\Lambda = -1$, which means $x^\top \Lambda x = 0$. To find $R_1$, notice that since $\frac{\mathrm{d} h_\Lambda}{\mathrm{d} q_\Lambda} \in \left[ -1, -\frac{\mu}{3} \right]$, we must have $-q_\Lambda + h_0 \leq h_\Lambda \leq -\frac{\mu}{3} q_\Lambda + h_0$. By Lemma 14, we know that $h_0 \in \left[ 0, 2\sqrt{\mu} \right]$. Therefore, we have $-x^\top \Lambda x \leq h_\Lambda(x) \leq -\frac{\mu}{3} x^\top \Lambda x + 3\sqrt{\mu}$ for any $x \in \mathcal{X}$ and $\Lambda \in \Theta$. Since $\|\Lambda\|_F \leq \left\|\hat{\Lambda}\right\|_F \leq 2 \left\|\Lambda^*\right\|_F$, we have $|h_\Lambda(x)| \leq 2 \left\|\Lambda^*\right\|_F M^2 := R_1$, which holds when $\mu \leq \frac{4}{9} \left\|\Lambda^*\right\|_F^2 M^4$. Then, by Lemma 9, we know that $\mathcal{R}_u(\mathcal{F}) \leq \frac{2 \|\Lambda^*\|_F M^2}{\sqrt{u}}$. Thus, plugging in values of $L_1$, $R_1$ and $\mathcal{R}_u(\mathcal{F})$ into Eq. (22) gives our first concentration inequality.

We can basically follow exactly the same strategy to prove the second concentration inequality. In particular, define $\tilde{h}_\Lambda(x) = P_\Lambda(x) x^\top \Lambda x = P_\Lambda(x) q_\Lambda(x) + P_\Lambda(x)$. Then, with probability at least $1 - \frac{\delta}{3}$, it holds simultaneously for any $\Lambda \in \Theta$ that

$$\left| \mathbb{E}_{X \sim \nu} \left[ \tilde{h}_\Lambda(X) \right] - \frac{1}{u} \sum_{i=1}^u \tilde{h}_\Lambda(x_i) \right| \leq 2 L_2 \cdot \mathcal{R}_u(\mathcal{F}) + R_2 \sqrt{\frac{2 \log(6/\delta)}{u}}, \tag{23}$$

where $\left| \tilde{h}_\Lambda(x) \right| \leq R_2$ for any $x \in \mathcal{X}$, $\Lambda \in \Theta$ and $\tilde{h}_\Lambda$ is $L_2$-Lipschitz in $q_\Lambda$.

To find $L_2$, we can compute

$$\frac{\mathrm{d} \tilde{h}_\Lambda}{\mathrm{d} q_\Lambda} = P_\Lambda + \frac{\mathrm{d} P_\Lambda}{\mathrm{d} q_\Lambda} \cdot x^\top \Lambda x.$$

By Lemma 14, we know that $\frac{dP_\Lambda}{dq_\Lambda} \in \left[0, \frac{1}{8\mu}\right]$. Thus, we have $\left|\frac{d\tilde{h}_\Lambda}{dq_\Lambda}\right| \leq 1 + \frac{\|\Lambda^*\|_F M^2}{4\mu} := L_2$. It is obvious that $\tilde{h}_\Lambda(x) \leq 2\|\Lambda^*\|_F M^2 := R_2$. Thus, by plugging the values of $L_2$, $R_2$ and $\mathcal{R}_u(\mathcal{F})$ into Eq. (23), we can obtain the second concentration inequality.

Finally, both concentration inequalities hold simultaneously with probability at least $1 - \frac{2\delta}{3}$ by a simple union bound. $\qquad\square$

**Lemma 9.** *If* $\left\|\hat{\Lambda}\right\|_F \leq 2\|\Lambda^*\|_F$, *then, we have* $\mathcal{R}_u(\mathcal{F}) \leq \sqrt{\frac{\mathbb{E}_{X\sim\nu}[(X^\top \hat{\Lambda}X)^2]}{u}} \leq \frac{2\|\Lambda^*\|_F M^2}{\sqrt{u}}$, *where* $\mathcal{F} = \left\{x \mapsto x^\top(s \cdot \hat{\Lambda})x : s \in [0,1]\right\}$.

*Proof.* Let $\sigma_1, \ldots, \sigma_u$ be i.i.d. Rademacher random variables, which are uniform over $\{-1, +1\}$. Let $x_1, \ldots, x_u \sim \nu$ be i.i.d. samples. Then, by definition of Rademacher complexity, we have

$$\mathcal{R}_u(\mathcal{F}) = \mathbb{E}\left[\sup_{q\in\mathcal{F}} \frac{1}{u} \sum_{i=1}^u \sigma_i q(x_i)\right]$$

$$= \mathbb{E}\left[\sup_{s\in[0,1]} \frac{1}{u} \sum_{i=1}^u \sigma_i x_i^\top (s\hat{\Lambda})x_i\right] \qquad \text{(By definition of } \mathcal{F}\text{)}$$

$$\overset{(i)}{=} \frac{1}{u}\mathbb{E}\left[\mathbb{1}\left\{\sum_{i=1}^n \sigma_i x_i^\top \hat{\Lambda}x_i \geq 0\right\} \sum_{i=1}^n \sigma_i x_i^\top \hat{\Lambda}x_i\right].$$

$$\leq \frac{1}{u}\mathbb{E}\left[\left|\sum_{i=1}^u \sigma_i x_i^\top \hat{\Lambda}x_i\right|\right]$$

$$\leq \frac{1}{u}\sqrt{\mathbb{E}\left[\left(\sum_{i=1}^u \sigma_i x_i^\top \hat{\Lambda}x_i\right)^2\right]} \qquad \text{(By Jensen's inequality)}$$

$$= \frac{1}{u}\sqrt{\mathbb{E}\left[\sum_{i=1}^u \left(x_i^\top \hat{\Lambda}x_i\right)^2\right]} \qquad \text{(Since } \sigma_i\text{'s are i.i.d. and } \mathbb{E}[\sigma_i] = 0\text{)}$$

$$= \sqrt{\frac{\mathbb{E}_{X\sim\nu}\left[\left(X^\top \hat{\Lambda}X\right)^2\right]}{u}} \leq \frac{2\|\Lambda^*\|_F M^2}{\sqrt{u}}.$$

Here, the equality (i) holds because when $\sum_{i=1}^n \sigma_i x_i^\top \hat{\Lambda}x_i < 0$, the supremum over $s \in [0,1]$ will be obtained by taking $s = 0$; otherwise, it will be obtained by taking $s = 1$. $\qquad\square$

### C.2.3 Other Lemmas

The following lemma basically shows that $f(\Lambda)$ is linear in scalar multiplication.

**Lemma 10.** *If* $D_E(\hat{\Lambda}) = \overline{D}_E(\hat{\Lambda})$, *with* $\hat{\Lambda} = \sum_{y\in\mathcal{Y}_\ell} \hat{\Lambda}_y$, *then, for any* $s \geq 0$, *it holds that* $D_E(s \cdot \hat{\Lambda}) = \overline{D}_E(s \cdot \hat{\Lambda})$, *where* $D_E$ *and* $\overline{D}_E$ *are defined in Eq.* (11).

*Proof.* It suffices to show that if $\sum_{y\in\mathcal{Y}_\ell} y^\top \hat{\Lambda}_y y = f(\hat{\Lambda})$, then $\sum_{y\in\mathcal{Y}_\ell} y^\top(s \cdot \hat{\Lambda}_y)y = f(s \cdot \hat{\Lambda})$ for any $s > 0$. By definition, we have

$$\begin{aligned}
f(s \cdot \hat{\Lambda}) = \max_{\Lambda_y} \quad & \sum_{y\in\mathcal{Y}_\ell} y^\top \Lambda_y y \\
\text{subject to} \quad & \sum_{y\in\mathcal{Y}_\ell} \Lambda_y = s \cdot \hat{\Lambda} \\
& \Lambda_y \succeq \mathbf{0}, \quad \forall y \in \mathcal{Y}_\ell.
\end{aligned}$$

For the above optimization problem, we can do a change of variable by setting $\Lambda'_y = \frac{1}{s} \cdot \Lambda_y \implies \Lambda_y = s \cdot \Lambda'_y$. Then, we have

$$
\begin{aligned}
f(s \cdot \hat{\Lambda}) = \max_{\Lambda_y} \quad & \sum_{y \in \mathcal{Y}_\ell} y^\top (s \cdot \Lambda'_y) y \\
\text{subject to} \quad & \sum_{y \in \mathcal{Y}_\ell} s \cdot \Lambda'_y = s \cdot \hat{\Lambda} \\
& s \cdot \Lambda'_y \succeq \mathbf{0}, \quad \forall y \in \mathcal{Y}_\ell.
\end{aligned}
$$

$$
\begin{aligned}
\implies f(s \cdot \hat{\Lambda}) = \max_{\Lambda_y} \quad & s \sum_{y \in \mathcal{Y}_\ell} y^\top \Lambda'_y y \\
\text{subject to} \quad & \sum_{y \in \mathcal{Y}_\ell} \Lambda'_y = \hat{\Lambda} \\
& \Lambda'_y \succeq \mathbf{0}, \quad \forall y \in \mathcal{Y}_\ell.
\end{aligned}
$$

$$
\implies f(s \cdot \hat{\Lambda}) = s \cdot f(\hat{\Lambda}) = s \cdot \sum_{y \in \mathcal{Y}_\ell} y^\top \Lambda_y y = \sum_{y \in \mathcal{Y}_\ell} y^\top (s \cdot \hat{\Lambda}_y) y.
$$

Thus, the proof is complete. $\qquad \square$

**Lemma 11.** *Let $f : \mathbb{R}^d \mapsto \mathbb{R}$ be a concave function with maximizer $x^*$ over the convex set $\mathcal{C}$. Further, assume that $f$ is $G$-strongly concave with respect to $\ell_2$ norm in region $\mathcal{S} \cap \mathcal{C}$, where $\mathcal{S} = \{x : \|x - x^*\|_2 \leq A\}$. If $f(x^*) - f(x) \leq \frac{AG}{2}$ and $c \in \mathcal{C}$, then $x \in \mathcal{S}$.*

*Proof.* By property of strong concavity, we know that, $f(x^*) - f(x) \geq \frac{G}{2} \|x - x^*\|_2$ for any $x \in \mathcal{S} \cap \mathcal{C}$. Now, suppose $x'$ satisfies $f(x^*) - f(x') \leq \frac{AG}{2}$, $x' \in \mathcal{C}$ and $x' \notin \mathcal{S}$. Then, we must have $\|x' - x^*\|_2 > A$.

Let $\gamma \in (0, 1)$ be some number such that $z = \gamma x' + (1 - \gamma) x^*$ lies on the boundary of $\mathcal{S}$. By convexity, we also have $z \in \mathcal{C}$. Then, since $f$ is concave, we have $f(z) \geq \gamma f(x') + (1 - \gamma) f(x^*) > f(x')$, where the second inequality is strict because $f$ is strongly concave in a region around $x^*$. Since $f(x^*) - f(x') \leq \frac{AG}{2}$, $f$ is $G$-strongly concave on $\mathcal{S}$ and $z$ lies on the boundary of $\mathcal{S}$, we have

$$
\frac{AG}{2} = \frac{G}{2} \|z - x^*\|_2 \leq f(x^*) - f(z) < f(x^*) - f(x') \leq \frac{AG}{2}.
$$

This is a contradiction and thus we must have $x' \in \mathcal{S}$. $\qquad \square$

The following lemma quantitatively describes how close $\widetilde{\Lambda}$ and $\Lambda^*$ needs to be to ensure an at most $\epsilon$ multiplicative constraint violation.

**Lemma 12.** *Assume $\|x\|_2 \leq M$ for any $x \in \mathcal{X}$. Let $\Sigma = \mathbb{E}_{X \sim \nu}\left[XX^\top\right] \succ \mathbf{0}$ and $\Lambda^* = \operatorname{argmax}_{\Lambda \succeq \mathbf{0}} \overline{D}(\Lambda)$. Then, for any $\epsilon > 0$, if we have*

$$
\left\| \widetilde{\Lambda} - \Lambda^* \right\|_F \leq \frac{8\mu^2 \lambda_{\min}(\Sigma)}{3M^2 \lambda_{\max}(\Sigma)} \cdot \frac{\epsilon}{1 + \epsilon},
$$

*then it holds that $y^\top \mathbb{E}_{X \sim \nu}\left[P_{\widetilde{\Lambda}}(X) XX^\top\right]^{-1} y \leq (1 + \epsilon) c_\ell^2$ for any $y \in \mathcal{Y}_\ell$.*

*Proof.* Fix some $\epsilon > 0$. First, notice that if we regard $P_\Lambda$ as a function of $q_\Lambda(x) = x^\top \Lambda x - 1$, it then holds that

$$
\|\nabla_\Lambda P_\Lambda(x)\|_2 = \left\| \frac{\mathrm{d}P_\Lambda}{\mathrm{d}q_\Lambda} \nabla_\Lambda q_\Lambda(x) \right\|_2 \leq \left| \frac{\mathrm{d}P_\Lambda}{\mathrm{d}q_\Lambda} \right| \|xx^\top\|_2 \leq \left| \frac{\mathrm{d}P_\Lambda}{\mathrm{d}q_\Lambda} \right| M^2 \leq \frac{M^2}{8\mu},
$$

where we obtain the last inequality by using Lemma 14. Therefore, for any $x \in \mathcal{X}$ and $\widetilde{\Lambda} \succeq \mathbf{0}$, we have $\left| P_{\widetilde{\Lambda}}(x) - P_{\Lambda^*}(x) \right| \leq \frac{M^2}{8\mu} \cdot \left\| \widetilde{\Lambda} - \Lambda^* \right\|_F$ by mean value theorem and Cauchy-Schwartz. inequality.

Therefore, if we have $\left\| \widetilde{\Lambda} - \Lambda^* \right\|_F \leq \delta$, then

$$
\left| P_{\widetilde{\Lambda}}(x) - P_{\Lambda^*}(x) \right| \leq \frac{M^2 \delta}{8\mu} \implies P_{\widetilde{\Lambda}}(x) \geq P_{\Lambda^*}(x) - \frac{M^2 \delta}{8\mu}
$$

$$\implies \mathbb{E}_{X\sim\nu}\left[P_{\widetilde{\Lambda}}(X)XX^\top\right] \succeq \mathbb{E}_{X\sim\nu}\left[P_{\Lambda^*}(X)XX^\top\right] - \frac{M^2\delta}{8\mu}\mathbb{E}_{X\sim\nu}\left[XX^\top\right].$$

By Lemma 13, we know that

$$y^\top\mathbb{E}_{X\sim\nu}\left[P_{\widetilde{\Lambda}}(X)XX^\top\right]^{-1}y \le c_\ell^2(1+\epsilon) \iff \mathbb{E}_{X\sim\nu}\left[P_{\widetilde{\Lambda}}(X)XX^\top\right] \succeq \frac{yy^\top}{(1+\epsilon)c_\ell^2}. \qquad (24)$$

Let $\Sigma^* = \mathbb{E}_{X\sim\nu}\left[P_{\Lambda^*}(X)XX^\top\right]$. Therefore, to guarantee the condition in Eq. (24), it is sufficient to guarantee that $\Sigma^* - \frac{M^2\delta}{8\mu}\Sigma \succeq \frac{yy^\top}{(1+\epsilon)c_\ell^2}$, which is equivalent to

$$w^\top\Sigma^*w - \frac{M^2\delta}{8\mu}w^\top\Sigma w \ge \frac{(w^\top y)^2}{c_\ell^2(1+\epsilon)}, \quad \forall\text{unit vector } w \in \mathbb{R}^d$$

$$\iff \frac{1}{w^\top\Sigma w}\cdot w^\top\left(\Sigma^* - \frac{yy^\top}{(1+\epsilon)c_\ell^2}\right)w \ge \frac{M^2\delta}{8\mu}, \quad \forall\text{unit vector } w \in \mathbb{R}^d.$$

Therefore, it is sufficient to choose $\delta$ such that

$$\frac{M^2\delta}{8\mu} \le \frac{1}{\lambda_{\max}(\Sigma)}\cdot\lambda_{\min}\left(\Sigma^* - \frac{yy^\top}{c_\ell^2(1+\epsilon)}\right) \le \min_{w:\|w\|_2=1}\frac{1}{w^\top\Sigma w}\cdot w^\top\left(\Sigma^* - \frac{yy^\top}{(1+\epsilon)c_\ell^2}\right)w.$$

Since $P_{\Lambda^*}$ satisfies the constraint defined in problem (12), we have $\Sigma^* \succeq \frac{yy^\top}{c_\ell^2}$. Meanwhile, by Lemma 14, we know that $P_{\Lambda^*}(x) \ge \frac{\mu}{3}$ for any $x \in \mathcal{X}$, which means that $\Sigma^* \succeq \frac{\mu}{3}\cdot\Sigma$. That is, for any unit vector $w \in \mathbb{R}^d$, we have

$$w^\top\Sigma^*w \ge \frac{(w^\top y)^2}{c_\ell^2} \quad \text{and} \quad w^\top\Sigma^*w \ge \frac{\mu}{3}\lambda_{\min}(\Sigma),$$

which together implies $w^\top\Sigma^*w \ge \max\left\{\frac{\mu}{3}\cdot\lambda_{\min}(\Sigma), \frac{(w^\top y)^2}{c_\ell^2}\right\}$. Therefore, it holds that

$$w^\top\Sigma w - \frac{(w^\top y)^2}{(1+\epsilon)c_\ell^2} \ge \max\left\{\frac{\mu}{3}\cdot\lambda_{\min}(\Sigma), \frac{(w^\top y)^2}{c_\ell^2}\right\} - \frac{(w^\top y)^2}{(1+\epsilon)c_\ell^2}$$

$$= \max\left\{\frac{\mu}{3}\cdot\lambda_{\min}(\Sigma) - \frac{(w^\top y)^2}{(1+\epsilon)c_\ell^2}, \frac{\epsilon(w^\top y)^2}{(1+\epsilon)c_\ell^2}\right\}$$

$$\ge \frac{\epsilon\mu}{3(1+\epsilon)}\cdot\lambda_{\min}(\Sigma)$$

$$\implies \lambda_{\min}\left(\Sigma^* - \frac{yy^\top}{c_\ell^2(1+\epsilon)}\right) \ge \frac{\epsilon\mu}{3(1+\epsilon)}\cdot\lambda_{\min}(\Sigma).$$

Therefore, to guarantee the condition in Eq. (24), it is sufficient to have

$$\frac{M^2\delta}{8\mu} = \frac{\epsilon\mu\lambda_{\min}(\Sigma)}{3(1+\epsilon)\lambda_{\max}(\Sigma)} \implies \mu = \frac{8\mu^2\lambda_{\min}(\Sigma)}{3M^2\lambda_{\max}(\Sigma)}\cdot\frac{\epsilon}{1+\epsilon},$$

Thus, the proof is complete. $\qquad\square$

The following lemma is a result of standard Schur complement technique.

**Lemma 13.** *If $\mathbb{E}_{X\sim\nu}\left[P(X)XX^\top\right]$ is invertible and $c_\ell > 0$, then*

$$y^\top\mathbb{E}_{X\sim\nu}\left[P(X)XX^\top\right]^{-1}y \le c_\ell^2 \iff \mathbb{E}_{X\sim\nu}\left[P(X)XX^\top\right] \succeq \frac{yy^\top}{c_\ell^2}.$$

*Proof.* For simplicity, let $A = \mathbb{E}_{X\sim\nu}\left[P(X)XX^\top\right] \succ \mathbf{0}$. Then, we consider the block matrix $\begin{bmatrix} A & y \\ y^\top & c_\ell^2 \end{bmatrix} \in \mathbb{R}^{(d+1)\times(d+1)}$. Let $\begin{bmatrix} u & a \end{bmatrix}^\top \in \mathbb{R}^{d+1}$ with $u \in \mathbb{R}^d$ be some vector.

Now, for one direction, suppose $y^\top A^{-1} y \le c_\ell^2$ holds. Consider

$$\begin{bmatrix} u & a \end{bmatrix} \begin{bmatrix} A & y \\ y^\top & c_\ell^2 \end{bmatrix} \begin{bmatrix} u \\ a \end{bmatrix} = u^\top A u + 2a u^\top y + 2 c_\ell^2 a^2 := r(u, a).$$

If we minimize $r(u, a)$ over $u$, which means to treat $a$ as fixed, we can get (by taking gradient and setting it to zero)

$$u^* = -a A^{-1} y \implies r(u^*, a) = a^2 (c_\ell^2 - y^\top A^{-1} y).$$

Since $y^\top A^{-1} y \le c_\ell^2$, we know that $r(u^*, a) \ge 0$, which means $r(u, a) \ge 0$ for any $\begin{bmatrix} u & a \end{bmatrix}^\top \in \mathbb{R}^{d+1}$.

Then, if we minimize $r(u, a)$ over $a$, we can get

$$a^* = -\frac{u^\top y}{c_\ell^2} \implies r(u, a^*) = u^\top A u - \frac{(u^\top y)^2}{c_\ell^2}.$$

Since $r(u, a) \ge 0$ for any $\begin{bmatrix} u & a \end{bmatrix}^\top \in \mathbb{R}^{d+1}$, we know that $u^\top A u - \frac{(u^\top y)^2}{c_\ell^2} \ge 0$ for any $u \in \mathbb{R}^d$. That is, we have $A \succeq \frac{y y^\top}{c_\ell^2}$.

The other direction simply takes the above calculation in a reversed way and thus the proof is complete. □

### C.2.4 Properties of $P_\Lambda$

A visualization of $P_\Lambda$ is given in Figure 2.

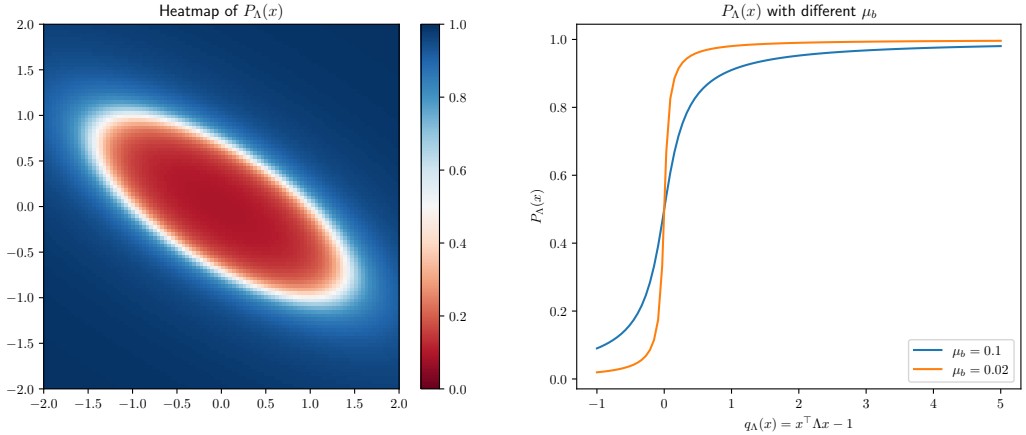

Figure 2: (left) A heatmap of some $P_\Lambda$ when problem dimension is $d = 2$, which shows that $P_\Lambda$ is approximately an 0-1 threshold rule characterized by an ellipsoid. (right) A plot of $P_\Lambda$ as a function of $q_\Lambda(x) = x^\top \Lambda x - 1$, which shows that the change of $P_\Lambda$ near the boundary of ellipsoid is sharper when the barrier weight $\mu$ is smaller.

**Lemma 14.** *The function $P_\Lambda(x)$ defined in (7), if regarding as a function of $q_\Lambda(x) = x^\top \Lambda x - 1 \ge -1$, satisfies*

- $\lim_{q_\Lambda \to 0} P_\Lambda = \frac{1}{2}$ *for any $\mu \in (0, 1)$*

- *When $q_\Lambda = -1$, $P_\Lambda = \frac{1}{2} + \mu - \frac{\sqrt{1 + 4\mu^2}}{2} \ge \frac{\mu}{3}$ and $P_\Lambda - \mu(\log(1 - P_\Lambda) + \log(P_\Lambda)) \le 2\sqrt{\mu}$ for any $\mu \in (0, 1)$.*

- $\frac{\mathrm{d} P_\Lambda}{\mathrm{d} q_\Lambda} = \frac{\mu \sqrt{q_\Lambda^2 + 4\mu^2} - 2\mu^2}{q_\Lambda^2 \sqrt{q_\Lambda^2 + 4\mu^2}}$ *decreases as $q_\Lambda^2$ increases. Further, $\frac{\mathrm{d} P_\Lambda}{\mathrm{d} q_\Lambda} \in [0, \frac{1}{8\mu}]$. Thus, $P_\Lambda$ increases monotonically as $q_\Lambda$ increases and $P_\Lambda(x) \ge \frac{\mu}{3}$ for any $x \in \mathcal{X}$ and $\Lambda \succeq \mathbf{0}$.*

- $\frac{\mathrm{d} P_\Lambda}{\mathrm{d} q_\Lambda}\big|_{q_\Lambda = \pm 1} \ge \frac{\mu}{10}$ *and $\frac{\mathrm{d} P_\Lambda}{\mathrm{d} q_\Lambda} \ge \frac{\mu}{2 q_\Lambda^2}$ when $q_\Lambda^2 \ge 12\mu^2$.*

*Proof.* For simplicity, we will drop the subscript $\Lambda$ and just treat $P$ as a function of $q$. That is, we have

$$P(q) = \frac{1}{2} - \frac{\mu}{q} + \frac{\sqrt{(2\mu - q)^2 + 4\mu q}}{2q}.$$

We prove each bullet point separately.

- Since $P(q)$ also satisfies Eq. (6), which in simpler form is $\frac{\mu}{1-P(q)} - \frac{\mu}{P(q)} = q$, we can easily see that $P(q) = \frac{1}{2}$ satisfies this equation when $q = 0$.

- By direction computation, we can get $P(-1) = \frac{1}{2} + \mu - \frac{\sqrt{1+4\mu^2}}{2}$. To show this is greater than $\frac{\mu}{3}$ for any $\mu \in [0,1]$, consider $\ell(\mu) = P(-1) - \frac{\mu}{3}$. It is easy to check that $\ell(0) = 0$ and $\ell(1) > 0$. Then, since $\ell'(\mu) = \frac{2}{3} - \frac{2\mu}{\sqrt{1+4\mu^4}}$ is initially greater than 0 and then smaller than 0, we know $\ell(\mu)$ first increases and then decreases on $[0,1]$. Thus, $\ell(\mu) \geq 0$ on $[0,1]$ and thus $P(-1) \geq \frac{\mu}{3}$ for any $\mu \in [0,1]$.

  For the second part, define $\tilde{\ell}(\mu) = 2\sqrt{\mu} - P(-1) + \mu\left(\log(1 - P(-1)) + \log(P(-1))\right)$. Then, by utilizing the fact that $P$ satisfies Eq. (6), we can compute its derivative and get $\frac{d\tilde{\ell}}{d\mu} = \frac{1}{\sqrt{\mu}} + \log(1 - P(-1)) + \log(P(-1))$. We can check that on the domain $(0,1)$, we have $\frac{d^2\tilde{\ell}}{d\mu^2} = -\frac{1}{2\mu^{3/2}} + \frac{1}{\mu} - \frac{2}{\sqrt{1+4\mu^2}} \cdot \frac{2\sqrt{\mu(1+4\mu^2)}-4\mu^{3/2}-\sqrt{1+4\mu^2}}{2\mu^{3/2}\sqrt{1+4\mu^2}} \leq 0$ on $(0,1)$, which means that $\frac{d\tilde{\ell}}{d\mu}$ is monotonically decreasing. To see why the second derivative is smaller than 0, we can compute

  $$\left(4\mu^{3/2} + \sqrt{1 + 4\mu^2}\right) - 4\mu\left(1 + 4\mu^2\right) = (1 - 2\mu)^2 + 8\mu^{3/2}\sqrt{1 + 4\mu^2} \geq 0.$$

  Thus, $\frac{d\tilde{\ell}}{d\mu}$ is initially greater than 0 and then smaller than 0 on $(0,1)$. It is easy to verify that $\lim_{\mu \to 0} \tilde{\ell} = 0$ and $\tilde{\ell}(1) > 0$. Therefore, we have $\tilde{\ell}(\mu) \geq 0$ for any $\mu \in (0,1)$.

- We can get $\frac{dP}{dq} = \frac{\mu\sqrt{q^2+4\mu^2}-2\mu^2}{q^2\sqrt{q^2+4\mu^2}}$ by direct computation. To show it is decreasing as $q^2$ increasing, we consider $\tilde{f}(z) = \frac{\mu\sqrt{z+4\mu^2}-2\mu^2}{z\sqrt{z+4\mu^2}}$ and it is sufficient to show that $\frac{d\tilde{f}}{dz} < 0$ for any $z > 0$. Again by direct computation, we have

  $$\frac{d\tilde{f}}{dz} = \frac{\mu\left(8\mu^3 + 3\mu z - (z + 4\mu^2)^{3/2}\right)}{z^2(z + 4\mu^2)^{3/2}}.$$

  By direct computation, We can show that $(z + 4\mu^2)^3 - (8\mu^3 + 3\mu z)^2 = z^3 + 3z^2\mu^2 > 0$ for any $z > 0$ and $\mu \in [0,1]$. Thus, $\frac{d\tilde{f}}{dz} < 0$ and thus $\frac{dP}{dq}$ is decreasing as $q^2$ increases.

  It is obvious that $\frac{dP}{dq} \geq 0$ for any $q^2 \geq 0$ and $\mu \in [0,1]$ since we always have $\mu\sqrt{q^2 + 4\mu^2} \geq 2\mu^2$. Thus, the maximum value could potentially happen is when $q^2 \to 0$, which can be evaluated by using L'Hospital's rule. A direct computation gives us $\lim_{q^2 \to 0} \frac{dP}{dq} = \frac{1}{8\mu}$. Thus, we can conclude that $\frac{dP}{dq} \in \left[0, \frac{1}{8\mu}\right]$. Therefore, $P$ increases monotonically as $q$ increases, which implies that $P_\Lambda(x) \geq \frac{\mu}{3}$ for any $x \in \mathcal{X}$ and $\Lambda$.

- By direct computation, we have $\frac{dP_\Lambda}{dq_\Lambda}|_{q_\Lambda = \pm 1} = \mu\left(1 - \frac{2\mu}{\sqrt{1+4\mu^2}}\right) \geq \mu\left(1 - \frac{2}{\sqrt{5}}\right) \geq \frac{\mu}{10}$ for any $\mu \in [0,1]$. The reason is that we can easily see $\frac{2\mu}{\sqrt{1+4\mu^2}}$ is increasing in $\mu$.

  Finally, notice that when $2\mu \leq \frac{1}{2}\sqrt{q^2 + 4\mu^2}$, which is equivalent to $q^2 \geq 12\mu^2$, we have

  $$\frac{dP}{dq} = \frac{\mu\sqrt{q^2 + 4\mu^2} - 2\mu^2}{q^2\sqrt{q^2 + 4\mu^2}} \geq \frac{\mu\sqrt{q^2 + 4\mu^2} - \frac{\mu}{2}\sqrt{q^2 + 4\mu^2}}{q^2\sqrt{q^2 + 4\mu^2}} = \frac{\mu}{2q^2}.$$

Thus, the proof is complete. □

## C.3 An Alternative Approach to OPTIMIZEDESIGN

Based on the analysis in Section C.1, we know that maximizing $\overline{D}(\cdot)$ is equivalent to maximizing $D(\cdot)$. Therefore, as an alternative to Algorithm 2, which maximizes $D(\cdot)$ through stochastic gradient ascent, it is natural to have an algorithm that directly maximizes $\overline{D}(\cdot)$. Here, we will consider subgradient ascent.

Recall that $\overline{D} : \mathbb{S}_+^d \mapsto \mathbb{R}$ is defined as

$$\overline{D}(\Lambda) = \mathbb{E}_{X \sim \nu} \left[ P_\Lambda(X) - \mu \left( \log(1 - P_\Lambda(X)) + \log(P_\Lambda(X)) \right) - P_\Lambda(X) X^\top \Lambda X \right] + \frac{1}{c_\ell^2} \cdot f(\Lambda),$$

where $f(\Lambda)$ is defined in problem (10). The subgradient of $\overline{D}(\Lambda)$ is

$$\partial \overline{D}(\Lambda) = \mathbb{E}_{X \sim \nu} \left[ \left( 1 + \frac{\mu}{1 - P_\Lambda(x)} - \frac{\mu}{P_\Lambda(X)} - X^\top \Lambda X \right) \nabla P_\Lambda(X) - P_\Lambda(X) X X^\top \right] + \frac{\partial f(\Lambda)}{c_\ell^2}$$
$$\text{(The first term is differentiable)}$$

$$= \frac{\partial f(\Lambda)}{c_\ell^2} - \mathbb{E}_{X \sim \nu} \left[ P_\Lambda(X) X X^\top \right]. \qquad \text{(Since } P_\Lambda(X) \text{ solves Eq. (6))}$$

Therefore, to run subgradient ascent, we only need to find an element in $\partial f(\Lambda)$, which can be obtained by solving the following optimization problem as claimed by Lemma 15.

$$\begin{aligned} \min_\Gamma \quad & \langle \Gamma, \Lambda \rangle \\ \text{subject to} \quad & \Gamma \succeq yy^\top, \quad \forall y \in \mathcal{Y}_\ell, \\ & \Gamma \preceq 2 \sum_{y \in \mathcal{Y}_\ell} yy^\top. \end{aligned} \qquad (25)$$

As a result, we have Algorithm 3 as an alternative to solve OPTIMIZEDESIGN. Compared to Algorithm 2, which needs to maintain $|\mathcal{Y}_\ell| d^2$ number of objective variables, Algorithm 3 only has $d^2$ variables. However, each iteration of Algorithm 3 is computationally more intensive since finding a subgradient needs to solve the problem (25).

---

**Algorithm 3** Projected Stochastic Subgradient Ascent to Solve OPTIMIZEDESIGN

---

1: **Input:** Number of iterations $K$; number of samples $u$; barrier weight $\mu_b \in (0,1)$
2: Initialize $\hat{\Lambda}^{(0)} = \mathbf{0}$
3: **for** $k = 0, 1, 2, \ldots, K-1$ **do**
4:     Sample $x_k \sim \nu$
5:     Solve problem (25) with current $\hat{\Lambda}^{(k)}$ to get $\Gamma^{(k)}$
6:     Set $g_k = \frac{\Gamma^{(k)}}{c_\ell^2} - P_{\hat{\Lambda}^{(k)}}(x_k) x_k x_k^\top$
7:     Set $\hat{\Lambda}^{(k+1)} \leftarrow \hat{\Lambda}^{(k)} + \eta_k g_k$, where $\eta_k = \frac{1}{\sqrt{2 \sum_{s=1}^k \|g_s\|_2^2}}$
8:     Update $\hat{\Lambda}^{(k+1)} \leftarrow \Pi_{\mathbb{S}_+^d} (\hat{\Lambda}^{(k+1)})$, a projection to the set of $d \times d$ PSD matrices
9: **end for**
10: Let $\hat{\Lambda} = \frac{1}{K} \sum_{k=1}^K \hat{\Lambda}^{(k)}$
11: Find $s^* \leftarrow \operatorname{argmax}_{s \in [0,1]} \overline{D}_E(s \cdot \hat{\Lambda})$, where $\overline{D}_E$ is the empirical version of $\overline{D}$, evaluated using $u$ i.i.d. samples
12: **return** $\widetilde{\Lambda} = s^* \cdot \hat{\Lambda}$

---

A result similar to Theorem 5 can also be obtained for Algorithm 3, which is given in Theorem 6. The bounds are almost identical except that the old lower bound for $K$ depends on $|\mathcal{Y}_\ell|^3$ while the new one depends on $|\mathcal{Y}_\ell|$. Steps identical to the proof of Theorem 5 will be skipped in the proof of Theorem 6.

**Theorem 6.** *Let $\Lambda^* \in \operatorname{argmax}_{\Lambda \succeq \mathbf{0}} \overline{D}(\Lambda)$ and take other settings the same as that in Theorem 5.*

*Then, $\Lambda^*$ is unique. Further, for any $\epsilon > 0$ and $\delta > 0$, suppose it holds that*

$$\mu \le \min\left\{ \sqrt{\frac{3\kappa(\Sigma) \left\|\Lambda^*\right\|_F M^2}{8} \cdot \frac{1+\epsilon}{\epsilon}}, \frac{4}{9} \left\|\Lambda^*\right\|_F^2 M^4, \frac{1}{2\sqrt{3}} \right\}$$

$$K \ge \frac{288\kappa(\Sigma)^2 \left\|\Lambda^*\right\|_F^4 M^4 (M^4 + 4\left|\mathcal{Y}_\ell\right| C_\ell^2) \cdot \left(2\left\|\Lambda^*\right\|_F M^2 + 1\right)^4 \log(6/\delta)}{\omega^2 \mu^6} \cdot \left(\frac{1+\epsilon}{\epsilon}\right)^2$$

$$u \ge \frac{576\kappa(\Sigma)^2 \left\|\Lambda^*\right\|_F^2 M^8 \cdot \left(2\left\|\Lambda^*\right\|_F M^2 + 1\right)^4 \log(6/\delta)}{\omega^2 \mu^6} \cdot \left(\frac{1+\epsilon}{\epsilon}\right)^2.$$

*Then, with probability at least $1 - \delta$, Algorithm 2 will output $\widetilde{\Lambda}$ that satisfies*

- $y^\top \mathbb{E}_{X \sim \nu} \left[ P_{\widetilde{\Lambda}}(X) XX^\top \right]^{-1} y \le (1+\epsilon)c_\ell^2, \quad \forall y \in \mathcal{Y}_\ell.$

- $\mathbb{E}_{X \sim \nu} \left[ P_{\widetilde{\Lambda}}(X) \right] \le \mathbb{E}_{X \sim \nu} \left[ \widetilde{P}(X) \right] + 4\sqrt{\mu}$, *where $\widetilde{P}$ is the optimal solution to problem* (20).

*Proof.* **First Bullet Point.** Similar to the proof of Theorem 5, let $\hat{\Lambda}$ be the parameter obtained by Algorithm 3 just before the re-scaling step (line 11). Then, by Theorem 3.13 of [21], with probability at least $1 - \frac{\delta}{3}$, it holds that

$$\overline{D}(\Lambda^*) - \overline{D}(\hat{\Lambda}) \le \frac{\text{Reg}(K) + 2\sqrt{2K\log(6/\delta)}}{K},$$

where $\text{Reg}(K)$ is the regret of running projected stochastic subgradient ascent for $K$ steps with $\eta_k$ specified in Algorithm 3. Meanwhile, by Theorem 4.14 of [21] also, we have $\text{Reg}(K) = \sqrt{2}B^2 \sqrt{\sum_{k=1}^K \|g_k\|_2^2}$, where $B = \|\Lambda^*\|_F$. Since $g_k = \frac{\Gamma^{(k)}}{c_\ell^2} - P_{\hat{\Lambda}^{(k)}}(x_k) x_k x_k^\top$ and $\left\|\Gamma^{(k)}\right\|_F \le 2\left\|\sum_{y \in \mathcal{Y}_\ell} yy^\top\right\|_F$, we can easily get $\|g_k\|_2^2 \le 2M^4 + \frac{8}{c_\ell^2} \sum_{y \in \mathcal{Y}_\ell} \|y\|_2^4 = 2M^4 + 8\left|\mathcal{Y}_\ell\right| C_\ell^2$. Thus, we have

$$\text{Reg}(K) \le 2\left\|\Lambda^*\right\|_F^2 \sqrt{M^4 + 4\left|\mathcal{Y}_\ell\right| C_\ell^2} \cdot \sqrt{K} := C_{\text{Reg}} \sqrt{K} \tag{26}$$

$$\implies \overline{D}(\Lambda^*) - \overline{D}(\hat{\Lambda}) \le \frac{C_{\text{Reg}} + 2\sqrt{2\log(6/\delta)}}{\sqrt{K}}, \tag{27}$$

We now consider the effect of using $u$ i.i.d. samples in the re-scaling step. Since re-scaling always increases the function value, we must have $\overline{D}_E(\hat{\Lambda}) \le \overline{D}_E(\widetilde{\Lambda})$.

Then, after **exactly the same** steps of analysis, we can get the following same lower bound for $K$,

$$K \ge \left( \frac{3\kappa(\Sigma)M^2 \left( C_{\text{Reg}} + 2\sqrt{2\log(6/\delta)} \right)}{2G\mu^2} \cdot \frac{1+\epsilon}{\epsilon} \right)^2, \tag{28}$$

with a different value of $C_{\text{Reg}}$.

**Second Bullet Point.** We then prove the upper bound for primal objective value $\mathbb{E}_{X \sim \nu} \left[ P_{\widetilde{\Lambda}}(X) \right]$, which explains the reason why an extra re-scaling step is needed. Let $\hat{\mathbf{\Lambda}} = (\hat{\Lambda}_y)_{y \in \mathcal{Y}_\ell}$ be a set of PSD matrices that solves problem (10) with parameter $\hat{\Lambda}$ and $\widetilde{\mathbf{\Lambda}} = s^* \cdot \hat{\mathbf{\Lambda}}$, where $s^* = \text{argmax}_{s \in [0,1]} \overline{D}_E(s \cdot \hat{\Lambda})$. Since the constraint in problem (10) requires $\sum_{y \in \mathcal{Y}_\ell} \hat{\Lambda}_y = \hat{\Lambda}$, we have $\sum_{y \in \mathcal{Y}_\ell} \widetilde{\Lambda}_y = \widetilde{\Lambda}$, which is the output of Algorithm 3.

Define $g(s) = D_E(s \cdot \widetilde{\mathbf{\Lambda}})$. By construction, we know that $g(s)$ is maximized at $s = 1$ because $\overline{D}_E(s \cdot \hat{\Lambda}) = D_E(s \cdot \hat{\Lambda})$ for any $s \ge 0$ as shown in Lemma 10, which means that $s^* = \text{argmax}_{s \in [0,1]} D_E(s \cdot \hat{\Lambda})$. Therefore, we have $g'(1) \ge 0$, which in turn gives us

$$g'(1) = \frac{1}{c_\ell^2} \sum_{y \in \mathcal{Y}_\ell} y^\top \widetilde{\Lambda}_y y - \frac{1}{u} \sum_{i=1}^u P_{\widetilde{\Lambda}}(x_i) x_i^\top \widetilde{\Lambda} x_i \ge 0.$$

Then, after **exactly the same** steps of analysis, we can get $\mathbb{E}_{X\sim\nu}\left[P_{\widetilde{\Lambda}}(X)\right] \leq \mathbb{E}_{X\sim\nu}\left[\widetilde{P}(X)\right] + 4\sqrt{\mu}$, where $\widetilde{P}$ is the optimal solution of the problem (20). $\qquad\square$

### C.3.1 Technical Lemmas

**Lemma 15.** *The optimal value of the optimization problem* (25) *with parameter* $\Lambda \succeq \mathbf{0}$ *is equal to* $f(\Lambda)$*. Further, let* $\Gamma^*(\Lambda)$ *be an optimal solution to* (25)*. Then, it holds that* $\Gamma^*(\Lambda) \in \partial f(\Lambda)$ *and* $\|\Gamma^*(\Lambda)\| \leq 2\left\|\sum_{y\in\mathcal{Y}_\ell} yy^\top\right\|_F$.

*Proof.* Alternatively, we first consider the following optimization problem.

$$
\begin{aligned}
\max_{\Lambda_y, \Sigma} \quad & \sum_{y\in\mathcal{Y}_\ell} y^\top (\Lambda_y - 2\Sigma)\, y \\
\text{subject to} \quad & \Lambda = \sum_{y\in\mathcal{Y}_\ell} \Lambda_y - \Sigma, \\
& \Sigma \succeq \mathbf{0}, \Lambda_y \succeq \mathbf{0}, \quad \forall y \in \mathcal{Y}_\ell.
\end{aligned}
\tag{29}
$$

Since $y^\top \Sigma y \geq 0$ for any $y \in \mathcal{Y}_\ell$ and $\Sigma \succeq \mathbf{0}$, it is clear that problem (29) has the same optimal value as problem (10). Then, let $\Gamma \in \mathbb{R}^{d\times d}$ be the dual variable for the equality constraint in problem (29). We can have its dual problem to be

$$
\min_\Gamma \max_{\substack{\Lambda_y \succeq \mathbf{0}, \forall y\in\mathcal{Y}_\ell, \\ \Sigma\succeq\mathbf{0}}} \sum_{y\in\mathcal{Y}_\ell} \langle yy^\top, \Lambda_y - 2\Sigma\rangle + \left\langle \Gamma, \Lambda + \Sigma - \sum_{y\in\mathcal{Y}_\ell} \Lambda_y \right\rangle
$$

$$
\implies \min_\Gamma \max_{\substack{\Lambda_y \succeq \mathbf{0}, \forall y\in\mathcal{Y}_\ell, \\ \Sigma\succeq\mathbf{0}}} \langle \Gamma, \Lambda\rangle + \left\langle \Sigma, \Gamma - 2\sum_{y\in\mathcal{Y}_\ell} yy^\top \right\rangle + \sum_{y\in\mathcal{Y}_\ell} \langle \Lambda_y, yy^\top - \Gamma\rangle.
$$

In order for the above optimization problem to have finite value, we must have $\Gamma \preceq 2\sum_{y\in\mathcal{Y}_\ell} yy^\top$ and $\Gamma \succeq yy^\top$ for any $y \in \mathcal{Y}_\ell$. Therefore, we obtain the following dual problem.

$$
\begin{aligned}
\min_\Gamma \quad & \langle \Gamma, \Lambda\rangle \\
\text{subject to} \quad & \Gamma \succeq yy^\top, \quad \forall y \in \mathcal{Y}_\ell, \\
& \Gamma \preceq 2\sum_{y\in\mathcal{Y}_\ell} yy^\top.
\end{aligned}
$$

This is exactly the problem (25). Then, we can notice the Slater's condition is clearly satisfied by problem (25), which means the strong duality holds. Therefore, problem (25) has the same optimal value as (29), which is the same as (10).

Since $f(\Lambda)$ is concave in $\Lambda$ as shown in Lemma 6, to show that $\Gamma^*(\Lambda) \in \partial f(\Lambda)$, consider arbitrary $\Lambda, \Lambda' \succeq \mathbf{0}$. Then, we have

$$
f(\Lambda) + \langle \Gamma^*(\Lambda), \Lambda' - \Lambda\rangle = \langle \Gamma^*(\Lambda), \Lambda\rangle + \langle \Gamma^*(\Lambda), \Lambda' - \Lambda\rangle = \langle \Gamma^*(\Lambda), \Lambda'\rangle \geq f(\Lambda').
$$

The first equality holds because the optimal value of problem (25) is $f(\Lambda)$ as just shown above. The last inequality holds because $\Gamma^*(\Lambda)$ is a feasible solution to the problem (25) with parameter $\Lambda'$. Therefore, we have $\Gamma^*(\Lambda) \in \partial f(\Lambda)$.

Finally, since the constraint of problem (25) requires $\Gamma^*(\Lambda) \preceq 2\sum_{y\in\mathcal{Y}_\ell} yy^\top$, we can obtain $\|\Gamma^*(\Lambda)\|_F \leq 2\left\|\sum_{y\in\mathcal{Y}_\ell} yy^\top\right\|_F$ as a direct consequence of Lemma 16. $\qquad\square$

**Lemma 16.** *For* $A, B \in \mathbb{S}^{d\times d}$*, if* $A \succeq B \succeq \mathbf{0}$*, then* $\|A\|_F \geq \|B\|_F$*.*

*Proof.* Let $\lambda_1, \ldots, \lambda_d$ and $\gamma_1, \ldots, \gamma_d$ be eigenvalues of $A$ and $B$, respectively. Let $v_1, \ldots, v_d$ be a set of orthogonal unit eigenvectors of matrix $A$. Then, we have

$$
\|A\|_F = \sqrt{\operatorname{tr}(AA)} = \sqrt{\operatorname{tr}\left(\left(\sum_{i=1}^d \lambda_i v_i v_i^\top\right)\left(\sum_{i=1}^d \lambda_i v_i v_i^\top\right)\right)} = \sqrt{\sum_{i=1}^d \lambda_i^2}.
$$

Similarly, we have $\|B\|_F = \sqrt{\sum_{i=1}^d \gamma_i^2}$. By Corollary 7.7.4 in [14], since $A \succeq B \succeq \mathbf{0}$, we know that $\lambda_i \geq \gamma_i \geq 0$ for each $i$. Therefore, we have $\|A\|_F \geq \|B\|_F$. $\qquad\square$

# D   Selective Sampling Algorithm for Unknown Distribution $\nu$

## D.1   Statement and proof of Theorem 7

Consider now the case where we do not know $\nu$ exactly, and are returned $(\widehat{P}_\ell, \widehat{\Sigma}_{\widehat{P}_\ell})$ that only approximate their ideals. Algorithm 1 can still be employed to solve this case where $\nu$ is unknown, but at the cost of sampling some historical data. Note that compared to the case where $\nu$ is know, it assumes the knowledge of an upper bound on $\sup_{x\in\text{support}(\nu)}\|x\|$. It also relies on a multiplicative factor change in the constraint of the optimization problem, in order to account for the possible constraint violation of the output of the subroutine. The last difference is the use of an approximation of the covariance matrix to compute the estimator. The covariance matrix is empirically approximated by injecting additional unlabeled samples (historical data). With that, although we do not know $\nu$ but we can approximate the relevant quantities, such as the covariance matrix $\mathbb{E}_{X\sim\nu}[XX^\top]$.

Let us detail the properties of the implementation of $\widehat{P}_\ell, \widehat{\Sigma}_{\widehat{P}_\ell} \leftarrow \text{OPTIMIZEDESIGN}(\mathcal{Z}_\ell, 2^{-\ell}, \tau)$ we use at each round $\ell$.

First, $\widehat{P}_\ell$ has the properties described in Theorem 4 (by using Algorithm 2). More explicitly, let $\epsilon_\ell := 2^{-\ell}$, $B < \infty$ such that $\max_{x\in\mathcal{X}}|\langle x, \theta_*\rangle| \le B$, and $\sigma < \infty$ such that $\mathbb{E}[(y_s - \langle\theta_*, x_s\rangle)^2|x_s] \le \sigma^2$. If

$$\beta_{\delta,\ell} := 4(1+\varepsilon)^2 \left(4\sqrt{B^2 + \sigma^2} + 1\right)^2 \log(4\ell^2|\mathcal{Z}|^2/\delta)$$

then $\widehat{P}_\ell$ is such that

- $\max_{z,z'\in\mathcal{Z}_\ell} \dfrac{\|z-z'\|^2_{\mathbb{E}_{X\sim\nu}[\tau\widehat{P}_\ell(X)XX^\top]^{-1}}}{\epsilon_\ell^2}\beta_{\delta,\ell} \le 1 + \varepsilon$.

- $\mathbb{E}_{X\sim\nu}\left[\widehat{P}_\ell(X)\right] \le \mathbb{E}_{X\sim\nu}\left[\widetilde{P}_\ell(X)\right] + 4\sqrt{\mu_b}$, where $\widetilde{P}_\ell$ is the optimal solution to problem (30).

$$\begin{aligned} \min_P \quad & \mathbb{E}_{X\sim\nu}\left[P(X)\right] \\ \text{subject to} \quad & \max_{z,z'\in\mathcal{Z}_\ell} \frac{\|z-z'\|^2_{\mathbb{E}_{X\sim\nu}[\tau P(X)XX^\top]^{-1}}}{\epsilon_\ell^2}\beta_{\delta,\ell} \le 1, \\ & 0 \le P(x) \le 1 - \mu_b, \quad \forall x \in \mathcal{X}. \end{aligned} \qquad (30)$$

where $\mu_b \ge 0$. The quantity $\mathbb{E}_{X\sim\nu}\left[\widetilde{P}_\ell(X)\right]$ that uses $\mu_b > 0$ is easily related to the value when $\mu_b = 0$ through a simple scaling factor of $\frac{1}{1-\mu_b}$ (see proof below).

$\widehat{\Sigma}_{\widehat{P}_\ell}$ is the empirical covariance matrix of $\Sigma_{\widehat{P}_\ell} := \mathbb{E}_{X\sim\nu}[\widehat{P}_\ell(X)XX^\top]$ using historical data and is such that

$$(1-\gamma)\Sigma_{\widehat{P}_\ell} \preceq \widehat{\Sigma}_{\widehat{P}_\ell} \preceq (1+\gamma)\Sigma_{\widehat{P}_\ell}$$

where $\gamma \ge 0$.

Again, while we think of historical data as independent data collected offline before the start of the game, in practice this historical data could just come from previous rounds (which is not technically correct since its use may introduce some dependencies).

**Theorem 7** (Upper bound). *Fix any $\delta \in (0,1)$. Let $\Delta = \min_{z\in\mathcal{Z}\backslash z_*}\langle z_* - z, \theta_*\rangle$ and set*

$$\beta_\delta = 256(1+\varepsilon)^2 \left(4\sqrt{B^2+\sigma^2}+1\right)^2 \log(4\log_2^2(\tfrac{4}{\Delta})|\mathcal{Z}|^2/\delta).$$

*For any $\tau \ge \rho(\nu)\beta_\delta$ there exists a $\delta$-PAC selective sampling algorithm that collects $\mathcal{T}$ historical data before the start of the game, observes $\mathcal{U}$ unlabeled examples, and requests just $\mathcal{L}$ labels that satisfies*

- $\mathcal{U} \le \log_2(\tfrac{4}{\Delta})\tau$,

- $\mathcal{L} \le \frac{1}{1-\mu_b}\min_{\lambda\in\triangle_{\mathcal{X}}}\rho(\lambda)\beta_\delta + \frac{5\tau}{1-\mu_b}\sqrt{\mu_b} \quad$ *subject to* $\quad \tau \ge \|\lambda/\nu\|_\infty\rho(\lambda)\beta_\delta$, *and*

- $\mathcal{T} \le \log_2(\tfrac{4}{\Delta})(K + u + \kappa_\delta)$

*with probability at least $1 - \delta$.*

*Here, the sample complexity for estimating the covariance matrix is bounded by $\kappa_\delta = \lceil 2K_{\psi_2}^2(\sqrt{d\ln 9/c_1} + \sqrt{\frac{\log(2/\delta)}{c_1}})\max\{1, 20\|\theta_*\|_{\mathbb{E}_{X\sim\nu}[XX^\top]}\}\rceil$ (where the sub-gaussian norm $K_{\psi_2} = \max_{s,P}\|\sqrt{P(\widetilde{x}_s)}\Sigma_P^{-1/2}\widetilde{x}_s\|_{\psi_2}$ ), and the contributions from the optimization problem to compute $\{\widehat{P}_\ell\}_\ell$ are*

$$K = \widetilde{O}\left(\frac{|\mathcal{Z}|^6\,\kappa(\Sigma)^2\,\|\Lambda^*\|_2^8\,M^{16}}{\omega^2\mu_b^6}\right)\cdot\left(\frac{1+\epsilon}{\epsilon}\right)^2, \quad u = \widetilde{O}\left(\frac{\kappa(\Sigma)^2\,\|\Lambda^*\|_2^6\,M^{16}}{\omega^2\mu_b^6}\right)\cdot\left(\frac{1+\epsilon}{\epsilon}\right)^2,$$

Naturally, we have a trade-off on the subroutine tolerance $\mu_b$. In order to get a better solution of the optimization over the selection rule $P$ (and thus get a smaller $\sum_{t=(\ell-1)\tau+1}^{\ell\tau}P(x_t)$ term), the subroutine needs more unlabeled samples. However, it suffices to take $\mu_b = \frac{1}{\tau^2}$ to make $\mathcal{U}$, and $\mathcal{L}$ roughly match those of the case when $\nu$ was known.

The proof of this theorem is established through several results, which we provide in Section D.2.

### D.2 Lemmas for the correctness

We first state here the correctness of Algorithm 1 in the case where $\nu$ is unknown.

**Lemma 17.** *With probability at least $1 - \delta$ we have for all stages $\ell \in \mathbb{N}$, we have that $z_* \in \mathcal{Z}_\ell$ and $\max_{z\in\mathcal{Z}_\ell}\langle z_* - z, \theta_*\rangle \leq 4\epsilon_\ell$.*

The proof of the correctness lemma is established though several lemmas. First we provide Lemma 18 guaranteeing concentration of empirical covariance matrices, which is obtained by sampling $\kappa$ additional measurements. Then we show in Proposition 3 that the RIPS estimator does not suffer from using that empirical covariance matrix.

**Lemma 18.** *For any $P : \mathcal{X} \to [0,1]$, let $\Sigma_P = \mathbb{E}_{X\sim\nu}[P(X)XX^\top]$, $\widehat{\Sigma}_P = \frac{1}{\kappa}\sum_{s=1}^{\kappa}P(\widetilde{x}_s)\widetilde{x}_s\widetilde{x}_s^\top$. Define $K_{\psi_2} = \max_s\|\sqrt{P(\widetilde{x}_i)}\Sigma_P^{-1/2}\widetilde{x}_s\|_{\psi_2}$. With probability at least $1 - 2\exp(-c_1t^2/K_{\psi_2}^4)$ holds*

$$(1-c)x^\top\Sigma_P x \leq x^\top\widehat{\Sigma}_P x \leq (1+c)x^\top\Sigma_P x$$

*where $c = \max\left\{\frac{C\sqrt{d}+t}{\sqrt{\kappa}}, \left(\frac{C\sqrt{d}+t}{\sqrt{\kappa}}\right)^2\right\}$, $C = K_{\psi_2}^2\sqrt{\ln 9/c_1}$ and $c_1$ is an absolute constant.*

Consequently for $\kappa \geq c_\delta := K_{\psi_2}^2(\sqrt{d\ln 9/c_1} + \sqrt{\frac{\log(2/\delta)}{c_1}})$, holds with probability at least $1 - \delta$

$$\left(1 - \frac{c_\delta}{\sqrt{\kappa}}\right)x^\top\Sigma_P x \leq x^\top\widehat{\Sigma}_P x \leq \left(1 + \frac{c_\delta}{\sqrt{\kappa}}\right)x^\top\Sigma_P x.$$

*Proof.* Let $A \in \mathbb{R}^{\kappa\times d}$ whose rows $A_i$ are independent sub-gaussian isotropic random vectors in $R^d$ and define $K_{\psi_2} = \max_i\|A_i\|_{\psi_2}$. We can apply Theorem 5.39 of [25] on $A$ to have that with probability at least $1 - 2\exp(-c_1t^2/K_{\psi_2}^4)$ holds

$$1 - \frac{C\sqrt{d}+t}{\sqrt{\kappa}} \leq \sigma_{\min}(A) \leq \sigma_{\max}(A) \leq 1 + \frac{C\sqrt{d}+t}{\sqrt{\kappa}},$$

where $C = K_{\psi_2}^2\sqrt{\ln 9/c_1}$ and $c_1$ is an absolute constant.

With Lemma 5.36 of [25], this implies that with probability at least $1 - 2\exp(-c_0t^2)$ holds

$$\|A^\top A - I\| \leq \max\left\{\frac{C\sqrt{d}+t}{\sqrt{\kappa}}, \left(\frac{C\sqrt{d}+t}{\sqrt{\kappa}}\right)^2\right\} =: c \tag{31}$$

Recall $\Sigma_P = \mathbb{E}_{X \sim \nu}[P(X)XX^\top]$, so $Y = \sqrt{P(X)}\Sigma_P^{-1/2}X$ satisfies $\mathbb{E}[YY^\top] = \mathbb{E}[\Sigma_P^{-1/2}P(X)XX^\top\Sigma_P^{-1/2}] = \Sigma_P^{-1/2}\Sigma_P\Sigma_P^{-1/2} = I$. So we can apply (31) to get $\|\Sigma_P^{-1/2}\widehat{\Sigma}_P\Sigma_P^{-1/2} - I\| \leq c$. Thus for any $y \in \mathbb{R}^d$,

$$1 - c \leq \frac{y^\top}{\|y\|}\Sigma_P^{-1/2}\widehat{\Sigma}_P\Sigma_P^{-1/2}\frac{y}{\|y\|} \leq 1 + c$$

so setting $y = \Sigma_P^{1/2}x$

$$(1 - c)x^\top\Sigma_P x \leq x^\top\widehat{\Sigma}_P x \leq (1 + c)x^\top\Sigma_P x.$$

Also, the sub-gaussian bound becomes $K_{\psi_2} = \max_i \|\sqrt{P(\widetilde{x}_i)}\Sigma_P^{-1/2}\widetilde{x}_i\|_{\psi_2}$. $\qquad\square$

**Proposition 3** (RIPS guarantees on empirical covariance matrix). *Let $x_1, \ldots, x_n$ and $\widetilde{x}_1, \ldots, \widetilde{x}_\kappa$ be drawn IID from a distribution $\nu$. For $s = 1, \ldots, n$, assume that $|\langle\theta, x_s\rangle| \leq B$ and $\mathbb{E}[|\langle\theta, x_s\rangle - y_s|^2] \leq \sigma_{noise}^2$. For $s = 1, \ldots, \kappa$, assume that $\mathbb{E}[|\langle\theta, x_s\rangle - y_s|^2] \leq \sigma_{noise}^2$. Let $P \in [0, 1]$ be arbitrary and let $Q_s(x_s) \sim Bernoulli(P)$ independently for all $s \in [n]$. Let $\Sigma_P = \mathbb{E}_{X \sim \nu}[P(X)XX^\top]$ and $\widehat{\Sigma}_P = \frac{1}{\kappa}\sum_{s=1}^\kappa P(\widetilde{x}_s)\widetilde{x}_s\widetilde{x}_s^\top$. Assume that $\Sigma_P$ is invertible and that there exists $\gamma \geq 0$ such that $(1 - \gamma)\Sigma_P \preceq \widehat{\Sigma}_P \preceq (1 + \gamma)\Sigma_P$. For a given finite set $\mathcal{V} \subset \mathbb{R}^d$ define*

$$w_v = \mathrm{Catoni}(\{\langle v, \widehat{\Sigma}_P^{-1}Q_s(x_s)x_sy_s\rangle\}_{s=1}^n),$$

*If $\widehat{\theta} = \arg\min_\theta \max_v \frac{|w_v - \langle\theta, v\rangle|}{\|v\|_{\widehat{\Sigma}_P^{-1}}}$ and $n \geq 4\log(2|\mathcal{V}|/\delta)$, then with probability at least $1 - \delta$, it holds that*

$$|\langle v, \widehat{\theta} - \theta\rangle| \leq 4\left(\sqrt{\frac{B^2 + \sigma^2}{(1 - \gamma)^2}} + \sqrt{n\gamma}\|\theta_*\|_{\mathbb{E}_{X \sim \nu}[XX^\top]}\right)\|v\|_{\mathbb{E}_{X \sim \nu}[nP(X)XX^\top]^{-1}}\sqrt{\log(2|\mathcal{V}|/\delta)}$$

We first state an intermediate matrix lemma before the proof of Proposition 3.

**Lemma 19.** *Assume that $\Sigma_P$ is invertible and that there exists $\gamma \in [0, 1/2]$ such that $(1 - \gamma)\Sigma_P \preceq \widehat{\Sigma}_P \preceq (1 + \gamma)\Sigma_P$. Then for any $v \in \mathcal{V}$*

$$\|v\|_{\widehat{\Sigma}_P^{-1}\Sigma_P\widehat{\Sigma}_P^{-1}}^2 \leq \frac{1}{(1 - \gamma)^2}\|v\|_{\Sigma_P^{-1}}^2.$$

*and*

$$\|v\|_{(I - \Sigma_P^{1/2}\widehat{\Sigma}_P^{-1}\Sigma_P^{1/2})^2} \leq \sqrt{1 - \frac{2}{1 + \gamma} + \frac{1}{(1 - \gamma)^2}}\|v\|_2 \leq \sqrt{10\gamma}\|v\|_2.$$

*Proof.* We know that taking the inverse of two ordered positive definite matrices will flip the order, so here

$$\frac{1}{(1 + \gamma)}\Sigma_P^{-1} \preceq \widehat{\Sigma}_P^{-1} \preceq \frac{1}{(1 - \gamma)}\Sigma_P^{-1}.$$

$(1 - \gamma)\Sigma_P \preceq \widehat{\Sigma}_P$ implies that for all $u \in \mathbb{R}^d$ holds $u^\top\Sigma_P u \leq 1/(1 - \gamma)u^\top\widehat{\Sigma}_P u$. So taking $u = \widehat{\Sigma}_P^{-1}v$, we get $v^\top\widehat{\Sigma}_P^{-1}\Sigma_P\widehat{\Sigma}_P^{-1}v \leq 1/(1 - \gamma)v^\top\widehat{\Sigma}_P^{-1}v$. Conclusion

$$v^\top\widehat{\Sigma}_P^{-1}\Sigma_P\widehat{\Sigma}_P^{-1}v = \frac{1}{1 - \gamma}v^\top\widehat{\Sigma}_P^{-1}v \leq \frac{1}{(1 - \gamma)^2}v^\top\Sigma_P^{-1}v$$

hence the first result of Lemma 19.

For the second one, we get

$$\|v\|^2_{\left(I-\Sigma_P^{1/2}\hat{\Sigma}_P^{-1}\Sigma_P^{1/2}\right)^2} = v^\top \left(I - \Sigma_P^{1/2}\hat{\Sigma}_P^{-1}\Sigma_P^{1/2}\right)^2 v$$

$$= \|v\|_2^2 - 2v^\top \Sigma_P^{1/2}\hat{\Sigma}_P^{-1}\Sigma_P^{1/2}v + v^\top \Sigma_P^{1/2}\hat{\Sigma}_P^{-1}\Sigma_P\hat{\Sigma}_P^{-1}\Sigma_P^{1/2}v$$

$$\overset{(i)}{\le} \|v\|_2^2 - \frac{2}{1+\gamma}\|v\|_2^2 + \frac{1}{1-\gamma}v^\top \Sigma_P^{1/2}\hat{\Sigma}_P^{-1}\Sigma_P^{1/2}v$$

$$\le \|v\|_2^2 - \frac{2}{1+\gamma}\|v\|_2^2 + \frac{1}{(1-\gamma)^2}\|v\|_2^2 \qquad \text{(Since } \hat{\Sigma}_P \preceq \tfrac{1}{1-\gamma}\Sigma_P\text{)}$$

$$\le \left(1 - \frac{2}{1+\gamma} + \frac{1}{(1-\gamma)^2}\right)\|v\|_2^2$$

$$\overset{(ii)}{\le} 10\gamma\|v\|_2^2.$$

The inequality (i) above holds because $\frac{1}{1+\gamma}\Sigma_P^{-1} \preceq \hat{\Sigma}_P^{-1}$ and $(1-\gamma)\Sigma_P \preceq \hat{\Sigma}_P \implies \Sigma_P \preceq \frac{1}{1-\gamma}\hat{\Sigma}_P$. The inequality (ii) above holds because for $\gamma \in \left[0, \frac{1}{2}\right]$, we have

$$1 - \frac{2}{1+\gamma} + \frac{1}{(1-\gamma)^2} \le 1 - 2(1-\gamma) + (1+2\gamma)^2 \le 10\gamma.$$

Taking square root on both sides gives us the results. $\qquad\square$

*Proof of Proposition 3.* This proof is analogous to the proof of Proposition 1. We first note that

$$\max_{v\in\mathcal{V}} \frac{|\langle\widehat{\theta}, v\rangle - \langle\theta, v\rangle|}{\|v\|_{\widehat{\Sigma}_P^{-1}}} = \max_{v\in\mathcal{V}} \frac{|\langle\widehat{\theta}, v\rangle - w_v + w_v - \langle\theta, v\rangle|}{\|v\|_{\widehat{\Sigma}_P^{-1}}}$$

$$\le \max_{v\in\mathcal{V}} \frac{|\langle\widehat{\theta}, v\rangle - w_v|}{\|v\|_{\widehat{\Sigma}_P^{-1}}} + \max_{v\in\mathcal{V}} \frac{|w_v - \langle\theta, v\rangle|}{\|v\|_{\widehat{\Sigma}_P^{-1}}}$$

$$= \min_{\theta'}\max_{v\in\mathcal{V}} \frac{|\langle\theta', v\rangle - w_v|}{\|v\|_{\widehat{\Sigma}_P^{-1}}} + \max_{v\in\mathcal{V}} \frac{|w_v - \langle\theta', v\rangle|}{\|v\|_{\widehat{\Sigma}_P^{-1}}}$$

$$\le 2\max_{v\in\mathcal{V}} \frac{|\langle\theta, v\rangle - w_v|}{\|v\|_{\widehat{\Sigma}_P^{-1}}}$$

So it suffices to show that each $|\langle\theta, v\rangle - w_v|$ is small. We begin by fixing some $v \in \mathcal{V}$ and bounding the variance of $v^\top\widehat{\Sigma}_P^{-1}Q_s(x_s)x_sy_s$ for any $s \le n$ which is necessary to use the robust estimator. Note that

$$\mathbb{V}\mathrm{ar}_{x_s\sim\nu, Q_s(x_s)\sim P(x_s)}(v^\top\widehat{\Sigma}_P^{-1}Q_s(x_s)x_sy_s) = \mathbb{E}_{x_s\sim\nu, Q_s(x_s)\sim P(x_s)}[(v^\top\widehat{\Sigma}_P^{-1}Q_s(x_s)x_sy_s)^2]$$
$$- \mathbb{E}_{x_s\sim\nu, Q_s(x_s)\sim P(x_s)}[v^\top\widehat{\Sigma}_P^{-1}Q_s(x_s)x_sy_s]^2$$

which means we can drop the second term to bound the variance by

$$\mathbb{E}_{x_s\sim\nu, Q_s(x_s)\sim P(x_s)}[\left((v^\top\widehat{\Sigma}_P^{-1}Q_s(x_s)x_sy_s)\right)^2]$$

$$= \mathbb{E}_{x_s\sim\nu, Q_s(x_s)\sim P(x_s)}[\left(v^\top\widehat{\Sigma}_P^{-1}Q_s(x_s)x_s(x_s^\top\theta + \xi_s)\right)^2]$$

$$= \mathbb{E}_{x_s\sim\nu}\left[\mathbb{E}_{Q_s(x_s)\sim P(s_s)}[\left(v^\top\widehat{\Sigma}_P^{-1}Q_s(x_s)x_s(x_s^\top\theta)\right)^2] + \mathbb{E}_{Q_s(x_s)\sim P(s_s)}[\left(v^\top\widehat{\Sigma}_P^{-1}Q_s(x_s)x_s\right)^2\xi_t^2]\right]$$

$$\le \mathbb{E}_{x_s\sim\nu}\left[B^2\mathbb{E}_{Q_s(x_s)\sim P(s_s)}[\left(v^\top\widehat{\Sigma}_P^{-1}Q_s(x_s)x_s\right)^2] + \sigma^2\mathbb{E}_{Q_s(x_s)\sim P(s_s)}[\left(v^\top\widehat{\Sigma}_P^{-1}Q_s(x_s)x_s\right)^2]\right]$$

$$= \mathbb{E}_{x_s\sim\nu}\left[(B^2 + \sigma^2)\mathbb{E}_{Q_s(x_s)\sim P(s_s)}[v^\top\widehat{\Sigma}_P^{-1}Q_s(x_s)x_sx_s^\top Q_s(x_s)\widehat{\Sigma}_P^{-1}v]\right]$$

$$= \mathbb{E}_{x_s\sim\nu}\left[(B^2 + \sigma^2)\mathbb{E}_{Q_s(x_s)\sim P(s_s)}[v^\top\widehat{\Sigma}_P^{-1}Q_s(x_s)x_sx_s^\top\widehat{\Sigma}_P^{-1}v]\right]$$

$$\le \mathbb{E}_{x_s\sim\nu}\left[(B^2 + \sigma^2)v^\top\widehat{\Sigma}_P^{-1}P(x_s)x_sx_s^\top\widehat{\Sigma}_P^{-1}v\right],$$

where we used that $Q_s^2(x_s) = Q_s(x_s)$. Thus, we have with Lemma 19

$$\mathbb{V}\mathrm{ar}(v^\top \widehat{\Sigma}_P^{-1} Q_s(x_s) x_s y_s) \leq (B^2 + \sigma^2) v^\top \widehat{\Sigma}_P^{-1} \mathbb{E}_{x_s \sim \nu}[P(x_s) x_s x_s^\top] \widehat{\Sigma}_P^{-1} v$$
$$= (B^2 + \sigma^2) \|v\|_{\widehat{\Sigma}_P^{-1} \Sigma_P \widehat{\Sigma}_P^{-1}}^2$$
$$\leq \frac{B^2 + \sigma^2}{(1-\gamma)^2} \|v\|_{\Sigma_P^{-1}}^2 .$$

We have

$$|\langle \theta_*, v \rangle - w_v| = |\langle \theta_*, v \rangle - \mathbb{E}[v^\top \widehat{\Sigma}_P^{-1} P(x_1) x_1 y_1] + \mathbb{E}[v^\top \widehat{\Sigma}_P^{-1} P(x_1) x_1 y_1] - w_v|$$
$$\leq |\langle \theta_*, v \rangle - \mathbb{E}[v^\top \widehat{\Sigma}_P^{-1} P(x_1) x_1 y_1]|$$
$$+ |\mathrm{Catoni}(\{\langle v, \widehat{\Sigma}_P^{-1} Q_s(x_s) x_s y_s \rangle\}_{s=1}^n) - \mathbb{E}_{X \sim \nu}[v^\top \widehat{\Sigma}_P^{-1} P(X) XY]|.$$

We now recall that we can write $y_t = x_t^\top \theta_* + \xi_t$ where $\xi_t$ is a mean-zero, independent random variable with variance at most $\sigma^2$. Thus, using Cauchy-Schwarz and applying Lemma 19, we get

$$|\langle \theta_*, v \rangle - \mathbb{E}[v^\top \widehat{\Sigma}_P^{-1} P(x_1) x_1 y_1]| = |v^\top \theta_* - v^\top \widehat{\Sigma}_P^{-1} \Sigma_P \theta_*|$$
$$= |v^\top (I - \widehat{\Sigma}_P^{-1} \Sigma_P) \theta_*|$$
$$= |v^\top \Sigma_P^{-1/2} (I - \Sigma_P^{1/2} \widehat{\Sigma}_P^{-1} \Sigma_P^{1/2}) \Sigma_P^{1/2} \theta_*|$$
$$\leq \|\Sigma_P^{-1/2} v\| \ \|\Sigma_P^{1/2} \theta_*\|_{(I - \Sigma_P^{1/2} \widehat{\Sigma}_P^{-1} \Sigma_P^{1/2})^2}$$
$$\leq \sqrt{10\gamma} \|\Sigma_P^{-1/2} v\| \ \|\Sigma_P^{1/2} \theta_*\|$$
$$= \sqrt{10\gamma} \|v\|_{\Sigma_P^{-1}} \|\theta_*\|_{\Sigma_P}.$$

By using the property of Catoni estimator stated in Definition 2, we have

$$|\langle \theta_*, v \rangle - w_v|$$
$$\leq |\mathrm{Catoni}(\{\langle v, \mathbb{E}_{X \sim \nu}[P(X) XX^\top]^{-1} Q_s(x_s) x_s y_s \rangle\}_{s=1}^n) - \mathbb{E}[\langle v, \mathbb{E}_{X \sim \nu}[P(X) XX^\top]^{-1} Q_s(x_s) x_s y_s \rangle]|$$
$$\qquad + \sqrt{10\gamma} \|\theta_*\|_{\mathbb{E}_{X \sim \nu}[XX^\top]} \|v\|_{(\mathbb{E}_{X \sim \nu}[P(X) XX^\top])^{-1}}$$
$$\leq \sqrt{2} \sqrt{(\mathbb{V}\mathrm{ar}(\langle v, \mathbb{E}_{X \sim \nu}[P(X) XX^\top]^{-1} Q_s(x_s) x_s y_s \rangle)) \frac{\log(\frac{2}{\delta})}{n/2}}$$
$$\qquad + \sqrt{10\gamma} \|\theta_*\|_{\mathbb{E}_{X \sim \nu}[XX^\top]} \|v\|_{(\mathbb{E}_{X \sim \nu}[P(X) XX^\top])^{-1}}$$
$$\qquad\qquad\qquad\qquad \text{(with probability at least } 1 - \delta \text{ if } n \geq 4 \log(2/\delta))$$
$$\leq \left( \sqrt{4} \sqrt{\frac{B^2 + \sigma^2}{(1-\gamma)^2}} + \sqrt{10n\gamma} \|\theta_*\|_{\mathbb{E}_{X \sim \nu}[XX^\top]} \right) \|v\|_{(\mathbb{E}_{X \sim \nu}[P(X) XX^\top])^{-1}} \sqrt{\frac{\log(\frac{2}{\delta})}{n}}$$
$$= \left( \sqrt{4} \sqrt{\frac{B^2 + \sigma^2}{(1-\gamma)^2}} + \sqrt{10n\gamma} \|\theta_*\|_{\mathbb{E}_{X \sim \nu}[XX^\top]} \right) \|v\|_{\mathbb{E}_{X \sim \nu}[n P(X) XX^\top]^{-1}} \sqrt{\log(2/\delta)}.$$

Finally, the proof is complete by taking union bounding over all $v \in \mathcal{V}$. $\qquad \square$

*Proof of Lemma 17.* Most of this proof is exactly the one of Section B.1 and Section B.1.1 so we only state the concentration bound. For any $\mathcal{V} \subseteq \mathcal{Z}$ and $z, z' \in \mathcal{V}$ define

$$\mathcal{E}_{z,z',\ell}(\mathcal{V}) = \{|\langle z - z', \widehat{\theta}_\ell(\mathcal{V}) - \theta_* \rangle| \leq \epsilon_\ell\}$$

where $\widehat{\theta}_\ell(\mathcal{V})$ is the estimator that would be constructed by the algorithm at stage $\ell$ with $\mathcal{Z}_\ell = \mathcal{V}$. Naturally we want to apply Proposition 3 with $\tau$ labeled samples to obtain that $\mathcal{E}_{z,z',\ell}(\mathcal{V})$ holds with probability at least $1 - \frac{\delta}{2\ell^2 |\mathcal{Z}|^2}$. Note that as Lemma 14 gives $P(x) \geq \mu/3$ so

$$\Sigma_P = \mathbb{E}_{X \sim \nu}[P(X) XX^\top] \geq \frac{\mu}{3} \mathbb{E}_{X \sim \nu}[XX^\top]$$

$\Sigma_P$ is invertible.

Defining $\delta_0 := \frac{\delta}{4\ell^2|\mathcal{Z}|^2}$ and setting $\kappa \geq 2c_{\delta_0}\max\{1, 20\|\theta_*\|^2_{\mathbb{E}_{X\sim\nu}[XX^\top]}\}$ where we recall that was defined $c_\delta = K^2_{\psi_2}(\sqrt{d\ln 9/c_1} + \sqrt{\frac{\log(2/\delta)}{c_1}})$, Lemma 18 leads to

$$\frac{c_{\delta_0}}{\kappa} \leq \frac{1}{2}\min\left\{1, \frac{1}{20\|\theta_*\|^2_{\mathbb{E}_{X\sim\nu}[XX^\top]}}\right\}$$

so that we can set $\gamma = c_{\delta_0}/(\tau\kappa)$ in the bound of Proposition 3 to get

$$\sqrt{10\tau\gamma}\|\theta_*\|_{\mathbb{E}_{X\sim\nu}[XX^\top]} \leq \frac{1}{2}$$

and

$$\sqrt{\frac{B^2 + \sigma^2}{(1-\gamma)^2}} \leq 2\sqrt{B^2 + \sigma^2}$$

So for $\delta_0 = \frac{\delta}{4\ell^2|\mathcal{Z}|^2}$ the event $\widetilde{\mathcal{E}}_{\text{cov}}$ defined as

$$\widetilde{\mathcal{E}}_{\text{cov}} := \left\{\left(1 - \frac{c_{\delta_0}}{\sqrt{\kappa}}\right)x^\top\Sigma_P x \leq x^\top\widehat{\Sigma}_P x \leq \left(1 + \frac{c_{\delta_0}}{\sqrt{\kappa}}\right)x^\top\Sigma_P x\right\}.$$

happen with probability at least $1 - \delta_0$.

Now, let us for now condition on $\widetilde{\mathcal{E}}_{\text{cov}}$. For fixed $\mathcal{V} \subset \mathcal{Z}$ and $\ell \in \mathbb{N}$ we apply Proposition 3, instantiating the arbitrary $P$ to $\widehat{P}_\ell$ (obtained with OPTIMIZEDESIGN, recall Section D.1) so that with probability at least $1 - \frac{\delta}{4\ell^2|\mathcal{Z}|^2}$ we have that for any $z, z' \in \mathcal{V}$ holds that the event $\widetilde{\mathcal{E}}_{\text{RIPS},z,z'}$ defined as

$$\widetilde{\mathcal{E}}_{\text{RIPS},z,z'} := \left\{|\langle z - z', \widehat{\theta}_\ell(\mathcal{V}) - \theta_*\rangle|\right.$$

$$\left. \leq 2\|z - z'\|_{\mathbb{E}_{X\sim\nu}[\tau\widehat{P}_\ell(X)XX^\top]^{-1}}\left(4\sqrt{B^2 + \sigma^2} + 1\right)\sqrt{\log(4\ell^2|\mathcal{Z}|^2/\delta)}\right\}$$

happen with probability at least $1 - \delta_0$.

So with probability at least $1 - \mathbb{P}(\widetilde{\mathcal{E}}^c_{\text{RIPS},z,z'}) - \mathbb{P}(\widetilde{\mathcal{E}}^c_{\text{cov}}) \geq 1 - \frac{\delta}{4\ell^2|\mathcal{Z}|^2} - \frac{\delta}{4\ell^2|\mathcal{Z}|^2} = 1 - \frac{\delta}{2\ell^2|\mathcal{Z}|^2}$, both events hold and we have that for any $z, z' \in \mathcal{V}$ holds

$$|\langle z - z', \widehat{\theta}_\ell(\mathcal{V}) - \theta_*\rangle| \leq 2\|z - z'\|_{\mathbb{E}_{X\sim\nu}[\tau\widehat{P}_\ell(X)XX^\top]^{-1}}\left(4\sqrt{B^2 + \sigma^2} + 1\right)\sqrt{\log(4\ell^2|\mathcal{Z}|^2/\delta)}$$

$$\leq 2(1+\varepsilon)\left(4\sqrt{B^2 + \sigma^2} + 1\right)\|z - z'\|_{\mathbb{E}_{X\sim\nu}[\tau\widehat{P}_\ell(X)XX^\top]^{-1}}\sqrt{\log(4\ell^2|\mathcal{Z}|^2/\delta)}$$

$$\leq \epsilon_\ell.$$

where we used the property of $\widehat{P}_\ell$ as detailed in Section D.1 to conclude. $\qquad\square$

*Proof of Theorem 7.* The total number of labels requested after $L$ rounds is equal to $\sum_{\ell=1}^L \sum_{t=(\ell-1)\tau+1}^{\ell\tau} \widehat{P}_\ell(x_t)$. Again by Freedman's inequality we have that

$$\sum_{\ell=1}^L \sum_{t=(\ell-1)\tau+1}^{\ell\tau} \widehat{P}_\ell(x_t) \leq 2\sum_{\ell=1}^L \tau\mathbb{E}_{X\sim\nu}[\widehat{P}_\ell(X)|\mathcal{Z}_\ell] + \log(1/\delta)$$

From Theorem 4, it holds for any $\ell$ that $\mathbb{E}_{X\sim\nu}[\widehat{P}_\ell(X)] \leq \mathbb{E}_{X\sim\nu}[\widetilde{P}_\ell(X)] + 4\sqrt{\mu}$ where $\widetilde{P}_\ell$ is the optimal solution to problem (20). So now, for some $\widetilde{\tau}$, we want to relate $\mathbb{E}_{X\sim\nu}[\widetilde{\tau}\widetilde{P}_\ell(X)]$ to $\mathbb{E}_{X\sim\nu}[\tau P_\ell(X)]$ where $P_\ell$ is the solution of problem (4). To do so, we rewrite problem (4) and problem (20) as

$$\begin{aligned}
\min_P \quad & \mathbb{E}_{X\sim\nu}[\tau P(X)] \\
\text{subject to} \quad & y^\top\mathbb{E}_{X\sim\nu}[\tau P(X)XX^\top]^{-1}y \leq c_\ell^2, \quad \forall y \in \mathcal{Y}_\ell, \\
& 0 \leq \tau P(x) \leq \tau, \quad \forall x \in \mathcal{X}.
\end{aligned} \tag{32}$$

and

$$\begin{aligned}
\min_P \quad & \mathbb{E}_{X\sim\nu}[\widetilde{\tau}P(X)] \\
\text{subject to} \quad & y^\top \mathbb{E}_{X\sim\nu}[\widetilde{\tau}P(X)XX^\top]^{-1}y \le c_\ell^2, \quad \forall y \in \mathcal{Y}_\ell, \\
& 0 \le \widetilde{\tau}P(x) \le \widetilde{\tau}(1-\mu_b), \quad \forall x \in \mathcal{X}.
\end{aligned} \tag{33}$$

where problem (32) is equivalent to problem (4) and problem (33) is equivalent to problem (20). Thus taking $\widetilde{\tau} = \frac{\tau}{1-\mu_b}$, problem (33) becomes

$$\begin{aligned}
\min_P \quad & \mathbb{E}_{X\sim\nu}\left[\frac{\tau}{1-\mu_b}P(X)\right] \\
\text{subject to} \quad & y^\top \mathbb{E}_{X\sim\nu}[\frac{\tau}{1-\mu_b}P(X)XX^\top]^{-1}y \le c_\ell^2, \quad \forall y \in \mathcal{Y}_\ell, \\
& 0 \le \frac{\tau}{1-\mu_b}P(x) \le \tau, \quad \forall x \in \mathcal{X}.
\end{aligned}$$

which, using $Q = \frac{P}{1-\mu_b}$ is equivalent to

$$\begin{aligned}
\min_Q \quad & \mathbb{E}_{X\sim\nu}[\tau Q(X)] \\
\text{subject to} \quad & y^\top \mathbb{E}_{X\sim\nu}[\tau Q(X)XX^\top]^{-1}y \le c_\ell^2, \quad \forall y \in \mathcal{Y}_\ell, \\
& 0 \le \tau Q(x) \le \tau, \quad \forall x \in \mathcal{X}.
\end{aligned} \tag{34}$$

And we can now see that (34) and (32) are the same optimization problem. And $Q_\ell^*$ the solution of (34) is equal to $\frac{\widetilde{P}_\ell}{1-\mu_b}$. Thus the result $\mathbb{E}_{X\sim\nu}\left[\widetilde{\tau}\widetilde{P}_\ell(X)\right] = \mathbb{E}_{X\sim\nu}[\tau P_\ell(X)]$.

Remains to bound $\sum_{\ell=1}^L \tau\mathbb{E}_{X\sim\nu}[P_\ell(X)]$ where

$$\sum_{\ell=1}^L \tau\mathbb{E}_{X\sim\nu}[P_\ell(X)|\mathcal{Z}_\ell]$$

$$= \sum_{\ell=1}^L \left[ \min_{P:\mathcal{X}\to[0,1]} \tau\mathbb{E}_{X\sim\nu}[P(X)] \quad \text{subject to} \quad \max_{z,z'\in\mathcal{Z}_\ell} \frac{\|z-z'\|^2_{\mathbb{E}_{X\sim\nu}[\tau P(X)XX^\top]^{-1}}}{\epsilon_\ell^2}\beta_{\delta,\ell} \le 1 \right],$$

where $\beta_{\delta,\ell}$ is defined in Section D.1 as

$$\beta_{\delta,\ell} := 4(1+\varepsilon)^2\left(4\sqrt{B^2+\sigma^2}+1\right)^2\log(4\ell^2|\mathcal{Z}|^2/\delta).$$

As in the case where the distribution $\nu$ is known (Section B.1), we use Lemma 3 to bound $\max_{z,z'\in\mathcal{Z}_\ell} \frac{\|z-z'\|^2_{\mathbb{E}_{X\sim\nu}[\tau P(X)XX^\top]^{-1}}}{\epsilon_\ell^2}\beta_{\delta,\ell}$ by $\max_{z\in\mathcal{Z}\setminus z_*} \frac{\|z-z_*\|^2_{\mathbb{E}_{X\sim\nu}[\tau P(X)XX^\top]^{-1}}}{\langle z-z_*,\theta_*\rangle^2}64\beta_{\delta,L}$. Last, the reparameterization of Proposition 2 also applies here.

In the unlabeled sample complexity, we get an additional $L\kappa = L\lceil 2K_{\psi_2}^2(\sqrt{d\ln 9/c_1} + \sqrt{\frac{\log(2/\delta)}{c_1}})\max\{1, 20\|\theta_*\|_{\mathbb{E}_{X\sim\nu}[XX^\top]}\}\rceil$ term from the estimation of the covariance matrix. Last, we get an additional $L(K+u)$, where $K$ and $u$ are such that

$$K \ge \widetilde{O}\left(\frac{|\mathcal{Z}|^3\kappa(\Sigma)^2\|\Lambda^*\|_2^8 M^{16}}{\beta^2\mu_b^6}\right)\cdot\left(\frac{1+\epsilon}{\epsilon}\right)^2, \quad u \ge \widetilde{O}\left(\frac{\kappa(\Sigma)^2\|\Lambda^*\|_2^6 M^{16}}{\beta^2\mu_b^6}\right)\cdot\left(\frac{1+\epsilon}{\epsilon}\right)^2,$$

from the sample complexity of the subroutine. $\qquad\square$

## E  Classification

In this section we adopt the implementation described in Section B.1. As described in the text, given a distribution $\pi \in \Delta_\mathcal{X}$, and a class of hypothesis $\mathcal{H}$, we can reduce classification to linear bandits by setting $\theta^* = [\theta_x^*]_{x\in\Delta_\mathcal{X}}$ where $\theta_x^* = 2\eta(x)-1$, and $\mathcal{Z} := \{z^{(h)}\}_{h\in\mathcal{H}} \subset [0,1]^{|\mathcal{X}|}$ where $z_x^{(h)} = \pi(x)\mathbf{1}\{h(x)=1\}$. With the quantities computed in Section 3, we now prove Theorem 3.

*Proof of Theorem 3.* We consider a slightly modified version of Algorithm 1 where we stop at round $L$ where $L_\epsilon = \lceil\log_2(4/\epsilon)\rceil$ and return $\arg\max_{z^{(h)}\in\mathcal{Z}_\ell}\langle z^{(h)},\widehat{\theta}_\ell\rangle$. By an identical analysis to that in the proof of Theorem 2, we are guaranteed that $h \in \widehat{\mathcal{S}}_\ell$, i.e. $R_\nu(h) - R_\nu(z^*) = \langle z^* - z, \theta_*\rangle \le 4\epsilon_\ell$.

In addition the analysis of the sample complexity given there immediately gives the first part of the theorem.

It remains to bound the sample complexity in terms of the disagreement coefficient. The total sample complexity is given by,

$$\sum_{\ell=1}^{L} \left[ \min_{P:\mathcal{X}\to[0,1]} \tau \mathbb{E}_{X\sim\nu}[P(X)] \quad \text{subject to} \quad \max_{z\in\mathcal{S}_\ell} \frac{\|z - z_*\|^2_{\mathbb{E}_{X\sim\nu}[\tau P(X)XX^\top]^{-1}}}{\epsilon_\ell^2} \beta_\delta \leq 1 \right]$$

where we recall $\beta_\delta = 2048 \log(2L^2|\mathcal{H}|/\delta)$ since we can take $B = 1$ and $\sigma = 1$.

We recall the proof of Theorem 2. From the proof, we see that with probability greater than $1 - \delta$, our sample complexity is obtained by summing up to round $L$

$$\sum_{\ell=1}^{L} \left[ \min_{P:\mathcal{X}\to[0,1]} \tau \mathbb{E}_{X\sim\nu}[P(X)] \quad \text{subject to} \quad \max_{z\in\mathcal{S}_\ell} \frac{\|z - z_*\|^2_{\mathbb{E}_{X\sim\nu}[\tau P(X)XX^\top]^{-1}}}{\epsilon_\ell^2} \beta_\delta \leq 1 \right]$$

By proposition 2 this is equivalent to

$$\sum_{\ell=1}^{L} \left[ \min_{\lambda\in\Delta_X} \rho_\ell(\lambda)\beta_\delta \quad \text{subject to} \quad \left\|\frac{\lambda}{\nu}\right\|_\infty \rho_\ell(\lambda)\beta_\delta \leq \tau \right], \quad \text{where } \rho_\ell(\lambda) := \max_{z\in\mathcal{S}_\ell} \frac{\|z - z_*\|^2_{\mathbb{E}_{X\sim\lambda}[XX^\top]^{-1}}}{\epsilon_\ell^2}.$$

Define

$$A_\ell = \{x \in \mathcal{X} : \exists h, h(x) \neq h^*(x), R_\nu(h) - R_\nu(h^*) \leq 4\epsilon_\ell\}, \ell \leq L$$

and let $\lambda_\ell = \dfrac{\mathbf{1}\{x \in A_\ell\}\nu(x)}{\mathbb{E}[\mathbf{1}\{x \in A_\ell\}]}$, so $\left\|\dfrac{\lambda}{\nu}\right\|_\infty = \dfrac{1}{\mathbb{E}[\mathbf{1}\{x \in A_i\}]}$.

We first argue that $\lambda_\ell$ is feasible for the previous program. Note,

$$\rho_\ell(\lambda_\ell) = \max_{h:R_\nu(h)-R_\nu(h^*)\leq 4\epsilon_\ell} \frac{\mathbb{E}_{X\sim\nu}\left[\frac{\mathbf{1}\{h(x)\neq h^*(x)\}}{\lambda_\ell(x)/\nu(x)}\right]}{\epsilon_\ell^2}$$

$$\overset{(i)}{=} \mathbb{E}[\mathbf{1}\{x \in A_\ell\}] \max_{h:R_\nu(h)-R_\nu(h^*)\leq 4\epsilon_\ell} \frac{\mathbb{E}_{X\sim\nu}[\mathbf{1}\{h(x) \neq h^*(x)\}]}{\epsilon_\ell^2}$$

$$\leq \mathbb{E}[\mathbf{1}\{x \in A_\ell\}] \max_{h:R_\nu(h)-R_\nu(h^*)\leq 4\epsilon_\ell} \frac{16\mathbb{E}_{X\sim\nu}[\mathbf{1}\{h(x) \neq h^*(x)\}]}{\max\{\epsilon_\ell^2, (R_\nu(h) - R_\nu(h^*))^2\}}$$

$$\leq \mathbb{E}[\mathbf{1}\{x \in A_\ell\}] \max_{h:R_\nu(h)-R_\nu(h^*)\leq 4\epsilon_\ell} \frac{16\mathbb{E}_{X\sim\nu}[\mathbf{1}\{h(x) \neq h^*(x)\}]}{\max\{(4\epsilon_\ell)^2, (R_\nu(h) - R_\nu(h^*))^2\}}$$

$$\overset{(ii)}{\leq} \mathbb{E}[\mathbf{1}\{x \in A_\ell\}] \max_{h:R_\nu(h)-R_\nu(h^*)\leq 4\epsilon_\ell} \frac{16\mathbb{E}_{X\sim\nu}[\mathbf{1}\{h(x) \neq h^*(x)\}]}{\max\{\epsilon^2, (R_\nu(h) - R_\nu(h^*))^2\}}$$

$$\leq \mathbb{E}[\mathbf{1}\{x \in A_\ell\}] \max_{h\in H} \frac{16\mathbb{E}_{X\sim\nu}[\mathbf{1}\{h(x) \neq h^*(x)\}]}{\max\{\epsilon^2, (R_\nu(h) - R_\nu(h^*))^2\}}$$

$$\leq 16\mathbb{E}[\mathbf{1}\{x \in A_\ell\}]\rho(\nu, \epsilon)$$

where the equality (i) holds because the following is true when we only consider $h$ such that $R_\nu(h) - R_\nu(h^*) \leq 4\epsilon_\ell$

$$\frac{\mathbf{1}\{h(x) \neq h^*(x)\}}{\mathbf{1}\{x : \exists h, h(x) \neq h^*(x), (R_\nu(h) - R_\nu(h^*)) \leq 4\epsilon_\ell\}} = \mathbf{1}\{h(x) \neq h^*(x)\}.$$

The inequality (ii) above is true because $4\epsilon_\ell \geq \epsilon$. Thus we see that $\rho_\ell(\lambda_\ell)\|\lambda/\nu\|_\infty\beta_\delta \leq 16\rho(\nu, \epsilon)\beta_\delta \leq \tau$. It remains to argue about the disagreement coefficient. Firstly note that for any $h$ such that $R_\nu(h) - R_\nu(h^*) \leq 4\epsilon_\ell$.

$$d_\nu(h, h^*) = \mathbb{E}_{X\sim\nu}[\mathbf{1}\{h(X) \neq h^*(X)\}] \leq \mathbb{E}_{X\sim\nu}[\mathbf{1}\{h(X) \neq Y\}] + \mathbb{E}_{X\sim\nu}[\mathbf{1}\{h^*(X) \neq Y\}] \tag{35}$$

$$\leq R_\nu(h) + R_\nu(h^*) \tag{36}$$

$$\leq 2R_\nu(h^*) + 4\epsilon_\ell \tag{37}$$

Using this we see that,

$$\min_{\lambda \in \Delta} \rho_\ell(\lambda) \text{ subject to } \rho_\ell(\lambda)\|\lambda/\nu\|_\infty \beta_\delta \le \tau$$

$$\le \rho_\ell(\lambda_\ell)\beta_\delta \qquad\qquad\qquad\qquad \text{(since } \lambda_\ell \text{ is feasible.)}$$

$$\le \mathbb{E}[\mathbf{1}\{x \in A_\ell\}] \max_{h:R_\nu(h)-R_\nu(h^*)\le 4\epsilon_\ell} \frac{\mathbb{E}_{X\sim\nu}[\mathbf{1}\{h(x)\ne h^*(x)\}]}{\epsilon_\ell^2}\beta_\delta$$

$$\text{(imitating the above computation)}$$

$$\le \frac{(2R(h^*)+4\epsilon_\ell)\mathbb{E}_{X\sim\nu}[\mathbf{1}\{\exists h : h(X)\ne h^*(X), d_\nu(h,h^*)\le 2R(h^*)+4\epsilon_\ell\}]}{\epsilon_\ell^2}\beta_\delta$$

$$\text{(Equation (35))}$$

$$\le \beta_\delta \begin{cases} \frac{9R(h^*)^2}{\epsilon_\ell^2} \frac{\mathbb{E}_{X\sim\nu}[\mathbf{1}\{\exists h:h(X)\ne h^*(X),d_\nu(h,h^*)\le 2R(h^*)+4\epsilon_\ell\}]}{2R(h^*)+4\epsilon_\ell} & 4\epsilon_\ell \le R(h^*) \\ \frac{144\mathbb{E}_{X\sim\nu}[\mathbf{1}\{\exists h:h(X)\ne h^*(X),d_\nu(h,h^*)\le 2R(h^*)+4\epsilon_\ell\}]}{2R(h^*)+4\epsilon_\ell} & 4\epsilon_\ell > R(h^*) \end{cases}$$

$$\le \left(\frac{9R(h^*)^2}{\epsilon_\ell^2}+144\right)\frac{\mathbb{E}_{X\sim\nu}[\mathbf{1}\{\exists h : h(X)\ne h^*(X), d_\nu(h,h^*)\le 2R(h^*)+4\epsilon_\ell\}]}{2R(h^*)+4\epsilon_\ell}\beta_\delta$$

Thus,

$$\sum_{\ell=1}^{L}\left[\min_{\lambda\in\Delta_X}\rho_\ell(\lambda)\beta_\delta \quad \text{subject to} \quad \left\|\frac{\lambda}{\nu}\right\|_\infty \rho_\ell(\lambda)\beta_\delta \le \tau\right]$$

$$\le \sum_{\ell=1}^{L}\rho_\ell(\lambda_\ell)\beta_\delta$$

$$\le \sum_{\ell=1}^{L}\left(\frac{9R(h^*)^2}{\epsilon_\ell^2}+144\right)\frac{\mathbb{E}_{X\sim\nu}[\mathbf{1}\{\exists h : h(X)\ne h^*(X), d_\nu(h,h^*)\le 2R(h^*)+4\epsilon_\ell\}]}{2R(h^*)+4\epsilon_\ell}\beta_\delta$$

$$\le \log_2\left(\frac{4}{\epsilon}\right)\sup_{\ell\le L}\left(\frac{9R(h^*)^2}{\epsilon_\ell^2}+144\right)\frac{\mathbb{E}_{X\sim\nu}[\mathbf{1}\{\exists h : h(X)\ne h^*(X), d_\nu(h,h^*)\le 2R(h^*)+\epsilon_\ell\}]}{2R(h^*)+4\epsilon_\ell}\beta_\delta$$

$$\le \log_2\left(\frac{4}{\epsilon}\right)\left(\frac{36R(h^*)^2}{\epsilon^2}+144\right)\sup_{\ell\le L}\frac{\mathbb{E}_{X\sim\nu}[\mathbf{1}\{\exists h : h(X)\ne h^*(X), d_\nu(h,h^*)\le 2R(h^*)+4\epsilon_\ell\}]}{2R(h^*)+4\epsilon_\ell}\beta_\delta$$

$$\le 36\log_2\left(\frac{4}{\epsilon}\right)\left(\frac{R(h^*)^2}{\epsilon^2}+4\right)\sup_{\xi\ge\epsilon}\theta^*(2R(h^*)+\xi,\nu)\beta_\delta$$

from which the result follows.

$$\square$$