# OpenReview forum: "Selective Sampling for Online Best-arm Identification"
_NeurIPS.cc/2021/Conference — NeurIPS 2021 Poster_

### Official Review · Reviewer_TR9q · 2021-07-13

**Rating:** 7
**Confidence:** 4

**Summary:**

This paper considers an interesting problem of selective-sampling for best-arm identification. The paper provides both lower and upper bounds that capture the trade-off between labeled samples and stopping time. The proposed algorithm is equipped with a clear optimization solver and achieves the near-optimal label complexity given a desired stopping time.

**Limitations And Societal Impact:**

Yes

**Main Review:**

This is a good paper. The considered selective-sampling for best-arm identification problem is very interesting, and the “selective-sampling” concept may be applicable to other bandit problems and real-world online learning platforms. The theoretical analysis seems complete and solid. In particular, the proposed projected stochastic gradient ascent method could be of independent interests in other optimal experiment design problems.

One of the weaknesses is that this paper does not present empirical evaluations. It would be better to present experimental results to demonstrate the empirical performance of the proposed algorithm.

**After Rebuttal**

I have read the rebuttal of the authors.
The authors gave experimental results to demonstrate good empirical performance of the proposed algorithms, which well addressed my concerns. I think that this paper makes nice technical contributions to the bandit community and is worth acceptance.
Hence, I decided to increase my score from 6 to 7.



**Time Spent Reviewing:**

4

---

> ### Author Response · Authors · 2021-08-11
> **Response to Reviewer TR9q**
>
> Thank you very much for your review and suggestions! The experimental details are provided at the top of the page as an official comment for the paper. In particular, the given plot shows the trade-off between label complexity $\mathcal{L}$ and sample complexity $\mathcal{U}$ characterized by our theorems. Meanwhile, it also shows that our algorithm has much less label complexity than the naive algorithm.

---

> > ### Comment · Reviewer_TR9q · 2021-08-15
> > **Response to Authors**
> >
> > Thank you for your reply and experiments. My concern is well addressed.

---

> > > ### Author Response · Authors · 2021-08-16
> > > **Response to Reviewer TR9q**
> > >
> > > I am glad we were able to address your concern. If there is nothing else, are you willing to increase your score?

---

> > > > ### Comment · Reviewer_TR9q · 2021-08-30
> > > > **Response to Authors**
> > > >
> > > > I appreciate your nice technical contributions and supplemented experimental results, so I increased my score to 7. Hope that you could include the complete empirical evaluations in the final version.

---

### Official Review · Reviewer_5GUj · 2021-07-13

**Rating:** 7
**Confidence:** 3

**Summary:**

This paper considers the problem of best-arm identification in a set Z. The environment samples a vector x from a set X and provides it to the learner. The learner may either choose to observe the reward corresponding to x (given by theta^T x + noise) or ignore it. The authors derive lower and upper bounds on U (number of samples provided by the environment) and L (number of rewards requested by the learner) in terms of a tuning parameter tau and the error probability delta.

The algorithm runs in rounds. In each round, there is an active set of arms that can potentially be the best arm. The learner solves an optimization problem to find a function P: X -> [0, 1], where P(x) is the probability with which it observes the reward for x. The goal of this optimization problem is to minimize the expected number of observations made in tau steps, while ensuring that P(x) does not reject so many samples that the problem of identifying the best arm with probability delta in the first tau steps becomes infeasible.

The authors also show how their approach can be used for binary classification as well, again with selective sampling of outputs.

**Limitations And Societal Impact:**

The authors can include a more detailed discussion about the limitations of their work. For example, their algorithm discards the collected data after each round. A more adaptive strategy will not do so.

**Main Review:**

Originality – The paper studies an interesting problem and provides a simple and intuitive solution.

Quality –
1.	The optimization problem (2) is well motivated, and the main results appear to be sound (although I have not checked all the details in the appendices).
2.	It is unclear how one would go about setting the value of tau in practice. As \rho(\lambda) cannot be computed apriori, just finding a value that ensures feasibility of the optimization problem seems to be tricky. Adding discussion on this will be helpful.
3.	As the algorithm solves a relaxed optimization problem (from (2) to (3)), it will be useful to have experiments to demonstrate the performance of the algorithm in practice. One can see the variation of L as a function of the size of Z for a fixed tau. It will also be interesting to see the effect of tau on L. Such experiments will also help the reader in the context of point 2.

Clarity – The paper is mostly well written; however, the organization is a bit awkward. More specifically, Section 3 breaks the continuity between Section 2 and 4. Consider moving Section 3 after Section 4.

Significance – The problem is interesting, and the contributions appear to be significant in my opinion. I especially appreciate the motivating examples in L43-56, which indicate that some practical applications may benefit from this work. Unfortunately, the authors did not include experiments to guide a reader in choosing parameters like tau.

Strengths – Interesting problem, well-motivated solution, mostly clearly written, intuitive and simple solution.

Weaknesses – No experiments (see more details above)


**Update**: I increased my score as the authors have addressed all of my concerns.

**Time Spent Reviewing:**

8

---

> ### Author Response · Authors · 2021-08-11
> **Response to Reviewer 5GUj**
>
> Thank you very much for your careful review and constructive suggestions on paper organization!
>
> - On the concern regarding setting $\tau$: this is a domain-specific parameter of the algorithm that can be tuned, which is illustrated via the experiments. Since $\tau$ traces out a Pareto frontier with respect to the number of measurements $\mathcal{L}$, there is no concept of a "best" $\tau$. To find a feasible $\tau$, which is claimed to satisfy $\tau\geq\rho(\nu)\beta_\delta$ in our Theorem 2, we just need to estimate an upper bound for $\rho(\nu)$. To do this, we can use a set of unlabeled i.i.d. samples from $\nu$ and replace $\max_{z\in\mathcal{Z}\setminus\lbrace z\_\*\rbrace}$ by $\max_{z, z'\in\mathcal{Z}, z\neq z'}$ to estimate $\rho(\nu)$'s numerator. Finally, an estimate for $\min_{z\in\mathcal{Z}\setminus\lbrace z\_\*\rbrace}\langle\theta_*, z-z_* \rangle$ can be done by using problem-dependent domain knowledge.
> - To shed light to the effect of relaxing the optimization problem, we provide here an experiment that compares using the oracle optimization problem - (2) in the paper - with using the optimization problem that does not assume the knowledge of $\theta_*$. From the plot above, we can see that although our relaxation from (2) to (3) results a little more label queries, it has the same trend as problem (2) and much less label complexity than the naive algorithm.
> - We evaluate the effect of $\tau$ on $\mathcal{L}$ in the experiments and is shown in the plot. We did not evaluate $\mathcal{L}$ versus $|\mathcal{Z}|$ because we believe the size of $\mathcal{Z}$ has no significant effects on the sample complexity, and can even be made completely independant as in [KJKJ].
>
> [KJKJ] Katz-Samuels, J., Jain, L., Karnin, Z., & Jamieson, K. (2020). An empirical process approach to the union bound: Practical algorithms for combinatorial and linear bandits. arXiv preprint arXiv:2006.11685.

---

> > ### Comment · Reviewer_5GUj · 2021-08-12
> > **Thanks for the response**
> >
> > Thank you for addressing my concerns and for providing the numerical result. I have increased my score.
> >
> > I just wanted to get one more clarification: One of the reviewers has raised an interesting point about $P^*$ being static. When you say that the $P$ used by you is adaptive, is it because you compute $P$ based on $\mathcal{Z}_\ell$, which changes after each elimination round? I wonder if a more efficient strategy is possible that can adapt faster as opposed to just adapting at the end of each round.

---

> > > ### Author Response · Authors · 2021-08-14
> > > **Thanks for the follow-up**
> > >
> > > Thank you very much for your increased score and the follow-up question. Yes, we say $P$ is adaptive because it evolves over rounds, updated after each elimination round. While we believe updating the sampling function $P$ more frequently (e.g. every sample) may be more effective in practice, a consequence of our lower bound is that this improvement in sample complexity is limited to a constant factor. However, one important advantage of updating $P$ only infrequently in contrast to every sample is that the optimization problem to solve for $P$, while efficient in the sense it is convex, is still time consuming and one wants to limit the number of times one must solve this optimization problem in practice. Thus, how often $P$ is updated is a trade off of sample complexity and computational complexity.

---

### Official Review · Reviewer_uXS1 · 2021-07-16

**Rating:** 6
**Confidence:** 3

**Summary:**

This paper considers the problem of selective sampling for best-arm identification. At each time step, nature reveals a potential measurement to the learner and the learner decides to either query (contributes 1 count to $\mathcal{L}$) or abstain. The objective of the leaner is to identify the best arm, w.r.t. the unknown environment parameter $\theta_*$, with probability at least $1-\delta$ at a learner specified stopping time $\mathcal{U}$. The main results of this paper characterize the information-theoretic trade-off between labeled samples $\mathcal{L}$ and stopping time $\mathcal{U}$, and presents an algorithm that nearly matches the lower bound. The framework is also extended to cover binary classification problems.

**Ethical Concerns:**

N.A.

**Limitations And Societal Impact:**

See comments on limitations above. There is no immediate societal impact.

**Main Review:**

Originality:

The authors present the fundamental limits on the trade-off between unlabeled data and labeled data, and a novel algorithm that nearly matches the lower bound.

Clarity:

The paper is well-written. I enjoy reading this paper.

Main Comments:

1) The optimization problem formulation and the algorithms are coupled with good intuitive explanations. I enjoy reading this paper.

2) **On the non-adaptive selective sampling**: From the optimization problem defined in Eq. (2), it seems that the problem focuses on the non-adaptive regime (i.e., the optimal $P^*$ is fixed through the whole sampling process, instead of being adaptive to the observations $y_t$). While the non-adaptive setting is valid itself, it is also interesting to see how the lower bound will change if we allow the sampling distribution $P$ to be adaptive (i.e., sampling distribution $P_t$ at time step $t$ can be adaptive to $y_\tau$ seen so far, where $\tau < t$).

3) **Follow-up on adaptive v.s. non-adaptive**: In the algorithm design, the sampling distribution $\widehat P_l$ seems to be adaptive to the historical responses $y_t$ (i.e., in equation (3), the set $\mathcal Z_l$ shrinking as $l$ getting larger is a result of being adaptive to $y_\tau$ seen so far). Therefore it seems that Theorem 2 is not rigorously comparable to Theorem 1 (as Theorem 1 is obtained with $P$ being non-adaptive). It would be good if the authors can clarify this.

4) **Possible Experiments**: Given this paper presents the full fundamental trade-off between the amount of unlabeled data seen (as opposed to a single point) and a corresponding matching algorithm, I'm wondering if the "curve" of such trade-off can be empirically verified by some (synthetic) experiments. This will help understanding/verifying both the validity of the lower bound and the optimality of the proposed algorithm.

Minor Comments:

1) There is a mixture of "labeled" and "labelled" in this paper.

2) It might be good to give a formal definition of $\lambda$ and $\nu$ in section 2.1. From my understanding, $\nu$ is the distribution of $x_t$ generated by nature, and $\lambda$ is the distribution of $x_t$ after selective sampling (those definitions are now buried in explanations).

**Time Spent Reviewing:**

5

---

> ### Author Response · Authors · 2021-08-11
> **Reponse to Reviewer uXS1**
>
> Thank you very much for your careful review and constructive suggestions! We will add more explanations on the interpretation of distributions $\nu$ and $\lambda$.
>
> First, we are indeed focusing on the adaptive regime as our proposed algorithm is an adaptive one. As for $P^*$, it mainly serves as a motivation of our algorithm and benchmark for comparison; however, we do not treat it as the optimal selective sampling rule.
>
> Meanwhile, for your concern about adaptivity and non-adaptitivity, the lower bound presented is true for *any adaptive selective sampling algorithm*. The definition of an adaptive selective sampling algorithm is presented prior to Definition 1 in lines 146-149 and allows for a variable sampling distribution $P_t$. Our result mirrors other lower bounds in the adaptive sampling literature that compare to an oracle sampling distribution ([11,20], reference in our paper). Thus, Theorem 1 and 2 are directly comparable.
>
> The experimental details are provided at the top of the page as an official comment for the paper. From the given plot, we can see that number of labels queried by the two selective sampling algorithms decrease as $\tau$ (number of unlabeled data seen in each round) increases. This is exactly the trade-off between label complexity $\mathcal{L}$ and sample complexity $\mathcal{U}$ characterized by our theorems.

---

> > ### Comment · Reviewer_uXS1 · 2021-08-20
> > **Thank you for the response**
> >
> > Thank you for the response. It addresses my concerns. While $P^*$ is good for serving as an intuitive motivation, it brings confusion on adaptivity. In the revision, it would be good if the author could give a sketch of proof of the lower bound, especially for adaptive $P_t$.
> >
> > I have also read other reviews and decide to keep my rating for this paper.

---

> > > ### Author Response · Authors · 2021-08-21
> > > **Thank you for your suggestion**
> > >
> > > Thank you very much for your suggestion! Since we would really like to improve our final version based on your comment, we want to clarify what is requested. Do you mean to add a sketch of the proof for Theorem 1 that is given in Appendix in the main body? Meanwhile, we hope to clarify that the lower bound is not proved by considering different types of $P_t$, but proved literally for **any selective sample algorithm**.

---

> > > > ### Comment · Reviewer_uXS1 · 2021-09-05
> > > > **Thank you for your reply**
> > > >
> > > > Yes, I hope that a sketch of the proof for Theorem 1 could be given in the main body, just to clarify that the lower bound is valid for any selective sample algorithm.
> > > >
> > > > I've read the appendix and understood that the lower bound is for any selective sample algorithm. Thank you for the clarification.

---

### Official Review · Reviewer_2aDW · 2021-07-20

**Rating:** 7
**Confidence:** 3

**Summary:**

This paper tackles the problem of best-arm identification under linear assumption. The arms are d-dimensional vectors. In addition, the learner has the possiblity to observe or not the (noisy) value of the arm x_t that is proposed by the environment at each time step. The difficulty of this problem lies in the tradeoff between the number L of labeled observations, and the total number U of labeled or unlabeled one. The goal is to minimize both L and U. In the case of i.i.d. x_t's, theoretical lower bounds on the expectation of L and U are provided for any delta-PAC algorithm ; almost matching upper bounds are given as well. A practical elimination-style algorithm is described: it depends on a subroutine solving an optimization problem by stochastic gradient ascent.

**Limitations And Societal Impact:**



**Main Review:**

This is a good paper, well written, the problem of interest is well motivated, in particular by an intuitive cocktail issue.
Strong theoretical guarantees are provided in terms of lower bounds, and of the existence of algorithms (almost) matching them. A practical algorithm is also provided.
The only flaw I see is the lack of experiments. Indeed, it would be interesting to compare (simply on simulated IID data) the L-complexity of your algorithm to the complexity of classical best-arm identification approach with no selective sampling.
Or maybe just from a theoretical point of view and in terms of lower bounds, could you discuss the power of selectiveness compared to classical best arm identification approaches without selective sampling?

**Time Spent Reviewing:**

1

---

> ### Author Response · Authors · 2021-08-11
> **Response to Reviewer 2aDW**
>
> Thank you very much for your careful review and constructive suggestions!
>
> Regarding experiments: Please see the Experiments Section addressed to all reviewers at the top.
> -  From a theoretical point of view, Theorem 1 and Theorem 2 characterize how selectiveness compares to a streaming counterpart of classical best arm identification approaches. Note that the classical best arm identification framework (e.g. [20, 10], references in our paper) assumes the availability of a fixed pool of measurement data $\mathcal{X}$, in contrast is assumed in our streaming setting that measurements come one at a time. With this in mind, Theorem 1 and 2 gives can be seen as the streaming counterparts of the matching lower and upper bounds results of past works on classical best arm identification.
> -  From an experimental point of view, experimental details are provided at the top of the page as an official comment for the whole paper. We benchmark our algorithm against the classical best-arm identification approach with no selective sampling, evaluating the sample complexity of the algorithms. We can see from the plot that the selective sampling algorithm can have much smaller label complexity.

---

### Author Response · Authors · 2021-08-11
**Experimental Setting and Results**

Let us begin by thanking the reviewers for their thoughtful comments.
While we do feel that the paper is a complete contribution as a primarily theoretical work as submitted, we value the feedback from the reviewers regarding experiments. In what follows we describe a benchmark experiment to validate the fundamental tradeoffs that we theoretically characterized.
The final version will include an experimental section, including the setting described below.

We propose the following experimental protocol where we compare 3 algorithms. The setup is the following:
- $d=2$, a two-dimensional problem
- $\mathcal{Z} = [\mathbf{e}_1, \mathbf{e}_2, (\cos(\omega), \sin(\omega))]$ for $\omega = 0.3$, where $\mathbf{e}_1, \mathbf{e}_2$ are canonical vectors (this is a variant on a common benchmark example in the linear bandits literature, from [20])
- $\theta_* = 2\mathbf{e}\_1$ and $y = x^\top \theta_* + \eta$, where $\eta\sim\mathcal{N}(0, 1)$
- The distribution $\nu$ for streaming measurements $x_t\overset{i.i.d.}{\sim}\nu$ is such that $x_t$ is sampled amount the $n$-th square roots of $1$ as $x_t = (\cos(2I\pi/N), \sin(2I\pi/N))$ such that $\mathbb{P}(I = i)\propto \cos(2i\pi/N)^2$ for all $i\in\{1\ldots, N\}$, where $N=30$

The first algorithm we run is the classical best-arm identification approach with no selective sampling (labeled with marker $\mathsf{A}$ below), the second one is our algorithm using $P^*$ (solution of optimization problem (2) in the paper, labeled with marker $\mathsf{B}$ below) and the third algorithm is our algorithm using $P_\ell$ (solution of optimization problem (3) in the paper, labeled with marker $\mathsf{C}$ below). We then run the 3 algorithms on the setup described above. We sweeped over the values of $\tau$ and plotted on the y-axis the amount of labeled data needed.

```
(label complexity (log10 scale))
7.31002486 |
7.14884556 | ⡇⠀⠀⠀⠀⠀⠀⠀⠀⠀⠀⠀⠀⠀⠀⠀⠀⠀⠀⠀⠀⠀⠀⠀⠀⠀⠀⠀⠀⠀⠀⠀⠀⠀⠀⠀⠀⠀⠀⠀
6.98766625 | ⡇⠀⠀⠀⠀⠀⠀⠀⠀⠀⠀⠀⠀⠀⠀⠀⠀⠀⠀⠀⠀⠀⠀⠀⠀⠀⠀⠀⠀⠀⠀⠀⣀A⣀⡠A⠀⠀⠀
6.82648694 | ⡇⠀⠀⠀⠀⠀⠀⠀⠀⠀⠀⠀⠀⠀⠀⠀⠀⠀⠀⠀⠀⠀⣀A⡠⠤A⠒⠊⠉A⠉⠀⠀⠀⠀⠀⠀⠀⠀
6.66530763 | ⡇⠀⠀⠀⠀⠀⠀⠀⠀⠀⠀⠀⠀⠀⠀⣀A⠔⠒⠊A⠉⠀⠀⠀⠀⠀⠀⠀⠀⠀⠀⠀⠀⠀⠀⠀⠀⠀⠀
6.50412833 | ⡇⠀⠀⠀⠀⠀⠀⠀⠀⠀⠀⣀⠤A⠊⠀⠀⠀⠀⠀⠀⠀⠀⠀⠀⠀⠀⠀⠀⠀⠀⠀⠀⠀⠀⠀⠀⠀⠀⠀
6.34294902 | ⡇⠀⠀⠀⠀⠀⠀⣀⠤⠒A⠀⠀⠀⠀⠀⠀⠀⠀⠀⠀⠀⠀⠀⠀⠀⠀⠀⠀⠀⠀⠀⠀⠀⠀⠀⠀⠀⠀⠀
6.18176971 | ⡇⠀⠀⠀⠀AA⠀⠀⠀⠀⠀⠀⠀⠀⠀⠀⠀⠀⠀⠀⠀⠀⠀⠀⠀⠀⠀⠀⠀⠀⠀⠀⠀⠀⠀⠀⠀⠀⠀
6.02059040 | ⡇⠀⠀AA⠀⠀⠀⠀⠀⠀⠀⠀⠀⠀⠀⠀⠀⠀⠀⠀⠀⠀⠀⠀⠀⠀⠀⠀⠀⠀⠀⠀⠀⠀⠀⠀⠀⠀⠀
5.85941110 | ⡇⠀AA⠀⠀⠀⠀⠀⠀⠀⠀⠀⠀⠀⠀⠀⠀⠀⠀⠀⠀⠀⠀⠀⠀⠀⠀⠀⠀⠀⠀⠀⠀⠀⠀⠀⠀⠀⠀
5.69823179 | ⡇AA⠀⠀⠀⠀⠀⠀⠀⠀⠀⠀⠀⠀⠀⠀⠀⠀⠀⠀⠀⠀⠀⠀⠀⠀⠀⠀⠀⠀⠀⠀⠀⠀⠀⠀⠀⠀⠀
5.53705248 | ⡇A⠀⠀⠀⠀⠀⠀⠀⠀⠀⠀⠀⠀⠀⠀⠀⠀⠀⠀⠀⠀⠀⠀⠀⠀⠀⠀⠀⠀⠀⠀⠀⠀⠀⠀⠀⠀⠀⠀
5.37587317 | ⡇A⠀⠀⠀⠀⠀⠀⠀⠀⠀⠀⠀⠀⠀⠀⠀⠀⠀⠀⠀⠀⠀⠀⠀⠀⠀⠀⠀⠀⠀⠀⠀⠀⠀⠀⠀⠀⠀⠀
5.21469386 | ⡇⠀⠀⠀⠀⠀⠀⠀⠀⠀⠀⠀⠀⠀⠀⠀⠀⠀⠀⠀⠀⠀⠀⠀⠀⠀⠀⠀⠀⠀⠀⠀⠀⠀⠀⠀⠀⠀⠀⠀
5.05351456 | ⡇⠀⠀⠀⠀⠀C⢄⡀⠀⠀⠀⠀⠀⠀⠀⠀⠀⠀⠀⠀⠀⠀⠀⠀⠀⠀⠀⠀⠀⠀⠀⠀⠀⠀⠀⠀⠀⠀⠀
4.89233525 | ⡇⠀⠀⠀⠀⠀⠀⠀⠈⠑C⠒⠤C⠤⢄C⣀⡀⠀⠀⠀⠀⠀⠀⠀⠀⠀⠀⠀⠀⠀⠀⠀⠀⠀⠀⠀⠀⠀
4.73115594 | ⡇B⠀⠀⠀⠀⠀⠀⠀⠀⠀⠀⠀⠀⠀⠀⠀⠀⠈⠉C⠉⠉C⠉⠉C⠉⠑⠒C⠒⠒C⠒⠒C⠀⠀⠀
4.56997663 | ⡇BB⠀⠀⠀⠀⠀⠀⠀⠀⠀⠀⠀⠀⠀⠀⠀⠀⠀⠀⠀⠀⠀⠀⠀⠀⠀⠀⠀⠀⠀⠀⠀⠀⠀⠀⠀⠀⠀
4.40879733 | ⡇⠀BBBBB⠤⢄⣀B⣀⣀B⣀⣀B⣀⣀⣀B⣀⣀B⣀⡀⠀⠀⠀⠀⠀⠀⠀⠀⠀⠀⠀⠀⠀⠀
4.24761802 | ⡇⠀⠀⠀⠀⠀⠀⠀⠀⠀⠀⠀⠀⠀⠀⠀⠀⠀⠀⠀⠀⠀⠀⠀⠀⠈B⠉⠉⠉B⠉⠉B⠉⠉B⠀⠀⠀
4.08643871 | ⡇⠀⠀⠀⠀⠀⠀⠀⠀⠀⠀⠀⠀⠀⠀⠀⠀⠀⠀⠀⠀⠀⠀⠀⠀⠀⠀⠀⠀⠀⠀⠀⠀⠀⠀⠀⠀⠀⠀⠀
-----------|-|---------|---------|---------|---------|-> (tau (*10^5))
           | 0         6         12        18        24

Legend:
-------
⠤A⠤ Naive Algorithm
⠤B⠤ Oracle Algorithm
⠤C⠤ Our Algorithm
```
From this plot, we can see that our algorithm can have much less label complexity than the naive algorithm. Meanwhile, it also shows the trade-off between label complexity $\mathcal{L}$ and sample complexity $\mathcal{U}$ characterized by our theorems.

---

### Decision · Program_Chairs · 2021-09-27

**Decision:**

Accept (Poster)

**Comment:**

This paper studies a selective sampling problem that appears in best-arm identification in linear bandits. It proves near matching upper and lower bounds on the number of labeled and unlabeled examples required, and gives a tractable implementation of the algorithm proposed.

In the initial reviews, all reviewers requested for experiments, and the experimental results in the rebuttal convinced the reviewers about the practical relevance of the proposed method. We also ask the authors to make a pass over the paper to incorporate the reviewers' presentation suggestions (in particular, experimental results, and a proof sketch of Theorem 1 suggested by Reviewer uXS1) in the final version.